# ON THE UNIVERSALITY OF SELF-SUPERVISED REPRESENTATION LEARNING

## ABSTRACT

In this paper, we investigate the characteristics that define a good representation or model. We propose that such a representation or model should possess universality, characterized by: (i) discriminability: performing well on training samples; (ii) generalization: performing well on unseen datasets; and (iii) transferability: performing well on unseen tasks with distribution shifts. Despite its importance, current self-supervised learning (SSL) methods lack explicit modeling of universality, and theoretical analysis remains underexplored. To address these issues, we aim to explore and incorporate universality into SSL. Specifically, we first revisit SSL from a task perspective and find that each mini-batch can be viewed as a multi-class classification task. We then propose that a universal SSL model should achieve: (i) learning universality by minimizing loss across all training samples, and (ii) evaluation universality by learning causally invariant representations that generalize well to unseen datasets and tasks. To quantify this, we introduce a $\sigma$-measurement that assesses the gap between the performance of SSL model and optimal task-specific models. Furthermore, to model universality, we propose the GeSSL framework. It first learns task-specific models by minimizing SSL loss, then incorporates future updates to enhance discriminability, and finally integrates these models to learn from multiple mini-batch tasks. Theoretical and empirical evidence supports the effectiveness of GeSSL.

## 1 INTRODUCTION

Self-supervised learning (SSL) has revolutionized machine learning by enabling models to learn meaningful representations from unlabeled data, thereby significantly reducing reliance on large labeled datasets (Gui et al., 2024). SSL methods are generally divided into two categories: discriminative SSL (D-SSL) and generative SSL (G-SSL). D-SSL approaches, such as SimCLR (Chen et al., 2020a), BYOL (Grill et al., 2020), and Barlow Twins (Zbontar et al., 2021), focus on distinguishing between different augmented views of the same image, learning representations by maximizing the similarity between positive pairs and minimizing it with negative ones. In contrast, G-SSL methods like MAE (Hou et al., 2022) aim to reconstruct missing or corrupted parts of the input data, learning representations by capturing inherent visual structures and patterns. Both D-SSL and G-SSL have demonstrated remarkable performance, excelling in tasks such as unsupervised learning, semi-supervised learning, transfer learning, and few-shot learning. Their capacity to learn good representations from unlabeled data has significantly advanced the field across diverse applications.

Whether using D-SSL or G-SSL methods, most research focuses on determining which factors, e.g., network architectures (Caron et al., 2021), optimization strategies (Ni et al., 2021), prior assumptions (Ermolov et al., 2021), inductive biases (Grill et al., 2020), etc., lead to effective representations or models. However, a fundamental question persists: **What exactly defines a "good" representation or model?** To address this, the common practice is to evaluate the learned representations or models on various downstream tasks, that is, if the performance is strong, the representation or model is deemed good. Yet, a key challenge remains in understanding why certain approaches result in higher performance. In other words, we often lack direct explanations of how specific methodological choices influence the quality of the representation or model. For instance, why does an asymmetric dual-branch network architecture in methods like BYOL enhance performance on downstream tasks? Similarly, why does enforcing a uniform distribution on feature representations serve as an inductive bias for obtaining good representations or models in methods like SimCLR?

To address the aforementioned challenges, this work shifts focus from considering SSL methods and their subsequent developments in terms of "what to do" to directly exploring what constitutes a good representation or model. We concentrate on the question: **What characteristics should a good representation or model possess**? Inspired by the evaluation methods of most SSL and unsupervised representation learning approaches (Chen et al., 2020a; Grill et al., 2020; Hou et al., 2022), we propose that a good representation or model should satisfy three constraints: 1) **Discriminability**: For a single task, the model should achieve the expected performance on the training set; 2) **Generalizability**: For a single task, the trained model should generalize to unseen datasets while maintaining its performance; 3) **Transferability**: The trained model should generalize to multiple different tasks while guaranteeing its performance. Based on these three constraints, we provide, for the first time in this paper, the formulation of a good representation or model—namely, **Universality**.

With the precise definition of "Universality" provided (i.e., Definition 3.1 within the main text), another significant challenge is formalizing the properties of discriminability, generalizability, and transferability within the SSL learning process. Notably, if a model can accurately predict all samples of a task based on the learned representations, it possesses good discriminability, which is reflected in a low training loss. Furthermore, as shown by Schölkopf et al. (2021) and Ahuja et al. (2023), the causality of representations is a sufficient condition for generalizability. Finally, Ni et al. (2021) demonstrates that SSL and meta-learning are closely related, and meta-learning is an effective approach to modeling transferability (Finn et al., 2017a). Therefore, designing a new SSL paradigm based on meta-learning can imbue the features learned by SSL methods with transferability. Based on these insights, we propose a novel SSL framework called GeSSL to explicitly model universality in the SSL learning process. Specifically, for discriminability, GeSSL employs the Kullback-Leibler divergence to enable the SSL model to use the future state to distill the current state, thereby achieving lower training loss. For generalizability, GeSSL extracts causal features by learning across multiple tasks. For transferability, GeSSL introduces a bi-level optimization mechanism to formulate the SSL learning behavior in a meta-learning style. In essence, **GeSSL incorporates discriminability, generalizability, and transferability into the SSL method from three dimensions: optimization objective, parameter update mechanism, and learning paradigm**.

**Our contributions**: **(i)** We theoretically define SSL universality, encompassing discriminability, generalizability, and transferability, and introduce a $\sigma$-measurement to quantify it (Sections 3.1, 3.2). **(ii)** We propose GeSSL, a novel framework that models universality through a self-motivated target for discriminability, a multi-batch collaborative update mechanism for generalizability, and a task-based bi-level learning paradigm for transferability (Section 3.3). **(iii)** Theoretical and empirical evaluations on benchmark datasets demonstrate the superior performance of GeSSL (Sections 4, 5).

## 2 REVISITING SSL FROM A TASK PERSPECTIVE

During the training phase, the data is organized into mini-batches, each denoted as $X_{tr,l} = \{x_i\}_{i=1}^{N,l}$, where $x_i$ represents the $i$-th sample in the mini-batch, $l$ is the index of mini-batch, and $N$ is the batch size. In D-SSL methods, each sample $x_i$ undergoes stochastic data augmentation to generate two augmented views, denoted as $x_i^1$ and $x_i^2$. In G-SSL methods, each sample $x_i$ is partitioned into multiple small blocks, some blocks are masked, and the remaining blocks are reassembled into a new sample $x_i^1$. The original sample is then referred to as $x_i^2$. Consequently, each augmented dataset in both D-SSL and G-SSL is represented as $X_{tr,l}^{aug} = \{x_i^1, x_i^2\}_{i=1}^{N,l}$. Each $\{x_i^1, x_i^2\}$ constitutes the $i$-th sample pair, and the SSL objective is to learn a feature extractor $f$ from these pairs.

D-SSL methods typically have two main objectives: alignment and regularization (Chen et al., 2020a; Grill et al., 2020; Zbontar et al., 2021; Oord et al., 2018; Hjelm et al., 2018). The alignment objective maximizes the similarity between paired samples in the embedding space, while the regularization objective constrains the learning behavior via inductive biases. For example, SimCLR (Chen et al., 2020a) enforces a uniform distribution over the feature representations. G-SSL methods (Hou et al., 2022) can also be viewed as implementing alignment within a pair using an encoding-decoding structure: sample $x_i^1$ is input into this structure to generate an output that is made as consistent as possible with sample $x_i^2$. Notably, alignment in D-SSL is often implemented using anchor points, where one sample in a pair is viewed as the anchor, and the training process gradually pulls the other sample towards this anchor. This concept of an anchor is also applicable to G-SSL, where $x_i^2$ is treated as the anchor, and the training process involves constraining $x_i^1$ to approach $x_i^2$.

Building upon the previous discussion, by considering the anchor as a positively labeled sample, each mini-batch in the SSL training phase can be viewed as a multi-class classification task. Specifically, the augmented dataset $X_{tr,l}^{aug} = \{x_i^1, x_i^2\}_{i=1}^{N,l}$ can be regarded as containing data from $N$ categories, where each pair $\{x_i^1, x_i^2\}$ represents the positive samples for the $i$-th category. Moreover, due to the variability of data across mini-batches, each batch corresponds to a distinct training task. More details about SSL task construction are provided in Appendix G.5.

## 3 METHODOLOGY

In this section, we first analyze the manifestation of universality in SSL and give a mathematical definition with theoretical support. Then, we propose the $\sigma$-measurement to help quantify universality. Next, we propose a novel SSL framework (GeSSL) to explicitly model universality in SSL. Finally, we illustrate the relationship between universality and GeSSL.

### 3.1 DEFINITION OF UNIVERSALITY

Wang et al. (2022) and Huang et al. (2021) theoretically proved that to obtain a good representation or model, SSL methods need to constrain the feature of samples to be discriminative, that is, compact within classes and separated between classes. However, this explanation does not clarify why an SSL model trained on one dataset (e.g., ImageNet (Deng et al., 2009a)) can generalize to different downstream tasks. Ni et al. (2021) explains the generalizability of SSL methods to different downstream tasks from the perspective of tasks but does not address the discriminability and generalizability of SSL models themselves. These gaps motivated us to propose new understandings and insights into the effectiveness of SSL methods in this paper. Therefore, we first provide a definition of a good representation or model, namely, universality. This definition suggests that a good representation should possess the properties of discriminability, generalizability, and transferability.

Considering a single mini-batch of SSL as a multi-class classification task, as mentioned in Section 2, we present the definition of *universality* as follows:

**Definition 3.1 (Universality)** *Given a set of training mini-batch tasks $X_{tr}^{aug} = \{X_{tr,l}^{aug}\}_{l=1}^{M_{tr}}$ and a set of target mini-batch tasks $X_{te}^{aug} = \{X_{te,l}^{aug}\}_{l=1}^{M_{te}}$ without class-level overlap, where each task contain $N$ samples, the model $f_\theta$ is said to exhibit universality if achieve:*

- *Learning universality: For $X_{tr}^{aug}$, the model $f_\theta$ trained on each task $X_{tr,l}^{aug}$ can achieve $\mathcal{L}(f_\theta, X_{tr,l}^{aug}) \leq \epsilon$ with iteration $t \leq T_{max}$ through few samples $|X_{tr,l}^{aug}| = \alpha N$, where $0 < \alpha \ll 1, \epsilon > 0, T_{max}$ and $\mathcal{L}(\cdot)$ denote the maximum number of iteration and the loss.*

- *Evaluation universality: For $X_{te}^{aug}$, the trained model $f_\theta$ can achieve $\mathcal{L}(f_\theta, X_{te}^{aug}) \leq \epsilon$ with all the optimal task-specific models on all the target tasks.*

For a specific mini-batch task, a model exhibits good discriminability if it can accurately predict all samples of the task based on the learned representations. This is reflected by the model $f_\theta$ achieving the lowest loss on all samples of the task. Therefore, discriminability is a key component of learning universality. According to Ahuja et al. (2020), if a model achieves good performance across multiple different tasks, it can be considered to have learned causal representations. Moreover, Schölkopf et al. (2021) and Ahuja et al. (2023) conclude that causality in representations is a sufficient condition for generalizability. Thus, a generalizable representation should be causally invariant across multiple tasks, enabling the model to achieve very low loss on these tasks using the same representation. Furthermore, since the training tasks and target tasks have no class-level overlap, the model's ability to perform well on unseen tasks demonstrates transferability. Consequently, evaluation universality encompasses both generalizability and transferability.

For the differences and relation of learning and evaluation universality: **(i) Differences**: learning universality refers to the rapid adaptation of the model to each task during training, referring to discriminability, while evaluation universality refers to the performance of the trained model in various tasks, referring to generalizability and transferability. The differences are reflected in the two stages of training vs. evaluation, and the performance of each single task vs. all tasks. **(ii) Relation**: they cover all stages of training and testing, and jointly require the model to be close to universality.

This paper introduces a fundamental concept of universality that surpasses previous works focused on universal representations (Eastwood et al., 2023; Balazevic et al., 2024) and transferability (Hsu et al., 2018; Ni et al., 2021) for two main reasons: **(i)** It incorporates both learning and evaluation universality, thereby constraining the discriminability, generalizability, and transferability of SSL, whereas prior studies focused solely on transferability. **(ii)** While earlier research emphasized task performance, particularly in meta-learning (Hsu et al., 2018; Ni et al., 2021), this work evaluates the model's effects across diverse samples and tasks. More specifically, meta-learning operates under a supervised framework, whereas GeSSL employs an unsupervised SSL approach. Additionally, the outer model's updates in meta-learning depend on the inner model's performance, often constrained by minimal samples, such as in 5-way 1-shot tasks, which can hinder discriminability. Furthermore, generalization is assessed through query sets, potentially leading to overfitting on training tasks and undermining broader generalizability.

## 3.2 MEASUREMENT OF UNIVERSALITY

In this subsection, we propose a $\sigma$-measurement to help quantify universality. According to Definition 3.1, the sufficient and necessary condition for universality is that the SSL model achieves low losses on all the training samples, unseen datasets, and unseen tasks. Thus, we propose $\sigma(f_\theta^*)$ to measure the performance gap between the trained SSL model $f_\theta^*$ and the task-specific optimal models (ground-truth with 100% accuracy, $f_{\phi_l}^*$). In other words, the more universality $f_\theta^*$ is, the more accurate the output is, the closer the effect on a specific task is to $f_{\phi_l}^*$. Thus, we propose the following definition:

**Definition 3.2 ($\sigma$-measurement)** *Given a set of unseen mini-batch tasks $X_{te}^{aug} = \{X_{te,l}^{aug}\}_{l=1}^{M_{te}}$, assume that the optimal parameter $\theta^*$ is independent of $X_{te}^{aug}$, i.e., not change due to the distribution of test tasks, and the covariance of $\theta^*$ satisfies $\mathrm{Cov}[\theta^*] = (R^2/d)\mathcal{I}_d$, where $R$ is a constant, $d$ is the dimension of the model parameter, and $\mathcal{I}_d$ is a identity matrix, the error rate $\sigma(f_\theta^*)$ is:*

$$\sigma(f_\theta^*) = \sum_{X_{te,l}^{aug} \in X_{te}^{aug}} \sum_{x \in X_{te,l}^{aug}} \mathrm{KL}(\pi(f_\theta^*(x))|\pi(f_{\phi_l}^*(x))), \tag{1}$$

*where $\mathrm{KL}(p|q) = \int p(x) \log\left(\frac{p(x)}{q(x)}\right) dx$ is the calculation of Kullback-Leibler Divergence which is estimated via variational inference, $\pi$ is the auxiliary classification head employed to generating the class probability distribution.*

Based on Section 2, a mini-batch can be regarded as an $N$-class classification task, where the samples in each pair are the positive samples of a particular class. Therefore, we can employ a classifier $\pi$ to output the class probability distribution for each sample. Also, this measurement directly inspires the design of the objective (Eq.4). More details are provided in Appendix B.1, including the analyses of assumptions, the detailed calculation of KL term, and the detailed implementation, etc. Meanwhile, we also conduct experiments to evaluate universality with $\sigma$-measurement in Appendix F.4.

## 3.3 EXPLICIT MODELING OF UNIVERSALITY

Based on the Definition 3.1 and 3.2, in this section, we explicitly model universality into SSL and propose a general SSL framework as shown in Figure 1, called GeSSL. It learns universal knowledge through a bi-level optimization over a set of SSL tasks conducted as described in Section 2. The whole learning process of GeSSL can be divided into three steps:

**Step 1:** In this step, GeSSL aims to learn task-specific models by minimizing SSL loss over mini-batches. The learning process of each mini-batch can be expressed as:

$$f_\theta^l \leftarrow f_\theta - \alpha \nabla_{f_\theta} \ell(f_\theta, X_{tr,l}^{aug}), \tag{2}$$

where $\ell(f_\theta, X_{tr,l}^{aug})$ denotes the SSL loss, utilized in methods such as SimCLR, BYOL, and Barlow Twins. Here, $\alpha$ is the learning rate, $f_\theta$ is the initialized neural network, $f_\theta^l$ is the task-specific model for the mini-batch task $X_{tr,l}^{aug}$.

Unlike existing SSL methods, we input $M$ mini-batches simultaneously in this step, resulting in $M$ task-specific models. During training, $f_\theta^l$ typically undergoes $K$ updates, executing Eq.2 $K$ times. This step is motivated by: (i) simulating a multi-task training environment to facilitate multi-task

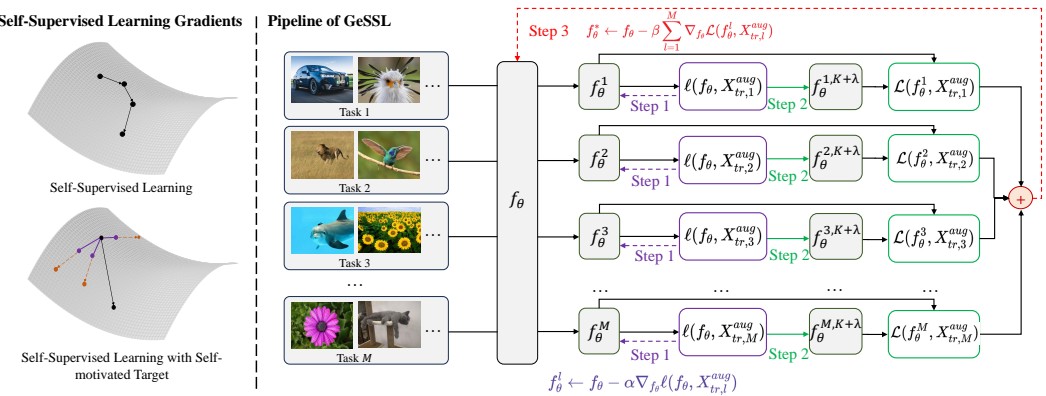

Figure 1: Overview of GeSSL and the learning gradients. The purple line refers to **Step 1**, the green line refers to **Step 2**, and the red line refers to **Step 3**. The **pseudo-code** is shown in Appendix A.

learning; and (ii) improving the discriminability of task-specific models, as multiple updates lead to a smaller training loss for $f_\theta^l$. Also, from a bi-level optimization perspective, this step can be regarded as the inner-loop optimization.

**Step 2:** Given the constraints of mini-batch size and training complexity, $K$ in **Step 1** is typically set to 1, leading to underfitting (Ravi & Larochelle, 2016; Wang et al., 2024a; Nakamura & Harada, 2019), which compromises the discriminability of task-specific models. To address this, we introduce $\sigma$-measurement and propose the following optimization objective:

$$\mathcal{L}(f_\theta^l, X_{tr,l}^{aug}) = \sum_{x \in X_{tr,l}^{aug}} \mathrm{KL}(\pi(f_\theta^{l,K+\lambda}(x))|\pi(f_\theta^{l,K}(x))), \tag{3}$$

where $f_\theta^{l,K}$ is the obtained $f_\theta^l$ that performs Eq.2 $K$ times and $f_\theta^{l,K+\lambda}$ is the obtained $f_\theta^l$ that performs Eq.2 another $\lambda$ times further. We call $f_\theta^{l,K+\lambda}$ the self-motivated target. Here, an auxiliary classification head is employed to implement $\pi$, generating the class probability distribution.

When $\pi(f_\theta^{l,K+\lambda}(x))$ is fixed, Eq.3 can be interpreted as distilling the current model using a more discriminative one, thereby enhancing its discriminability. Instead of directly performing $K + \lambda$ updates in **Step 1**, we use Eq.3 to improve the discriminability of the task-specific model. This approach is chosen because (i) the optimal $K + \lambda$ is unknown, and (ii) as noted in Zou et al. (2022), Wang et al. (2024a), and Chen et al. (2022), excessively large $K + \lambda$ values may lead to overfitting. Compared to direct updates, Eq.3 offers better control and acts primarily as a regularizer, reducing constraints on the task-specific model and partially mitigating overfitting.

**Step 3:** In this step, GeSSL aims to learn the final model $f_\theta^*$ based on task-specific models and Eq.3. The learning process can be expressed as:

$$f_\theta^* \leftarrow f_\theta - \beta \sum_{l=1}^{M} \nabla_{f_\theta} \mathcal{L}(f_\theta^l, X_{tr,l}^{aug}), \tag{4}$$

where $\beta$ is the learning rate, $\mathcal{L}(f_\theta^l, X_{tr,l}^{aug})$ is given in Eq.3, and $\pi(f_\theta^{l,K+\lambda}(x))$ is fixed.

First, as shown in Eq.4, the goal of GeSSL is to derive $f_\theta^*$, which is based on multiple task-specific models and mini-batch tasks, framing the learning process as a multi-task process. Second, from Eq.2, $f_\theta^l$ is a function of the initialized neural network $f_\theta$. Moreover, from Eq.3, $\mathcal{L}(f_\theta^l, X_{tr,l}^{aug})$ is a function of $f_\theta^l$, making it a first-order gradient function of $f_\theta$. Consequently, the optimization of $f_\theta^*$ is a second-order gradient-based process with respect to $f_\theta$. Finally, from a bi-level optimization perspective, this step corresponds to outer-loop optimization.

In summary, GeSSL initially constructs a series of mini-batch tasks to learn intermediate task-specific models. It then introduces a distillation loss, whose minimization enhances the performance of these intermediate models. Finally, by simulating the multi-task learning paradigm, minimizing the

distillation loss, and employing a bi-level optimization mechanism, GeSSL yields the final model. Besides, the key idea of GeSSL is concluded as: **Once a model reaches optimality, future updates will no longer affect it. However, if the model remains suboptimal, these updates will enhance its performance. Step 3 essentially constrains the model to achieve optimality, as its optimization process aims to align the performance of the current model with that of the future model obtained through further updates. This alignment is only possible if $f_\theta$ reaches optimality.**

### 3.4 The Relationship between Universality and GeSSL

According to Subsection 3.3, the objective function of GeSSL can be written as:

$$\min_{f_\theta} \sum_{l=1}^M \mathcal{L}(f_\theta^l, X_{tr,l}^{aug}), \qquad \text{s.t. } f_\theta^l = \arg\min_{f_\theta} \ell(f_\theta, X_{tr,l}^{aug}), l = 1, ..., M. \tag{5}$$

Based on the Definition 3.1, we obtain that modeling discriminability, generalizability, and transferability is the key to modeling universality. From Eq.5, the constraints on the three properties of universality within GeSSL are reflected in the following:

- **Discriminability**: Based on the illustration presented in the last paragraph of Subsection 3.3, we can conclude that optimizing Eq.5 enables GeSSL to learn a better model compared with traditional SSL methods that only update Eq.2 once. The key reason is that we minimize the term $\mathcal{L}(f_\theta^l, X_{tr,l}^{aug})$. Thus, we can safely assert that GeSSL enhances model discriminability by minimizing the loss $\sum_{l=1}^M \mathcal{L}(f_\theta^l, X_{tr,l}^{aug})$.

- **Generalizability**: As shown in Subsection 3.1, training a model across different tasks enables it to extract causal features from the data, thereby endowing the model with generalizability. Specifically, as illustrated in **Step 1** and **Step 3**, GeSSL learns $f_\theta$ through multiple mini-batch tasks. To ensure that the final model achieves optimal performance on all training tasks, GeSSL proposes updating the network parameters using a second-order gradient method. Therefore, we conclude that GeSSL models generalizability through a parameter update mechanism involving second-order gradients.

- **Transferability**: From Figure 1, we observe that the training process of GeSSL can be regarded as an episodic learning process. Specifically, each episode of GeSSL consists of $M$ mini-batch tasks, and the entire learning process can be divided into multiple episodes. Based on Section 2, we consider the learning process of GeSSL as estimating the true task distribution from discrete training tasks, which enables the GeSSL model to generalize to new, unseen tasks (i.e., test tasks). Therefore, we conclude that GeSSL achieves model transferability through its learning paradigm.

Finally, GeSSL models discriminability, generalizability, and transferability into the SSL method from three dimensions: optimization objective, parameter update mechanism, and learning paradigm.

## 4 Theoretical Evaluation

In this section, we provide performance guarantees for GeSSL. Specifically, we restrict our attention to the noise-less setting (true expectation) and analyze how the performance around $f_\theta^l$ changes by updating $f_\theta$. We assume the output of $\mathrm{KL}(\cdot)$ is differentiable and convex with a minimum value of 0.

**Theorem 4.1** *Let $\tilde{f}_\theta$ and $f_\theta$ be SSL models before and after learning universal knowledge based on Eq.4, and $\mathrm{KL}^f(f_{\theta_1}(X_{tr,l}^{aug}), f_{\theta_2}(X_{tr,l}^{aug}))$ be the the abbreviation of $\sum_{x \in X_{tr,l}^{aug}} \mathrm{KL}(\pi(f_{\theta_1}(x))|\pi(f_{\theta_2}(x)))$, the update process for each mini-batch $X_{tr,l}^{aug}$ satisfies:*

$$\tilde{f}_\theta - f_\theta = \frac{\beta}{\alpha} \mathrm{KL}^f(f_\theta^{l,K+\lambda}(X_{tr,l}^{aug}), f_\theta^{l,K} - \alpha \mathcal{G}^\top \mathrm{g}(X_{tr,l}^{aug}))$$
$$- \frac{\beta}{\alpha} \mathrm{KL}^f(f_\theta^{l,K+\lambda}(X_{tr,l}^{aug}), f_\theta^{l,K}(X_{tr,l}^{aug})) + o(\beta(\alpha + \beta)), \tag{6}$$

*where $\mathcal{G}^\top = \mathcal{M}^\top \mathcal{M} \in \mathbb{R}^{n_\theta \times n_\theta}$ with the (transposed) Jacobian $\mathcal{M}$ of $f_\theta^{l,K}$. When the learning rates $\alpha$ and $\beta$ are sufficiently small, there exists a self-motivated target that yields $\tilde{f}_\theta - f_\theta \leq o(\beta(\alpha + \beta))$.*

Table 1: The Top-1 and Top-5 classification accuracies of linear classifier on the ImageNet-100 dataset and ImageNet dataset (200 Epochs) with ResNet-50 as feature extractor.

| Method | ImageNet-100 | | ImageNet | |
|---|---|---|---|---|
| | Top-1 | Top-5 | Top-1 | Top-5 |
| SimCLR Chen et al. (2020a) | 70.15 ± 0.16 | 89.75 ± 0.14 | 68.32 ± 0.31 | 89.76 ± 0.23 |
| MoCo Chen et al. (2020b) | 72.80 ± 0.12 | 91.64 ± 0.11 | 67.55 ± 0.27 | 88.42 ± 0.11 |
| SimSiam Chen & He (2021) | 73.01 ± 0.21 | 92.61 ± 0.27 | 70.02 ± 0.14 | 88.76 ± 0.23 |
| Barlow Twins Zbontar et al. (2021) | 75.97 ± 0.23 | 92.91 ± 0.19 | 69.94 ± 0.32 | 88.97 ± 0.27 |
| SwAV Caron et al. (2020) | 75.78 ± 0.16 | 92.86 ± 0.15 | 69.12 ± 0.24 | 89.38 ± 0.20 |
| DINO Caron et al. (2021) | 75.43 ± 0.18 | 93.32 ± 0.19 | 70.58 ± 0.24 | 91.32 ± 0.27 |
| W-MSE Ermolov et al. (2021) | 76.01 ± 0.27 | 93.12 ± 0.21 | 70.85 ± 0.31 | 91.57 ± 0.20 |
| RELIC v2 Tomasev et al. (2022) | 75.88 ± 0.15 | 93.52 ± 0.13 | 70.98 ± 0.21 | 91.15 ± 0.26 |
| LMCL Chen et al. (2021) | 75.89 ± 0.19 | 92.89 ± 0.28 | 70.83 ± 0.26 | 90.04 ± 0.21 |
| ReSSL Zheng et al. (2021) | 75.77 ± 0.21 | 92.91 ± 0.27 | 69.92 ± 0.24 | 91.25 ± 0.12 |
| CorInfoMax Ozsoy et al. (2022) | 75.54 ± 0.20 | 92.23 ± 0.25 | 70.83 ± 0.15 | 91.53 ± 0.22 |
| MEC Liu et al. (2022a) | 75.38 ± 0.17 | 92.84 ± 0.20 | 70.34 ± 0.27 | 91.25 ± 0.38 |
| VICRegL Bardes et al. (2022) | 75.96 ± 0.19 | 92.97 ± 0.26 | 70.24 ± 0.27 | 91.60 ± 0.24 |
| SimCLR + GeSSL | 72.43 ± 0.18 | 91.87 ± 0.21 | 69.65 ± 0.16 | 90.98 ± 0.19 |
| MoCo + GeSSL | 73.78 ± 0.19 | 93.28 ± 0.23 | 69.47 ± 0.28 | 90.34 ± 0.28 |
| SimSiam + GeSSL | 75.48 ± 0.19 | 94.83 ± 0.31 | 71.74 ± 0.19 | 89.28 ± 0.30 |
| Barlow Twins + GeSSL | 76.83 ± 0.19 | 93.23 ± 0.18 | 71.89 ± 0.22 | 89.32 ± 0.14 |
| SwAV + GeSSL | 76.38 ± 0.20 | **95.47 ± 0.19** | 71.47 ± 0.10 | 90.28 ± 0.28 |
| DINO + GeSSL | 76.84 ± 0.25 | 94.98 ± 0.24 | 72.84 ± 0.19 | **93.54 ± 0.18** |
| VICRegL + GeSSL | **77.58 ± 0.22** | 95.46 ± 0.15 | **73.54 ± 0.29** | 93.17 ± 0.30 |

Table 2: The semi-supervised learning accuracies (± 95% confidence interval) on the ImageNet dataset with the ResNet-50 pre-trained on the Imagenet dataset.

| Method | Epochs | 1% | | 10% | |
|---|---|---|---|---|---|
| | | Top-1 | Top-5 | Top-1 | Top-5 |
| MoCo Chen et al. (2020b) | 200 | 43.8 ± 0.2 | 72.3 ± 0.2 | 61.9 ± 0.1 | 84.6 ± 0.2 |
| BYOL Grill et al. (2020) | 200 | 54.8 ± 0.2 | 78.8 ± 0.1 | 68.0 ± 0.2 | 88.5 ± 0.2 |
| MoCo + GeSSL | 200 | 46.2 ± 0.3 | 74.3 ± 0.2 | 63.4 ± 0.2 | 85.3 ± 0.1 |
| BYOL + GeSSL | 200 | **56.9 ± 0.2** | **79.6 ± 0.1** | **70.8 ± 0.2** | **89.9 ± 0.2** |
| SimCLR Chen et al. (2020a) | 1000 | 48.3 ± 0.2 | 75.5 ± 0.1 | 65.6 ± 0.1 | 87.8 ± 0.2 |
| MoCo Chen et al. (2020b) | 1000 | 52.3 ± 0.1 | 77.9 ± 0.2 | 68.4 ± 0.1 | 88.0 ± 0.2 |
| BYOL Grill et al. (2020) | 1000 | 56.3 ± 0.2 | 79.6 ± 0.2 | 69.7 ± 0.2 | 89.3 ± 0.1 |
| SimSiam Chen & He (2021) | 1000 | 54.9 ± 0.2 | 79.5 ± 0.2 | 68.0 ± 0.1 | 89.0 ± 0.3 |
| Barlow Twins Zbontar et al. (2021) | 1000 | 55.0 ± 0.1 | 79.2 ± 0.1 | 67.7 ± 0.2 | 89.3 ± 0.2 |
| RELIC v2 Tomasev et al. (2022) | 1000 | 55.2 ± 0.2 | 80.0 ± 0.1 | 68.0 ± 0.2 | 88.9 ± 0.2 |
| LMCL Chen et al. (2021) | 1000 | 54.8 ± 0.2 | 79.4 ± 0.2 | 70.3 ± 0.1 | 89.9 ± 0.2 |
| ReSSL Zheng et al. (2021) | 1000 | 55.0 ± 0.1 | 79.6 ± 0.3 | 69.9 ± 0.1 | 89.7 ± 0.1 |
| SSL-HSIC Li et al. (2021) | 1000 | 55.4 ± 0.3 | 80.1 ± 0.2 | 70.4 ± 0.1 | 90.0 ± 0.1 |
| CorInfoMax Ozsoy et al. (2022) | 1000 | 55.0 ± 0.2 | 79.6 ± 0.3 | 70.3 ± 0.2 | 89.3 ± 0.2 |
| MEC Liu et al. (2022a) | 1000 | 54.8 ± 0.1 | 79.4 ± 0.2 | 70.0 ± 0.1 | 89.1 ± 0.1 |
| VICRegL Bardes et al. (2022) | 1000 | 54.9 ± 0.1 | 79.6 ± 0.2 | 67.2 ± 0.1 | 89.4 ± 0.2 |
| SimCLR + GeSSL | 1000 | 50.4 ± 0.2 | 77.5 ± 0.1 | 66.9 ± 0.2 | 89.4 ± 0.3 |
| MoCo + GeSSL | 1000 | 53.5 ± 0.2 | 78.7 ± 0.1 | 70.9 ± 0.2 | 89.0 ± 0.2 |
| BYOL + GeSSL | 1000 | **58.7 ± 0.3** | **81.4 ± 0.2** | **71.5 ± 0.1** | 90.7 ± 0.2 |
| Barlow Twins + GeSSL | 1000 | 57.4 ± 0.2 | 80.2 ± 0.1 | 68.8 ± 0.2 | **91.4 ± 0.2** |

Table 3: The results of transfer learning on object detection and instance segmentation with C4-backbone as the feature extractor. "AP" is the average precision, "$AP_N$" represents the average precision when the IoU (Intersection and Union Ratio) threshold is $N\%$.

| Method | VOC 07 detection | | | VOC 07+12 detection | | | COCO detection | | | COCO instance segmentation | | |
|---|---|---|---|---|---|---|---|---|---|---|---|---|
| | $AP_{50}$ | AP | $AP_{75}$ | $AP_{50}$ | AP | $AP_{75}$ | $AP_{50}$ | AP | $AP_{75}$ | $AP_{50}^{mask}$ | $AP^{mask}$ | $AP_{75}^{mask}$ |
| Supervised | 74.4 | 42.4 | 42.7 | 81.3 | 53.5 | 58.8 | 58.2 | 38.2 | 41.2 | 54.7 | 33.3 | 35.2 |
| SimCLR Chen et al. (2020a) | 75.9 | 46.8 | 50.1 | 81.8 | 55.5 | 61.4 | 57.7 | 37.9 | 40.9 | 54.6 | 33.3 | 35.3 |
| MoCo Chen et al. (2020b) | 77.1 | 46.8 | 52.5 | 82.5 | 57.4 | 64.0 | 58.9 | 39.3 | 42.5 | 55.8 | 34.4 | 36.5 |
| BYOL Grill et al. (2020) | 77.1 | 47.0 | 49.9 | 81.4 | 55.3 | 61.1 | 57.8 | 37.9 | 40.9 | 54.3 | 33.2 | 35.0 |
| SimSiam Chen & He (2021) | 77.3 | 48.5 | 52.5 | 82.4 | 57.0 | 63.7 | 59.3 | 39.2 | 42.1 | 56.0 | 34.4 | 36.7 |
| Barlow Twins Zbontar et al. (2021) | 75.7 | 47.2 | 50.3 | 82.6 | 56.8 | 63.4 | 59.0 | 39.2 | 42.5 | 56.0 | 34.3 | 36.5 |
| SwAV Caron et al. (2020) | 75.5 | 46.5 | 49.6 | 82.6 | 56.1 | 62.7 | 58.6 | 38.4 | 41.3 | 55.2 | 33.8 | 35.9 |
| MEC Liu et al. (2022a) | 77.4 | 48.3 | 52.3 | 82.8 | 57.5 | 64.5 | 59.8 | 39.8 | 43.2 | 56.3 | 34.7 | 36.8 |
| RELIC v2 Tomasev et al. (2022) | 76.9 | 48.0 | 52.0 | 82.1 | 57.3 | 63.9 | 58.4 | 39.3 | 42.3 | 56.0 | 34.6 | 36.3 |
| CorInfoMax Ozsoy et al. (2022) | 76.8 | 47.6 | 52.2 | 82.4 | 57.0 | 63.4 | 58.8 | 39.6 | 42.5 | 56.2 | 34.8 | 36.5 |
| VICRegL Bardes et al. (2022) | 75.9 | 47.4 | 52.3 | 82.6 | 56.4 | 62.9 | 59.2 | 38.9 | 42.1 | 56.5 | 35.1 | 36.8 |
| SimCLR + GeSSL | 77.4 | 49.1 | 51.2 | 84.3 | 57.4 | 62.9 | 58.5 | 39.6 | 43.1 | 56.3 | 35.0 | 36.1 |
| MoCo + GeSSL | 78.5 | 49.3 | **53.9** | **85.2** | **59.3** | **65.5** | 60.7 | 41.6 | 44.2 | **58.2** | 36.1 | 38.0 |
| BYOL + GeSSL | 78.5 | 49.4 | 51.7 | 83.5 | 57.9 | 63.2 | 59.8 | 39.1 | 43.0 | 55.6 | 34.6 | 37.9 |
| SimSiam + GeSSL | **79.3** | **50.0** | 53.7 | 84.6 | 58.9 | 65.2 | 61.5 | 41.7 | 43.4 | 57.6 | 36.5 | 39.0 |
| SwAV + GeSSL | 77.2 | 48.8 | 51.0 | 84.1 | 57.5 | 65.0 | 61.4 | 39.7 | 43.3 | 56.2 | 36.5 | 37.4 |
| VICRegL + GeSSL | 77.4 | 49.7 | 53.2 | 84.5 | 58.0 | 64.7 | **62.1** | **41.9** | **44.6** | 58.1 | **36.8** | **38.4** |

The theorem shows that any self-motivated target, even in the absence of noise, can drive model updates towards better performance, i.e., as $\alpha$ and $\beta$ become small or even zero, we get $\tilde{f}_\theta - f_\theta \leq 0$ where $\tilde{f}_\theta$ achieves performance improvements over previous $f_\theta$. By using KL divergence to quantify the difference between the model's output distributions, the theorem ensures that controlled gradient updates gradually reduce the model's deviation from the target distribution. As the parameter $\beta$ decreases, the KL divergence term diminishes, indicating the model's steady convergence towards a more optimal state. The proof of this theorem and more analyses are provided in Appendix B.

## 5 EMPIRICAL EVALUATION

In this section, we first introduce the datasets in Section 5.1. Next, we conduct experiments on multiple scenarios for evaluation in Sections 5.2-5.5, including unsupervised learning, semi-supervised learning, transfer learning, and few-shot learning. We introduce the experimental setups in the corresponding sections. More details are provided in Appendix C. Finally, we perform ablation studies in Section 5.6. All results reported are the averages of five runs performed on NVIDIA RTX 4090 GPUs. More experiments are shown in Appendix F and G due to space limitations.

### 5.1 BENCHMARK DATASETS

For unsupervised learning, we evaluate GeSSL on CIFAR-10 Krizhevsky et al. (2009), CIFAR-100 Krizhevsky et al. (2009), STL-10 Coates et al. (2011), Tiny ImageNet Le & Yang (2015), ImageNet-100 Tian et al. (2020a) and ImageNet Deng et al. (2009a). For semi-supervised learning, we evaluate

Table 4: Few-shot learning accuracies ($\pm$ 95% confidence interval) on miniImageNet, Omniglot, and CIFAR-FS with C4. See Appendix E for the baselines' details, and Appendix F for full results.

| Method | Omniglot | | | *mini*ImageNet | | | CIFAR-FS | | |
|---|---|---|---|---|---|---|---|---|---|
| | (5,1) | (5,5) | (20,1) | (5,1) | (5,5) | (20,1) | (5,1) | (5,5) | (20,1) |
| *Unsupervised Few-shot Learning* | | | | | | | | | |
| CACTUs Hsu et al. (2018) | 65.29 ± 0.21 | 86.25 ± 0.19 | 49.54 ± 0.21 | 39.32 ± 0.28 | 53.54 ± 0.27 | 31.99 ± 0.29 | 40.02 ± 0.23 | 58.16 ± 0.22 | 35.88 ± 0.25 |
| UMTRA Khodadadeh et al. (2019) | 83.32 ± 0.37 | 94.23 ± 0.35 | 75.84 ± 0.34 | 39.23 ± 0.34 | 51.78 ± 0.32 | 30.27 ± 0.34 | 41.61 ± 0.40 | 60.55 ± 0.38 | 37.10 ± 0.39 |
| LASIUM Khodadadeh et al. (2020) | 82.38 ± 0.36 | 95.11 ± 0.36 | 70.23 ± 0.36 | 42.12 ± 0.38 | 54.98 ± 0.37 | 34.26 ± 0.35 | 45.33 ± 0.32 | 62.65 ± 0.33 | 38.40 ± 0.33 |
| SVEBM Kong et al. (2021) | 87.07 ± 0.28 | 94.13 ± 0.27 | 73.33 ± 0.28 | 44.74 ± 0.29 | 58.38 ± 0.28 | 39.71 ± 0.30 | 47.24 ± 0.25 | 63.10 ± 0.28 | 40.10 ± 0.28 |
| GMVAE Lee et al. (2021) | 90.89 ± 0.32 | 96.05 ± 0.32 | 81.51 ± 0.33 | 42.28 ± 0.36 | 56.97 ± 0.38 | 39.83 ± 0.36 | 47.45 ± 0.36 | 63.20 ± 0.35 | 41.55 ± 0.35 |
| PsCo Jang et al. (2023) | **96.18 ± 0.21** | 98.22 ± 0.23 | 89.32 ± 0.23 | 46.35 ± 0.24 | 63.05 ± 0.23 | 40.84 ± 0.27 | 51.77 ± 0.27 | 69.66 ± 0.26 | 45.08 ± 0.27 |
| *Self-supervised Learning* | | | | | | | | | |
| SimCLR Chen et al. (2020a) | 90.83 ± 0.21 | 97.67 ± 0.21 | 81.67 ± 0.23 | 42.32 ± 0.38 | 51.10 ± 0.37 | 36.36 ± 0.36 | 49.44 ± 0.30 | 60.02 ± 0.29 | 39.29 ± 0.30 |
| MoCo Chen et al. (2020b) | 87.83 ± 0.20 | 95.52 ± 0.19 | 80.03 ± 0.21 | 40.56 ± 0.34 | 49.41 ± 0.37 | 36.52 ± 0.38 | 45.35 ± 0.31 | 58.11 ± 0.32 | 37.89 ± 0.32 |
| SwAV Caron et al. (2020) | 91.28 ± 0.19 | 97.21 ± 0.20 | 82.02 ± 0.20 | 44.39 ± 0.36 | 54.91 ± 0.36 | 37.13 ± 0.37 | 49.39 ± 0.29 | 62.20 ± 0.30 | 40.19 ± 0.32 |
| SimCLR + GeSSL | 94.15 ± 0.26 | **98.46 ± 0.15** | 90.15 ± 0.19 | 46.34 ± 0.25 | 62.18 ± 0.20 | 39.28 ± 0.19 | **52.18 ± 0.32** | 67.01 ± 0.19 | 46.23 ± 0.27 |
| MoCo + GeSSL | 92.78 ± 0.24 | 97.26 ± 0.23 | 88.01 ± 0.24 | 46.66 ± 0.25 | 60.48 ± 0.25 | 40.38 ± 0.19 | 50.98 ± 0.24 | 65.56 ± 0.11 | 44.23 ± 0.17 |
| SwAV + GeSSL | 95.48 ± 0.16 | 97.98 ± 0.20 | **91.17 ± 0.25** | **48.15 ± 0.18** | **63.28 ± 0.09** | **41.32 ± 0.28** | 51.98 ± 0.31 | **69.28 ± 0.29** | **47.28 ± 0.18** |

GeSSL on ImageNet Deng et al. (2009a). For transfer learning, we select PASCAL VOC Everingham et al. (2010) and COCO Lin et al. (2014a) for analysis. For few-shot learning, we select Omniglot Lake et al. (2019), miniImageNet Vinyals et al. (2016a), and CIFAR-FS Bertinetto et al. (2018). More details are provided in Appendix D.

## 5.2 Unsupervised Learning

**Experimental setup.** We adopt the most commonly used protocol Chen et al. (2020a), freezing the feature extractor and training a supervised linear classifier on top of it. We use the Adam optimizer Kingma & Ba (2014) with Momentum and weight decay set at $0.8$ and $10^{-4}$. The linear classifier runs for 500 epochs with a batch size of 128 and a learning rate that starts at $5 \times 10^{-2}$ and decays to $5 \times 10^{-6}$. We use ResNet-18 as the feature extractor for small-scale datasets (CIFAR-10, CIFAR-100, STL-10, and Tiny ImageNet), while using ResNet-50 for the medium-scale dataset (ImageNet-100) and the large-scale dataset (ImageNet). The $\lambda$ of the self-motivated target is set to 10.

Table 5: Ablation study of hyperparameter $\lambda$ for self-motivated target with different $K$ on miniImageNet.

| K | | | | λ | | | | K+λ | | | | | | | | | | Acc (%) | Training Time (h) |
|---|---|---|---|---|---|---|---|---|---|---|---|---|---|---|---|---|---|---|---|
| 1 | 5 | 10 | 15 | 1 | 5 | 10 | 15 | 2 | 6 | 10 | 11 | 15 | 16 | 20 | 25 | 30 | | |
| ✓ | | | | ✓ | | | | ✓ | | | | | | | | | 41.1 ± 0.3 | 3.15 |
| | ✓ | | | | ✓ | | | | ✓ | | | | | | | | 44.3 ± 0.4 | 3.28 |
| | | ✓ | | | | ✓ | | | | ✓ | | | | | | | 46.5 ± 0.3 | 3.40 |
| | | | ✓ | | | | ✓ | | | | | ✓ | | | | | 45.7 ± 0.3 | 3.51 |
| ✓ | | | | ✓ | | | | | | | ✓ | | | | | | 45.4 ± 0.2 | 3.69 |
| | ✓ | | | | | ✓ | | | | | | | ✓ | | | | 47.0 ± 0.3 | 3.80 |
| | | ✓ | | | | | ✓ | | | | | | | ✓ | | | 46.9 ± 0.3 | 4.01 |
| | ✓ | | | | ✓ | | | | | | | | | | ✓ | | 47.1 ± 0.3 | 4.27 |
| | ✓ | | | | | | ✓ | | | | | | | | ✓ | | 46.8 ± 0.4 | 4.52 |
| | | ✓ | | | ✓ | | | | | | | | | | | ✓ | 47.2 ± 0.3 | 5.07 |

**Results.** Table 1 shows the top-1 and top-5 linear classification accuracies on ImageNet-100 and ImageNet. We can observe that applying GeSSL significantly outperforms the state-of-the-art (SOTA) methods on all datasets and all the SSL baselines. The results demonstrate its ability to enhance the performance of SSL. The full results and more analyses are provided in Appendix F.1.

## 5.3 Semi-supervised Learning

**Experimental setup.** We adopt the commonly used protocol Zbontar et al. (2021) and create two balanced subsets by sampling 1% and 10% of the training dataset. We fine-tune the models for 50 epochs with different learning rates, i.e., 0.05 and 1.0 for the classifier and 0.0001 and 0.01 for the backbone on the 1% and 10% subsets. The $\lambda$ is set to 10 with $K = 1$.

**Results.** Table 2 shows the results on ImageNet. We can observe that the performance after applying our GeSSL is superior to the SOTA methods. Specifically, when only 1% of the labels are available in 1000 epochs, the improvement brought by GeSSL reaches an average of 2.7% on Top-1 and an average of 1.4% on Top-5. When only 10% of the labels are available in 1000 epochs, applying GeSSL yields better top-1 and top-5 accuracy, increasing by 1.3% and 2.0%, respectively.

## 5.4 Transfer Learning

We construct three experiments for transfer learning, including the most commonly used object detection and instance segmentation protocol Chen et al. (2020a); Zbontar et al. (2021), transfer to other domains, and transfer on video-based tasks. The last two scenarios are shown in Appendix F.2.

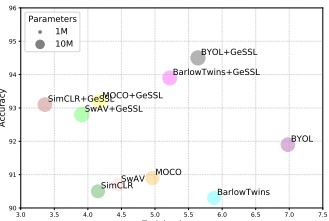

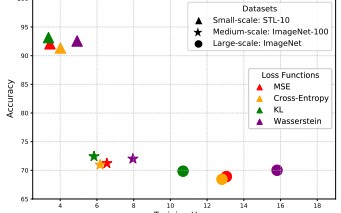

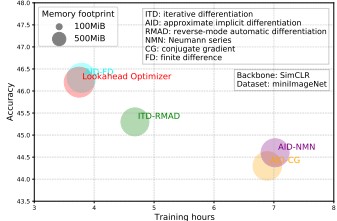

Figure 2: Model efficiency with the same batch size and official code configuration.

Figure 3: Ablation study of the loss functions. See Appendix G.3 for more details.

Figure 4: Implementation of bi-level optimization. See Appendix G.4 for more details.

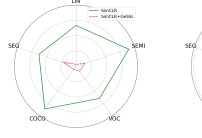
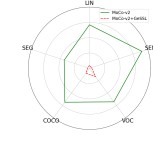
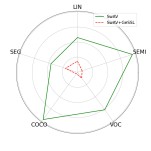
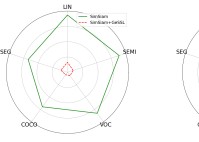
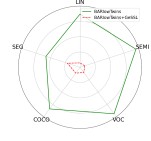

(a) SimCLR     (b) BYOL     (c) MoCo     (d) SwAV     (e) SimSiam     (f) BarlowTwins

Figure 5: Universality performance of different models on five image-based tasks.

**Experimental setup.** We use Faster R-CNN Ren et al. (2015) for VOC detection and use Mask R-CNN He et al. (2017) and a $1\times$ schedule for COCO detection and segmentation with the same C4-backbone Wu et al. (2019). During training, we train the Faster R-CNN model on the VOC 07+12 set (16K images) and reduce the initial learning rate by 10 at 18K and 22K iterations, while training on the VOC 07 set (5K images) with fewer iterations. For the Mask R-CNN, we train it on the COCO 2017 train split and report the results on the val split. More details are shown in Appendix F.2.

**Results.** Table 3 shows the transfer learning results. The results show the great performance improvements achieved by GeSSL: (i) for the VOC 07 detection task, SimSiam + GeSSL and MoCo + GeSSL achieve the best performance; (ii) for the VOC 07+12 detection task, MoCo + GeSSL outperforms other methods; (iii) for the COCO detection task, GeSSL applied to VICRegL obtains the best results; and (iv) for the COCO instance segmentation task, MoCo + GeSSL and VICRegL + GeSSL obtain the best results. Therefore, our GeSSL continues to exhibit remarkable performance.

## 5.5 FEW-SHOT LEARNING

**Experimental setup.** We adopt the commonly used protocol Jang et al. (2023) and select on three benchmarks, i.e., miniImageNet, Omniglot, and CIFAR-FS. For the few-shot SSL task, we randomly select $N$ samples without class-level overlap for each task, and then apply 2-times data augmentation, obtaining a $N$-way 2-shot task with $N$ classes and $2N$ samples. We use the stochastic gradient descent (SGD) optimizer, setting the momentum and weight decay values to $0.9$ and $10^{-4}$ respectively. We evaluate the trained model's performance in some unseen samples sampled from a new class.

**Results.** Table 4 shows the standard few-shot learning results of GeSSL compared with the baselines. From the results, we can see that our framework still achieves remarkable performance improvement, demonstrating the superiority of GeSSL. Specifically, we can observe that: (i) applying GeSSL outperforms the SOTA few-shot learning baselines on almost all the datasets; and (ii) applying GeSSL to the SSL models results in significant performance improvement (an average of nearly 8%) across all tasks. We also conduct experiments on the cross-domain few-shot learning scenario, and the results still proved the outstanding effect of our method. More details are illustrated in Appendix F.3.

## 5.6 ABLATION STUDY AND ANALYSIS

**Influence of $\lambda$.** We evaluate the performance of SimCLR + GeSSL with different $\lambda$ under different $K$ (the number of inner-loop update steps), following the same settings in Section 5.5. The results in Table 5 show that the trade-off performance is optimal when $\lambda = 10$ under $K = 1$ or $K = 5$, which is also the hyperparameter setting for implementation. Meanwhile, the limited performance variation with changes in $K$ suggests its adaptability and ease of adjustment in practical applications.

**Model efficiency.** We evaluate the trade-off performance of multiple baselines using GeSSL on STL-10 Coates et al. (2011) with ResNet-50 backbone. The results in Figure 2 show that GeSSL achieves great performance and efficiency improvements with acceptable parameter size. Combining the full results shown in Appendix G.4, although GeSSL brings a larger memory footprint and parameter size costs, it is relatively negligible compared to the performance and efficiency improvements.

**Role of loss.** We evaluate the impact of the loss functions in the outer-loop optimization. We record the accuracy and training time of SimCLR+GeSSL with different losses, i.e., MSE Tsai et al. (2020), Cross-Entropy De Boer et al. (2005), KL divergence Hershey & Olsen (2007), and Wasserstein distance Panaretos & Zemel (2019). Figure 3 shows that KL divergence is the best choice.

**Implementation of the bi-level optimization.** The gradient update requires composing best-response Jacobians via the chain rule, and the way of differentiation directly affects the model efficiency. Therefore, we analyze the accuracy, training time, and memory footprint of different differentiation methods following Choe et al. (2022); Liu et al. (2018); Zhang et al. (2019). Figure 4 shows that approximate implicit differentiation with finite difference (AID-FD) achieves the optimal results.

**Evaluation of universality.** We quantify the universality of SSL baselines before and after the introduction of GeSSL based on a provable $\sigma-$measurement (See Appendix F.4 for more results and analysis). We choose 5 image-based and 5 video-based tasks following Liu et al. (2022b). Figure 5 shows the comparison results on image-based tasks. We can observe that the existing SSL model has limited universality with higher $\sigma$-measurement error, but is highly improved by introducing GeSSL.

## 6 Related Work

SSL learns representations by transferring knowledge from pretext tasks without annotation. Following Jaiswal et al. (2020); Kang et al. (2023), existing SSL paradigms methods can mainly be divided into two types, i.e., discriminative SSL and generative SSL. The discriminative SSL methods, e.g., SimCLR Chen et al. (2020a), BYOL Grill et al. (2020), and Barlow Twins Zbontar et al. (2021), mainly use stochastic data augmentation to produce two augmented views from the same input sample, and then maximize the similarity of the same pair in the embedding space to learn representations. In contrast, the generative SSL methods, e.g., MAE Hou et al. (2022) and VideoMAE Tong et al. (2022), mainly use an encoding-decoding structure to segment the input samples into multiple blocks, with some blocks being masked and the remaining blocks reassembled in their original positions to learn representation, and then use it to create a new sample. However, despite the empirical effect of the existing SSL methods has been proven, they still face challenges Jaiswal et al. (2020). For example, SSL models generalize poorly (i) when data are scarce Krishnan et al. (2022), or (ii) in real life that have a lot of noise Goyal et al. (2021). SSL models also result in overfitting or underfitting when facing semantic inconsistency or ambiguous data Araslanov & Roth (2021); Li et al. (2020), e.g., the object orientation in rotation prediction is not fixed. Moreover, their performance is affected by the matching between pretext and downstream tasks and may be difficult to transfer well Tendle & Hasan (2021). The experiments in Section 5 and Section F also demonstrate it. Meanwhile, existing theories on universality remain unclear. Previous SSL studies Oord et al. (2018); Hjelm et al. (2018); Mizrahi et al. (2024); Tian et al. (2020b); Oquab et al. (2023) are generally framed as "employing certain methods to obtain a good representation" through experiments, without considering "what constitutes a good representation". In this study, we address this gap by explicitly defining "a good representation" through formalized language, characterizing its core attributes as discriminability, generalizability, and transferability. We also propose corresponding learning objectives to enhance feature interpretability, enabling constraining on the universality of representations or models.

## 7 Conclusion

In this study, we explore the universality of SSL. We first unify SSL paradigms, i.e., discriminative and generative SSL, from the task perspective and propose the definition of SSL universality. It is a fundamental concept that involves discriminability, generalizability, and transferability. Then, we propose GeSSL to explicitly model universality into SSL through bi-level optimization, which introduces a $\sigma$-measurement-based self-motivated target to guide the model learn in the best direction. Extensive theoretical and empirical analyses demonstrate the superior effectiveness of GeSSL.

## REPRODUCIBILITY STATEMENT

This work provides the source code for the algorithm with detailed implementation details which has been submitted as supplementary material. Meanwhile, the appendix of this work also includes clear assumptions and complete proofs for the theoretical analysis and results. For the extensive experiments, detailed descriptions of the data processing steps and experimental setup for each experiment (Table 6 shows the list) are also provided in the appendix.

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

APPENDIX

The appendix is organized into several sections:

- Appendix A encompasses the pseudo-code of our GeSSL's learning process.
- Appendix B contains the analyses and proofs of the presented definitions and theorems.
- Appendix C presents the implementation and architecture of our GeSSL.
- Appendix D provides details for all datasets used in the experiments.
- Appendix E provides details for the baselines mentioned in the main text.
- Appendix F showcases additional experiments, full results, and experimental details of the comparison experiments that were omitted in the main text due to page limitations.
- Appendix G provides the additional experiments and full details of the ablation studies that were omitted in the main text due to page limitations.
- Appendix H illustrates the differences between GeSSL and meta-learning in detail.
- Appendix I illustrates the how GeSSL deal with data issues.

Note that before we illustrate the details and analysis, we provide a brief summary about all the experiments conducted in this paper, as shown in Table 6.

Table 6: Illustration of the experiments conducted in this work. Note that all experimental results are obtained after five rounds of experiments.

| Experiments | Location | Results |
| --- | --- | --- |
| Experiments of unsupervised learning on six benchmark dataset | Section 5.2 and Appendix F.1 | Table 1, Table 8, Table 7, Table 9, and Table 13 |
| Experiments of semi-supervised learning on on ImageNet with two settings | Section 5.3 | Table 2 and Table 14 |
| Experiment of transfer learning on three scenarios | Section 5.4 and Appendix F.2 | Table 3, Table 10, and Table 11 |
| Experiment of few-shot learning on standard and cross-domain scenarios | Section 5.5 and Appendix F.3 | Table 4 and Table 12 |
| Ablation study-Influence of $\lambda$ | Section 5.6 and Appendix G.1 | Table 5 |
| Ablation study-Model efficiency | Section 5.6 and Appendix G.2 | Figure 2 and Table 21 |
| Ablation study-Role of loss | Section 5.6 and Appendix G.3 | Figure 3 |
| Ablation study-Implementation of the bi-level optimization | Section 5.6 and Appendix G.4 | Figure 4 |
| Ablation study-SSL task construction and batchsize | Appendix G.5 | Figure 8 |
| Ablation study-The impact of the update frequency $n$ | Appendix G.5 | Figure 9 |
| Universality of existing SSL methods | Appendix F.4 | Figure 6 and Table 15 |
| Evaluation of generative SSL on three scenarios | Appendix F.5 | Figure 7, Table 16, Table 17, and Table 18 |
| Evaluation on more modalities | Appendix F.6 | Table 19 |

---

**Algorithm 1** Pseudo-Code of the proposed GeSSL

---

**Input:** Candidate pool $\mathcal{D}$; Randomly initialized model $f_\theta$ with a feature extractor $g(\cdot)$, a projection head $h(\cdot)$, and a additional classification head $\pi(\cdot)$
**Parameter:** Mini-batch $N$; The number of update steps $K$; The hyperparameter $\lambda$ in the self-motivated target; Learning rates $\alpha$ and $\beta$
**Output:** The SSL model $f_\theta$ of GeSSL

1: **for** each task **do**
2:     Sample a mini-batch $X_{tr,l}$ from $\mathcal{D}$
3:     Apply random data augmentations to $X_{tr,l}$, obtaining the mini-batch task $X_{tr,l}^{aug}$
4: **end for**
5: **for** $l = 1, ..., M$ **do**
6:     **for** $k = 1, ..., K$ **do**
7:         Update $f_\theta^l$ on the mini-batch task $X_{tr,l}^{aug}$ using Eq.2
8:     **end for**
9:     Obtain $f_\theta^{l,K}$
10:     Obtain the probabilistic distribution $\pi(f_\theta^{l,K}(x))$
11:     **for** $\iota = 1, ..., \lambda$ **do**
12:         Update $f_\theta^l$ on the mini-batch task $X_{tr,l}^{aug}$ using Eq.2
13:     **end for**
14:     Obtain the self-motivated target $f_\theta^{l,K+\lambda}$
15:     Obtain the probabilistic distribution $\pi(f_\theta^{l,K+\lambda}(x))$
16: **end for**
17: Update $f_\theta$ using Eq.4

---

## A    PSEUDO-CODE

The pseudo-code of GeSSL is provided in Algorithm 1.

## B    ANALYSES AND PROOFS

### B.1    DETAILS OF DEFINITION 3.2

**Definition 3.2** ($\sigma$-**measurement**) *Given a set of unseen mini-batch tasks $X_{te}^{aug} = \{X_{te,l}^{aug}\}_{l=1}^{M_{te}}$, assume that the optimal parameter $\theta^*$ is independent of $X_{te}^{aug}$, i.e., not change due to the distribution of test tasks, and the covariance of $\theta^*$ satisfies $\mathrm{Cov}[\theta^*] = (R^2/d)\mathcal{I}_d$, where $R$ is a constant, $d$ is the dimension of the model parameter, and $\mathcal{I}_d$ is a identity matrix, the error rate $\sigma(f_\theta^*)$ is:*

$$\sigma(f_\theta^*) = \sum_{X_{te,l}^{aug} \in X_{te}^{aug}} \sum_{x \in X_{te,l}^{aug}} \mathrm{KL}(\pi(f_\theta^*(x))|\pi(f_{\phi_l}^*(x))), \tag{7}$$

*where $\mathrm{KL}(p|q) = \int p(x) \log\left(\frac{p(x)}{q(x)}\right) dx$ is the calculation of Kullback-Leibler Divergence which is estimated via variational inference, $\pi$ is the auxiliary classification head employed to generating the class probability distribution.*

This definition provides the assumption, i.e., "the optimal parameter $\theta^*$ is independent of $\mathcal{X}^{te}$, i.e., not change due to the distribution of test tasks, and the covariance of $\theta^*$ satisfies $\mathrm{Cov}[\theta^*] = (R^2/d)\mathcal{I}_d$". We will explain these assumptions one by one, including the meaning of the assumptions and their effects:

- $\theta^*$ is independent of $\mathcal{X}^{te}$: Assuming that the optimal parameter $\theta^*$ is not affected by the distribution of test tasks $\mathcal{X}^{te}$, it means that $\theta^*$ contains enough information to cope with various possible test tasks during training. This is a common assumption in machine learning, which is consistent with the training mechanism, i.e., the model approaches the optimal based on the training data. It makes the connection between the model in the training and testing phases clearer and more stable, and the approximation of the training model can be achieved only by relying on the training data.

- $\text{Cov}[\theta^*] = \frac{R^2}{d}\mathcal{I}_d$: This assumption states that the covariance matrix of $\theta^*$ is the product of a scaling factor $\frac{R^2}{d}$ and an identity matrix $\mathcal{I}_d$. This means that the variance of $\theta^*$ in each dimension is equal and different dimensions are independent of each other. The identity matrix form of the covariance matrix in this assumption means that the changes in the model parameters in each dimension are uniform and there is no preference in a specific direction. It ensures that the model can obtain information from different data, eliminates the uneven influence of the parameter dimension $d$ on parameter estimation, and makes the analysis results more universal and robust. This assumption is also a common assumption of machine learning models.

**KL Divergence Calculation**    The KL term $\text{KL}(\pi(f_\theta^*(x)) \,||\, \pi(f_{\phi_l}^*(x)))$ evaluates the difference between the output class probability distribution $\pi(f_\theta^*(x))$ of model $f_\theta^*$ and the output distribution $\pi(f_{\phi_l}^*(x))$ of the task-specific optimal model $f_{\phi_l}^*$. Since Section 6 treats an SSL mini-batch as a multi-class task, "$\pi$ is the auxiliary classification head" that outputs the class probability distribution for a sample. Specifically, $\pi(f_\theta^*(x))$ represents the predicted result of model $f_\theta^*$, i.e., the predicted class vector. $\pi(f_{\phi_l}^*(x))$ represents the output of the task-specific optimal model $f_{\phi_l}^*$, which is assumed to output the ground truth (line 199), i.e., the true one-hot vector of the label. Thus, the KL term calculates the difference between the predicted class vector of model $f_\theta^*$ for sample $x$ and the corresponding true label vector.

**Example** Suppose a specific task $X_{te,l}^{aug}$ contains four original images. After augmentation, we obtain eight samples corresponding to four classes (pseudo-labels), with two samples per class. Suppose sample $x$ belongs to the first class, so its true class probability distribution is $[1, 0, 0, 0]$, which is also the output of $f_{\phi_l}^*$. If $\pi(f_\theta^*(x))$ outputs $[0.81, 0.09, 0.03, 0.07]$, indicating that $x$ is predicted to belong to the first class, the KL term measures the difference between $[0.81, 0.09, 0.03, 0.07]$ and the true label $[1, 0, 0, 0]$, i.e.,

$$D_{KL}(P||Q) = \sum_{i=1}^{4} P(i) \log\left(\frac{P(i)}{Q(i)}\right) = 0.0924.$$

**How $\sigma(f_\theta^*)$ is Calculated in Practice**    First, "$\sigma(f_\theta^*)$ measures the performance gap between the trained SSL model $f_\theta^*$ and the task-specific optimal models," i.e.,

$$\sigma(f_\theta^*) = \sum_{X_{te,l}^{aug} \in X_{te}^{aug}} \sum_{x \in X_{te,l}^{aug}} \text{KL}(\pi(f_\theta^*(x)) \,||\, \pi(f_{\phi_l}^*(x))).$$

Second, the KL term measures the performance of $f_\theta^*$ on each sample. Therefore, in practice, $\sigma(f_\theta^*)$ is calculated by evaluating the KL divergence between the output of model $f_\theta^*$ and the true class probability distribution across all samples ($\sum_{x \in X_{te,l}^{aug}}$) in all training tasks ($\sum_{X_{te,l}^{aug} \in X_{te}^{aug}}$).

B.2    PROOFS OF THEOREM 4.1

**Theorem 4.1** *Let $\tilde{f}_\theta$ and $f_\theta$ be SSL models before and after learning universal knowledge based on Eq.4, and $\text{KL}^f(f_{\theta_1}(X_{tr,l}^{aug}), f_{\theta_2}(X_{tr,l}^{aug}))$ be the abbreviation of $\sum_{x \in X_{tr,l}^{aug}} \text{KL}(\pi(f_{\theta_1}(x))|\pi(f_{\theta_2}(x)))$, the update process for each mini-batch $X_{tr,l}^{aug}$ satisfies:*

$$\tilde{f}_\theta - f_\theta = \frac{\beta}{\alpha}\text{KL}^f(f_\theta^{l,K+\lambda}(X_{tr,l}^{aug}), f_\theta^{l,K} - \alpha\mathcal{G}^\top\text{g}(X_{tr,l}^{aug}))$$

$$-\frac{\beta}{\alpha}\text{KL}^f(f_\theta^{l,K+\lambda}(X_{tr,l}^{aug}), f_\theta^{l,K}(X_{tr,l}^{aug})) + o(\beta(\alpha+\beta)), \tag{8}$$

*where $\mathcal{G}^\top = \mathcal{M}^\top\mathcal{M} \in \mathbb{R}^{n_\theta \times n_\theta}$ with the (transposed) Jacobian $\mathcal{M}$ of $f_\theta^{l,K}$. When the learning rates $\alpha$ and $\beta$ are sufficiently small, there exists a self-motivated target that yields $\tilde{f}_\theta - f_\theta \leq o(\beta(\alpha+\beta))$.*

**Proofs**. To facilitate the proof, we first introduce some useful notations. We let:

$$\text{g} = \nabla_{f_\theta}\ell(f_\theta, X_{tr,l}^{aug})$$

$$\mathcal{G}^\top\text{g} = f_\theta - \alpha\nabla_{f_\theta}\ell(f_\theta, X_{tr,l}^{aug}) \tag{9}$$

$$\text{Q}\mu = \sum_{\mathcal{T}_x^l \in \mathcal{T}_x} \nabla_{f_\theta}\mathcal{L}(f_\theta^l, X_{tr,l}^{aug})$$

Then we get $\tilde{f}_\theta = f_\theta - \beta Q\mu$, by first-order Taylor series expansion of the SSL model with respect to $f_\theta$ around $\tilde{f}_\theta$:

$$
\begin{aligned}
\tilde{f}_\theta &= f_\theta + \beta \left\langle Qg, \tilde{f}_\theta - f_\theta \right\rangle + o(\beta^2 \left\| Q\mu \right\|_2^2) \\
&= f_\theta - \beta \left\langle Qg, Q\mu \right\rangle + o(\beta^2 \left\| Q\mu \right\|_2^2) \\
&= f_\theta - \beta \left\langle \mu, \mathcal{G}^\top g \right\rangle + o(\beta^2 \left\| \mu \right\|_{\mathcal{G}^\top}^2)
\end{aligned}
\tag{10}
$$

then:

$$
\tilde{f}_\theta - f_\theta = -\beta \left\langle \mu, \mathcal{G}^\top g \right\rangle + o(\beta^2 \left\| \mu \right\|_{\mathcal{G}^\top}^2)
\tag{11}
$$

Combining the above formulas with the inner-loop optimization (Eq.2) and outer-loop optimization (Eq.4), we can obtain:

$$
\begin{aligned}
\tilde{f}_\theta - f_\theta &= \beta(\tfrac{1}{\alpha}(\mu(\tilde{f}_\theta, \mathcal{G}^\top g) - \mu(\tilde{f}_\theta, f_\theta^l))) + o(\alpha\beta \left\| Qw \right\|_2^2) + o(\beta^2 \left\| Q\mu \right\|_2^2) \\
&= \tfrac{\beta}{\alpha}(\mu(\tilde{f}_\theta, \mathcal{G}^\top g) - \mu(\tilde{f}_\theta, f_\theta^l)) + o(\alpha\beta \left\| Qw \right\|_2^2 + \beta^2 \left\| Q\mu \right\|_2^2) \\
&= \tfrac{\beta}{\alpha}\mu(\tilde{f}_\theta, \mathcal{G}^\top g) - \tfrac{\beta}{\alpha}\mu(\tilde{f}_\theta, f_\theta^l) + o(\alpha\beta \left\| Qw \right\|_2^2 + \beta^2 \left\| Q\mu \right\|_2^2) \\
&\leq \tfrac{\beta}{\alpha}\mu(\tilde{f}_\theta, \mathcal{G}^\top g) - \tfrac{\beta}{\alpha}\mu(\tilde{f}_\theta, f_\theta^l) + o(\alpha\beta + \beta^2) \\
&= \tfrac{\beta}{\alpha}\mu(\tilde{f}_\theta, \mathcal{G}^\top g) - \tfrac{\beta}{\alpha}\mu(\tilde{f}_\theta, f_\theta^l) + o(\beta(\alpha + \beta))
\end{aligned}
\tag{12}
$$

The first item in this formula measures the distance between the two set of distributions $\pi_{\tilde{f}_\theta}$ (the set of self-motivated targets distributions) and $\pi_{f_\theta}$ (the distribution of $f_\theta$), and the distance measures the learning effect. In our setting, the meta-objective is to minimize the distance between two distributions. Therefore, the first term can be approximately 0. Finally, residuals capture distortions due to same objective of every term in this equation. Then:

$$
\tilde{f}_\theta - f_\theta \leq 0 - \tfrac{\beta}{\alpha}\mu(\tilde{f}_\theta, f_\theta^l) + o(\beta(\alpha + \beta)) = \tfrac{\beta}{\alpha}\mu(\tilde{f}_\theta, f_\theta^l) + o(\beta(\alpha + \beta))
\tag{13}
$$

As $\alpha$ and $\beta$ become small or even zero, the residuals disappear exponentially, where $o(\beta(\alpha + \beta)) \approx 0$. Then when all the above conditions are met, $\tilde{f}_\theta - f_\theta \leq 0$ which means $\tilde{f}_\theta$ achieves performance improvements over previous $f_\theta$. So far, the performance guarantee of self-motivated meta-training is completed.

## C  IMPLEMENTATION DETAILS

**Task Construction.** We build tasks based on images with a batch size of $B = 16$. For data augmentation, we use the same data augmentation scheme as SimCLR to augment each image in the batch 5 times. In simple terms, we draw a random patch ($224 \times 224$) from the original image, and then apply a random augmentation sequence composed of random horizontal flip, cropping, color jitter, etc.

**Architecture and Settings.** We use C4-backbone, ResNet-18, and ResNet-50 backbones as our encoders for a fair comparison with different methods. The convolutional layers are followed by batch normalization, ReLU nonlinearity, and max pooling (strided convolution) respectively. The last layer is fed into a softmax classifier (a classification head). These architectures are pre-trained and kept fixed during training. We optimize our model with a Stochastic Gradient Descent (SGD) optimizer, setting the momentum and weight decay values to $0.9$ and $10^{-4}$ respectively. The specific adjustments of the experimental settings corresponding to different experiments are illustrated in Section 5.2-Subsection 5.5 of the main text. In the ablation experiments, we adopt the experimental settings used in the corresponding dataset, i.e., the experiment of "Influence of $\lambda$" is conducted on miniImageNet, so we adopt the experimental settings described in Section 5.5. All the experiments are apples-to-apples comparisons and performed on NVIDIA RTX 4090 GPUs.

Note that the training of the model is based on a mini-batch task perspective as mentioned in Section 2. Taking a mini-batch $X = (x_i^1, x_i^2)_{i=1}^N$ as an example, it can be regarded as a multi-class classification task. Then, define the sample in the $i$-th augmentation pair as the anchor, and another augmented sample in the same pair as the positive sample, then, we get $(x_i^a, x_i^+)$, where $x_i^+$ is the center of a

class cluster and $x_i^+$ is considered as belong to this class. All other augmented samples in the dataset are treated as negative samples. In other words, given an anchor sample, the entire augmentation dataset is split into a binary classification problem. For each sample, the value of it belonging to class $x_i^a$ as $v_a$, then, the corresponding one-hot vector for it is $[v_a, v_a]$. Then, the whole multi-class classification task is divided into many binary classification problem, and each one corresponds to an anchor sample. This approach effectively models the classifier $\pi(\cdot)$ in SSL settings.

## D  BENCHMARK DATASETS

In this section, we briefly introduce all datasets used in our experiments. In summary, the benchmark datasets can be divided into four categories: (i) for unsupervised learning, we evaluate GeSSL on six benchmark datasets, including CIFAR-10 Krizhevsky et al. (2009), CIFAR-100 Krizhevsky et al. (2009), STL-10 Coates et al. (2011), Tiny ImageNet Le & Yang (2015), ImageNet-100 Tian et al. (2020a) and ImageNet Deng et al. (2009a); (ii) for semi-supervised learning, we evaluate GeSSL on ImageNet Deng et al. (2009a); (iii) for transfer learning, we select two scenarios: instance segmentation (PASCAL VOC Everingham et al. (2010)) and object detection (COCO Lin et al. (2014a)) for analysis; (iv) for few-shot learning, we select three benchmarks for evaluation, including Omniglot Lake et al. (2019), miniImageNet Vinyals et al. (2016a), and CIFAR-FS Bertinetto et al. (2018). The composition of the data set is as follows:

- CIFAR-10 Krizhevsky et al. (2009) is a prevalent image classification benchmark comprising 10 classes, each containing 5000 32×32 resolution images.

- CIFAR-100 Krizhevsky et al. (2009), another widely used image classification benchmark, consists of 100 classes, each containing 5000 images at a resolution of 32×32.

- STL-10 Coates et al. (2011) encompasses 10 classes with 500 training and 800 test images per class at a high resolution of 96x96 pixels. It also includes 100,000 unlabeled images for unsupervised learning.

- Tiny ImageNet Le & Yang (2015), a subset of ImageNet by Stanford University, comprises 200 classes, each with 500 training, 50 verification, and 50 test images.

- ImageNet-100 Tian et al. (2020a), a subset of ImageNet, includes 100 classes, each containing 1000 images.

- ImageNet Deng et al. (2009a), organized by the WordNet hierarchy, is a renowned dataset featuring 1.3 million training and 50,000 test images across 1000+ classes.

- PASCAL VOC dataset Everingham et al. (2010), known for object classification, detection, and segmentation, encompasses 20 classes with a total of 11,530 images split between VOC 07 and VOC 12.

- COCO dataset Lin et al. (2014a), primarily used for object detection and segmentation, comprises 91 classes, 328,000 samples, and 2,500,000 labels.

- miniImageNet Vinyals et al. (2016a) is a few-shot learning dataset that consists of 100 classes, each with 600 images. The images have a resolution of 84x84 pixels.

- Omniglot Lake et al. (2019) is another dataset for few-shot learning, which comprises 1623 different handwritten characters from 50 different alphabets. The 1623 characters were drawn by 20 different people online using Amazon's Mechanical Turk. Each image is paired with stroke data $[x, y, t]$ sequences and time (t) coordinates (ms).

- CIFAR-FS Bertinetto et al. (2018) is also a dataset for few-shot learning research, derived from the CIFAR-100 dataset. It consists of 100 classes, each with a small training set of 500 images and a test set of 100 images. The images have a resolution of $32 \times 32$ pixels.

In addition, we further construct cross-domain few-shot learning experiments in Appendix F.3 and introduced six benchmark data sets, including:

- CUB Welinder et al. (2010) is a dataset of 200 bird species, with 11,788 images in total and about 60 images per species. Each image has detailed annotations, including subcategory labels, 15 part locations, 312 binary attributes, and a bounding box.

- Cars Krause et al. (2013) is a dataset of 196 car models, with 16,185 images in total and about 80 images per model. Each image has a subcategory label, indicating the manufacturer, model, and year of the car.

- Places Zhou et al. (2017) is a dataset of 205 scene categories, with 2.5 million images in total and about 12,000 images per category. The scene categories are defined by their functions, representing the entry-level of the environment.

- CropDiseases Mohanty et al. (2016) is a dataset of 24,881 images of crop pests and diseases, with 22 categories, each including different pests and diseases of 4 crops (cashew, cassava, maize, and tomato).

- ISIC Codella et al. (2018) is a dataset of over 13,000 dermoscopic images of skin lesions, which is the largest publicly available quality-controlled archive of dermoscopic images. The dataset includes 8 common types of skin lesions, such as melanoma, basal cell carcinoma, squamous cell carcinoma, etc.

- ChestX Wang et al. (2017) is a dataset of 112,120 chest X-ray images, with 14 common types of chest diseases, such as pneumonia, emphysema, fibrosis, etc. The dataset was collected from 30,805 unique patients (from 1992 to 2015) of the National Institutes of Health Clinical Center (NIHCC).

## E    BASELINES

In this section, we briefly introduce all baselines used in the experiments for comparison. We select fifteen representative self-supervised methods as baselines. These methods cover almost all the classic and SOTA self-supervised methods, including:

- SimCLR Chen et al. (2020a) learns visual representations by contrastive learning of augmented image pairs. It uses a neural network to maximize the similarity of positive pairs and minimize the similarity of negative pairs.

- MoCo v2 Chen et al. (2020b) improves MoCo Chen et al. (2020b), another contrastive learning method for visual representation learning. MoCo v2 introduces a momentum encoder, a memory bank, and a shuffling BN layer to handle limited batch size and noisy negatives. MoCo v2 also adopts SimCLR's data augmentation and loss function to boost the performance.

- BYOL Grill et al. (2020) does not need negative pairs or a large batch size. It uses two neural networks, an online network and a target network, that learn from each other. The online network predicts the target network's representation of an augmented image, while the target network is updated by a slow-moving average of the online network.

- SimSiam Chen & He (2021) simplifies BYOL by removing the momentum encoder and the prediction MLP. It consists of two Siamese networks that map an input image to a feature vector, and a small MLP head that projects the feature vector to the contrastive learning space. SimSiam applies a stop-gradient operation to one of the MLP outputs, and uses a negative cosine similarity loss to maximize the similarity between the two outputs.

- Barlow Twins Zbontar et al. (2021) learns representations by enforcing that the cross-correlation matrix between the outputs of two identical networks fed with different augmentations of the same image is close to the identity matrix. This encourages the networks to produce similar representations for the positive pair, while reducing the redundancy between the representation dimensions.

- DeepCluster Caron et al. (2018) is a clustering-based method for self-supervised learning. It iteratively groups the features produced by a convolutional network into clusters, and uses the cluster assignments as pseudo-labels to update the network parameters by supervised learning. DeepCluster can discover meaningful clusters that are discriminative and invariant to transformations, and can learn competitive features for various downstream tasks.

- SwAV Caron et al. (2020) uses online swapping of cluster assignments between multiple views of the same image to learn visual features. SwAV first computes prototypes (cluster centers) from a large set of features, and then assigns each feature to the nearest prototype.

The assignments are then swapped across the views, and the network is trained to predict the swapped assignments.

- DINO Caron et al. (2021) learns visual features by using a teacher-student architecture and a distillation loss. The teacher network is an exponential moving average of the student network, and the distillation loss makes the student features similar to the teacher features. DINO also applies a centering and sharpening operation to the teacher features, which prevents feature collapse and increases feature diversity.

- W-MSE Ermolov et al. (2021) learns features by using a weighted mean squared error (MSE) loss, which assigns higher weights to the informative and less noisy features, and lower weights to the less informative and more noisy features.

- RELIC v2 Tomasev et al. (2022) learns visual features by predicting relative location of image patches. RELIC v2 divides an image into a grid of patches, and randomly selects a query and a target patch. The network is trained to predict the relative location of the target patch with respect to the query patch, using a cross-entropy loss.

- LMCL Chen et al. (2021) learns visual features by using a large margin cosine loss (LMCL). LMCL is a metric learning loss that makes the features of the same class closer and the features of different classes farther in the cosine space.

- ReSSL Zheng et al. (2021) learns visual features by using a reconstruction loss and a contrastive loss. ReSSL applies random cropping and resizing to generate two views of the same image, and then feeds them to a reconstruction network and a contrastive network. The reconstruction network is trained to reconstruct the original image from the cropped view, while the contrastive network is trained to maximize the similarity between the features of the two views.

- SSL-HSIC Li et al. (2021) learns visual features by using a Hilbert-Schmidt independence criterion (HSIC) loss. HSIC is a measure of statistical dependence between two random variables, and can be used to align the features of different views of the same image.

- CorInfoMax Ozsoy et al. (2022) learns visual features by maximizing the correlation and mutual information between the features of augmented image pairs and the image labels. CorInfoMax aims to learn features that are both discriminative and consistent, and outperform previous methods on image classification and segmentation tasks.

- MEC Liu et al. (2022a) is a clustering algorithm that can handle large-scale data with limited memory by using a memory-efficient clustering (MEC) loss. MEC first samples a subset of features, and then performs k-means clustering on the subset. The cluster assignments are then propagated to the rest of the features by a nearest neighbor search.

- VICRegL Bardes et al. (2022) learns visual features by using a variance-invariance-covariance regularization loss (VICRegL).

In addition, for the few-shot learning scenario, we choose six advanced unsupervised few-shot learning methods as comparison baselines.

- CACTUs Hsu et al. (2018) uses clustering and augmentation to create pseudo-labels for unlabeled data. It then trains a classifier on the labeled data and fine-tunes it on a few labeled examples from the target task.

- UMTRA Khodadadeh et al. (2019) uses random selection and augmentation to create tasks with pseudo-labels from unlabeled data. It then trains a classifier on each task and adapts it to the target task using a few labeled examples.

- LASIUM Khodadadeh et al. (2020) uses latent space interpolation to generate tasks with pseudo-labels from a generative model. It then trains an energy-based model on each task and adapts it to the target task using a few labeled examples.

- SVEBM Kong et al. (2021) uses a symbol-vector coupling energy-based model to learn from unlabeled data. It then adapts the model to the target task using a diffusion process.

- GMVAE Lee et al. (2021) uses a Gaussian mixture variational autoencoder to perform learning, and then adapts the model to the target task using a variational inference process.

Table 7: The classification accuracies ($\pm$ 95% confidence interval) of a linear classifier (linear) and a 5-nearest neighbors classifier (5-nn) with a ResNet-18 as the feature extractor. The comparison baselines cover almost all types of methods mentioned in Section 6. The "-" denotes that the results are not reported. More details of the baselines are provided in Appendix E.

| Method | CIFAR-10 | | CIFAR-100 | | STL-10 | | Tiny ImageNet | |
|---|---|---|---|---|---|---|---|---|
| | linear | 5 $-$ nn | linear | 5 $-$ nn | linear | 5 $-$ nn | linear | 5 $-$ nn |
| SimCLR Chen et al. (2020a) | $91.80 \pm 0.15$ | $88.42 \pm 0.15$ | $66.83 \pm 0.27$ | $56.56 \pm 0.18$ | $90.51 \pm 0.14$ | $85.68 \pm 0.10$ | $48.84 \pm 0.15$ | $32.86 \pm 0.25$ |
| MoCo Chen et al. (2020b) | $91.69 \pm 0.12$ | $88.66 \pm 0.14$ | $67.02 \pm 0.16$ | $56.29 \pm 0.25$ | $90.64 \pm 0.28$ | $88.01 \pm 0.19$ | $50.92 \pm 0.22$ | $35.55 \pm 0.16$ |
| BYOL Grill et al. (2020) | $91.93 \pm 0.22$ | $89.45 \pm 0.22$ | $66.60 \pm 0.16$ | $56.82 \pm 0.17$ | $91.99 \pm 0.13$ | $88.64 \pm 0.20$ | $51.00 \pm 0.12$ | $36.24 \pm 0.28$ |
| SimSiam Chen & He (2021) | $91.71 \pm 0.27$ | $88.65 \pm 0.17$ | $67.22 \pm 0.26$ | $56.36 \pm 0.19$ | $91.01 \pm 0.19$ | $88.16 \pm 0.19$ | $51.14 \pm 0.20$ | $35.67 \pm 0.16$ |
| Barlow Twins Zbontar et al. (2021) | $90.88 \pm 0.19$ | $89.68 \pm 0.21$ | $66.13 \pm 0.10$ | $56.70 \pm 0.25$ | $90.38 \pm 0.13$ | $87.13 \pm 0.23$ | $49.78 \pm 0.26$ | $34.18 \pm 0.18$ |
| SwAV Caron et al. (2020) | $91.03 \pm 0.19$ | $89.52 \pm 0.24$ | $66.56 \pm 0.17$ | $57.01 \pm 0.25$ | $90.72 \pm 0.29$ | $86.24 \pm 0.26$ | $52.02 \pm 0.26$ | $37.40 \pm 0.11$ |
| DINO Caron et al. (2021) | $91.83 \pm 0.25$ | $90.15 \pm 0.33$ | $67.15 \pm 0.21$ | $56.48 \pm 0.19$ | $91.03 \pm 0.12$ | $86.15 \pm 0.25$ | $51.13 \pm 0.30$ | $37.86 \pm 0.19$ |
| W-MSE Ermolov et al. (2021) | $91.99 \pm 0.12$ | $89.87 \pm 0.25$ | $67.64 \pm 0.16$ | $56.45 \pm 0.26$ | $91.75 \pm 0.23$ | $88.59 \pm 0.15$ | $49.22 \pm 0.16$ | $35.44 \pm 0.10$ |
| RELIC v2 Tomasev et al. (2022) | $91.92 \pm 0.14$ | $90.02 \pm 0.22$ | $67.66 \pm 0.20$ | $57.03 \pm 0.18$ | $91.10 \pm 0.23$ | $88.66 \pm 0.12$ | $49.33 \pm 0.13$ | $35.52 \pm 0.22$ |
| LMCL Chen et al. (2021) | $91.91 \pm 0.25$ | $88.52 \pm 0.29$ | $67.01 \pm 0.18$ | $56.86 \pm 0.14$ | $90.87 \pm 0.18$ | $85.91 \pm 0.25$ | $49.24 \pm 0.18$ | $32.88 \pm 0.13$ |
| ReSSL Zheng et al. (2021) | $90.20 \pm 0.16$ | $88.26 \pm 0.18$ | $66.79 \pm 0.12$ | $53.72 \pm 0.28$ | $88.25 \pm 0.14$ | $86.33 \pm 0.17$ | $46.60 \pm 0.18$ | $32.39 \pm 0.20$ |
| SSL-HSIC Li et al. (2021) | $91.95 \pm 0.14$ | $89.99 \pm 0.17$ | $67.23 \pm 0.26$ | $57.01 \pm 0.27$ | $92.09 \pm 0.20$ | $88.91 \pm 0.29$ | $51.37 \pm 0.15$ | $36.03 \pm 0.12$ |
| CorInfoMax Ozsoy et al. (2022) | $91.81 \pm 0.11$ | $89.85 \pm 0.13$ | $67.09 \pm 0.24$ | $56.92 \pm 0.23$ | $91.85 \pm 0.25$ | $89.99 \pm 0.24$ | $51.23 \pm 0.14$ | $35.98 \pm 0.09$ |
| MEC Liu et al. (2022a) | $90.55 \pm 0.22$ | $87.80 \pm 0.10$ | $67.36 \pm 0.27$ | $57.25 \pm 0.25$ | $91.33 \pm 0.14$ | $89.03 \pm 0.33$ | $50.93 \pm 0.13$ | $36.28 \pm 0.14$ |
| VICRegL Bardes et al. (2022) | $90.99 \pm 0.13$ | $88.75 \pm 0.26$ | $68.03 \pm 0.32$ | $57.34 \pm 0.29$ | $92.12 \pm 0.26$ | $90.01 \pm 0.20$ | $51.52 \pm 0.13$ | $36.24 \pm 0.16$ |
| SimCLR + GeSSL | $93.15 \pm 0.25$ | $91.02 \pm 0.16$ | $69.23 \pm 0.20$ | $58.56 \pm 0.18$ | $93.15 \pm 0.28$ | $\mathbf{91.55 \pm 0.17}$ | $53.54 \pm 0.21$ | $37.16 \pm 0.27$ |
| MoCo + GeSSL | $92.78 \pm 0.19$ | $89.15 \pm 0.22$ | $68.16 \pm 0.14$ | $59.22 \pm 0.24$ | $93.17 \pm 0.18$ | $88.96 \pm 0.30$ | $52.07 \pm 0.15$ | $37.22 \pm 0.13$ |
| BYOL + GeSSL | $\mathbf{93.85 \pm 0.22}$ | $\mathbf{92.44 \pm 0.30}$ | $69.15 \pm 0.22$ | $58.99 \pm 0.16$ | $94.45 \pm 0.18$ | $90.50 \pm 0.17$ | $\mathbf{54.84 \pm 0.19}$ | $37.54 \pm 0.26$ |
| Barlow Twins + GeSSL | $92.99 \pm 0.18$ | $91.02 \pm 0.17$ | $69.56 \pm 0.19$ | $59.93 \pm 0.17$ | $93.84 \pm 0.09$ | $89.46 \pm 0.25$ | $52.65 \pm 0.14$ | $35.15 \pm 0.16$ |
| SwAV + GeSSL | $93.17 \pm 0.20$ | $89.98 \pm 0.26$ | $69.98 \pm 0.24$ | $59.36 \pm 0.25$ | $92.85 \pm 0.29$ | $91.68 \pm 0.24$ | $51.89 \pm 0.24$ | $36.78 \pm 0.34$ |
| DINO + GeSSL | $92.77 \pm 0.23$ | $92.12 \pm 0.23$ | $\mathbf{70.85 \pm 0.18}$ | $\mathbf{61.68 \pm 0.33}$ | $\mathbf{94.48 \pm 0.29}$ | $91.48 \pm 0.19$ | $53.51 \pm 0.26$ | $\mathbf{37.89 \pm 0.24}$ |

- PsCo Jang et al. (2023) uses a probabilistic subspace clustering model to learn from unlabeled data. It then adapts the model to the target task using a few labeled examples and a subspace alignment process.

## F    ADDITIONAL EXPERIMENTS

In this section, we introduce the additional experiments, full results, and experimental details of the comparison experiments, including unsupervised learning (Appendix F.1, also Section 5.2 of the main text), transfer learning (Appendix F.2, also Section 5.4 of the main text), and few-shot learning (Appendix F.3, also Section 5.5 of the main text). Next, we conduct experiments based on the proposed $\sigma$-measurement (Definition 3.2) to evaluate the universality of existing SSL methods in Appendix F.4. Finally, we apply our method to the generative self-supervised learning task and other modalities, e.g., text, to further evaluate the effectiveness of GeSSL in Appendix F.5 and F.6.

### F.1    UNSUPERVISED LEARNING

In this section, we present additional results of the unsupervised learning experiments. Specifically, Table 7 shows the results on four small-scale datasets. We can observe that applying the proposed GeSSL framework significantly outperforms the state-of-the-art (SOTA) methods on all four datasets. Table 7 shows the results on four small-scale datasets. Table 8 provides the full comparison results of our proposed GeSSL on the medium-scale dataset, i.e., ImageNet-100. The results still demonstrate the proposed GeSSL's ability to enhance the performance of self-supervised learning methods, achieving significant improvements over the original models on all baselines. Moreover, applying our GeSSL framework to all four types of representative SSL models as described in Section 6, including SimCLR, MoCo, BYOL, Barlow Twins, SwAV, and DINO, achieves an average improvement of 3% compared to the original frameworks. Table

Table 8: The Top-1 and Top-5 classification accuracies of linear classifier on ImageNet-100 with ResNet-50 as feature extractor.

| Method | Top-1 | Top-5 |
|---|---|---|
| SimCLR Chen et al. (2020a) | $70.15 \pm 0.16$ | $89.75 \pm 0.14$ |
| MoCo Chen et al. (2020b) | $72.80 \pm 0.12$ | $91.64 \pm 0.11$ |
| BYOL Grill et al. (2020) | $71.48 \pm 0.15$ | $92.32 \pm 0.14$ |
| SimSiam Chen & He (2021) | $73.01 \pm 0.21$ | $92.61 \pm 0.27$ |
| Barlow Twins Zbontar et al. (2021) | $75.97 \pm 0.23$ | $92.91 \pm 0.19$ |
| SwAV Caron et al. (2020) | $75.78 \pm 0.16$ | $92.86 \pm 0.15$ |
| DINO Caron et al. (2021) | $75.43 \pm 0.18$ | $93.32 \pm 0.19$ |
| W-MSE Ermolov et al. (2021) | $76.01 \pm 0.27$ | $93.12 \pm 0.21$ |
| RELIC v2 Tomasev et al. (2022) | $75.88 \pm 0.15$ | $93.52 \pm 0.13$ |
| LMCL Chen et al. (2021) | $75.89 \pm 0.19$ | $92.89 \pm 0.28$ |
| ReSSL Zheng et al. (2021) | $75.77 \pm 0.21$ | $92.91 \pm 0.27$ |
| SSL-HSIC Li et al. (2021) | $74.99 \pm 0.19$ | $93.01 \pm 0.20$ |
| CorInfoMax Ozsoy et al. (2022) | $75.54 \pm 0.20$ | $92.23 \pm 0.25$ |
| MEC Liu et al. (2022a) | $75.38 \pm 0.17$ | $92.84 \pm 0.20$ |
| VICRegL Bardes et al. (2022) | $75.96 \pm 0.19$ | $92.97 \pm 0.26$ |
| SimCLR + GeSSL | $72.43 \pm 0.18$ | $91.87 \pm 0.21$ |
| MoCo + GeSSL | $73.78 \pm 0.19$ | $93.28 \pm 0.23$ |
| SimSiam + GeSSL | $75.48 \pm 0.19$ | $94.83 \pm 0.31$ |
| Barlow Twins + GeSSL | $76.83 \pm 0.19$ | $93.23 \pm 0.18$ |
| SwAV + GeSSL | $76.38 \pm 0.20$ | $\mathbf{95.47 \pm 0.19}$ |
| DINO + GeSSL | $76.84 \pm 0.25$ | $94.98 \pm 0.24$ |
| LMCL + GeSSL | $77.38 \pm 0.21$ | $95.10 \pm 0.25$ |
| ReSSL + GeSSL | $76.98 \pm 0.23$ | $94.88 \pm 0.24$ |
| VICRegL + GeSSL | $\mathbf{77.58 \pm 0.22}$ | $95.46 \pm 0.15$ |

9 provides the comparison results of our proposed GeSSL on a large-scale dataset, i.e., ImageNet. The results show that, (i) the self-supervised learning model applying GeSSL achieves the state-of-the-art result (SOTA) performance under all epoch conditions; and (ii) after applying the proposed GeSSL, the self-supervised learning models consistently outperforms the original frameworks in terms of average classification accuracy at 100, 200 and 400 epochs. For 1000 epochs, VICRegL + GeSSL yields the best result among other state-of-the-art methods, with an average accuracy of 78.72%.

**More recent methods** The effect of GeSSL is reflected in the performance improvement when applying it to the SSL baselines. The experimental results above have demonstrated that after the introduction of GeSSL, the effects of all SSL baselines have been significantly improved. These results have shown the outstanding effectiveness and robustness of GeSSL. The SSL baselines we use cover all SOTA methods on the leaderboard of the adopted benchmark datasets (before submission). The methods proposed in 2023-24 mainly are variants of the currently used comparison baselines.

To evaluate the effect of GeSSL on recently proposed methods, we select the two SSL methods published in ICML23 for testing Baevski et al. (2023); Joshi & Mirzasoleiman (2023), where we follow the same experimental settings. The results are shown in Tables 13 and 14. The results still prove the effectiveness of GeSSL. We will supplement these results in the final version.

F.2    TRANSFER LEARNING

As mentioned in Section 5.4, we construct three sets of transfer learning experiments, including the most commonly used object detection and instance segmentation protocol Chen et al. (2020a); Zbontar et al. (2021); Grill et al. (2020), transfer to other domains (different datasets), and transfer learning on video-based tasks. The results of the first experiment are illustrated in Section 5.4, and the other two sets of experiments are described below.

**Transfer to other domains.** To explore the nature of transfer learning of the proposed framework, we leverage models that had been pre-trained on the CIFAR100 dataset, including SimCLR Chen et al. (2020a), BYOL Grill et al. (2020), and Barlow Twins Zbontar et al. (2021), on the CIFAR100 dataset. We then applied these models to four distinct datasets, including CIFAR10 Krizhevsky et al. (2009), Flower102 Nilsback & Zisserman (2008), Food101 Bossard et al. (2014), and Aircraft Maji et al. (2013). We first calculate the classification performance (Top-1) based on the existing self-supervised model on different data sets, recorded as $acc(\text{method}, \text{dataset})$, such as $acc(\text{SimCLR}, \text{Flower102})$. Then, we calculate the model's classification performance by incorporating GeSSL on those data sets, which is recorded as $acc(\text{method} + \text{GeSSL}, \text{dataset})$. Finally, we get the improvement $\Delta(\text{method}, \text{dataset}) = acc(\text{method} + \text{GeSSL}, \text{dataset}) - acc(\text{method}, \text{dataset})$ in classification performance on each dataset, as shown in Table 10. The results show that the migration effect of the model after applying the GeSSL framework has been steadily improved, proving that GeSSL has effectively improved the versatility of the SSL model.

**Video-based Task** In order to assess the performance of our method with video-based tasks, we transition our pre-trained model to handle a variety of video tasks, utilizing the UniTrack evaluation framework Wang et al. (2021) as our testing ground. The findings are compiled in Table 11, which includes results from five distinct tasks, drawing on the features from [layer3/layer4] of the Resnet-50. The data indicates that existing SSL methods incorporating our GeSSL significantly surpass original SSL approaches, with SimCLR achieving more than a 2% improvement in VOS Perazzi et al. (2016), and BYOL seeing over a 3% gain in MOT Milan et al. (2016).

F.3    FEW-SHOT LEARNING

The outstanding performance of GeSSL in the few-shot learning scenario has been confirmed in Section 5.5, where it can produce good results with limited data. However, the situation becomes complicated in scenarios where data collection is infeasible in real life, such as medical diagnosis and satellite imagery (Zheng, 2015; Tang et al., 2012). Therefore, the performance of the model on cross-domain few-shot learning tasks is crucial, as it determines the applicability of the learning model (Guo et al., 2020). To ensure that GeSSL can achieve robust performance in real-world applications, we further conduct comparative experiments on cross-domain few-shot learning.

Table 9: The Top-1 and Top-5 classification accuracies of linear classification on the ImageNet dataset with ResNet-50 as the feature extractor. We record the comparison results from 100, 200, 400, and 1000 epochs.

| Method | 100 Epochs | | 200 Epochs | | 400 Epochs | 1000 Epochs |
|---|---|---|---|---|---|---|
| | Top-1 | Top-5 | Top-1 | Top-5 | Top-1 | Top-1 |
| Supervised | 71.93 | - | 73.45 | - | 74.92 | 76.35 |
| SimCLR Chen et al. (2020a) | $66.54 \pm 0.22$ | $88.14 \pm 0.26$ | $68.32 \pm 0.31$ | $89.76 \pm 0.23$ | $69.24 \pm 0.21$ | $70.45 \pm 0.30$ |
| MoCo Chen et al. (2020b) | $64.53 \pm 0.25$ | $86.17 \pm 0.11$ | $67.55 \pm 0.27$ | $88.42 \pm 0.11$ | $69.76 \pm 0.14$ | $71.16 \pm 0.23$ |
| BYOL Grill et al. (2020) | $67.65 \pm 0.27$ | $88.95 \pm 0.11$ | $69.94 \pm 0.21$ | $89.45 \pm 0.27$ | $71.85 \pm 0.12$ | $73.35 \pm 0.27$ |
| SimSiam Chen & He (2021) | $68.14 \pm 0.26$ | $87.12 \pm 0.26$ | $70.02 \pm 0.14$ | $88.76 \pm 0.23$ | $70.86 \pm 0.34$ | $71.37 \pm 0.22$ |
| Barlow Twins Zbontar et al. (2021) | $67.24 \pm 0.22$ | $88.66 \pm 0.19$ | $69.94 \pm 0.32$ | $88.97 \pm 0.27$ | $70.22 \pm 0.15$ | $73.29 \pm 0.13$ |
| SwAV Caron et al. (2020) | $66.55 \pm 0.27$ | $88.42 \pm 0.22$ | $69.12 \pm 0.24$ | $89.38 \pm 0.20$ | $70.78 \pm 0.34$ | $75.32 \pm 0.11$ |
| DINO Caron et al. (2021) | $67.23 \pm 0.19$ | $88.48 \pm 0.21$ | $70.58 \pm 0.24$ | $91.32 \pm 0.27$ | $71.98 \pm 0.26$ | $73.94 \pm 0.29$ |
| W-MSE Ermolov et al. (2021) | $67.48 \pm 0.29$ | $90.39 \pm 0.27$ | $70.85 \pm 0.31$ | $91.57 \pm 0.20$ | $72.49 \pm 0.24$ | $72.84 \pm 0.18$ |
| RELIC v2 Tomasev et al. (2022) | $66.38 \pm 0.23$ | $90.89 \pm 0.21$ | $70.98 \pm 0.21$ | $91.15 \pm 0.26$ | $71.84 \pm 0.21$ | $72.17 \pm 0.20$ |
| LMCL Chen et al. (2021) | $66.75 \pm 0.13$ | $89.85 \pm 0.36$ | $70.83 \pm 0.26$ | $90.04 \pm 0.21$ | $72.53 \pm 0.24$ | $72.97 \pm 0.29$ |
| ReSSL Zheng et al. (2021) | $67.41 \pm 0.27$ | $90.55 \pm 0.23$ | $69.92 \pm 0.24$ | $91.25 \pm 0.12$ | $72.46 \pm 0.29$ | $72.91 \pm 0.30$ |
| CorInfoMax Ozsoy et al. (2022) | $70.13 \pm 0.12$ | $91.14 \pm 0.25$ | $70.83 \pm 0.15$ | $91.53 \pm 0.22$ | $73.28 \pm 0.24$ | $74.87 \pm 0.36$ |
| MEC Liu et al. (2022a) | $69.91 \pm 0.10$ | $90.67 \pm 0.15$ | $70.34 \pm 0.27$ | $91.25 \pm 0.38$ | $72.91 \pm 0.27$ | $75.07 \pm 0.24$ |
| VICRegL Bardes et al. (2022) | $69.99 \pm 0.25$ | $91.27 \pm 0.16$ | $70.24 \pm 0.27$ | $91.60 \pm 0.24$ | $72.14 \pm 0.20$ | $75.07 \pm 0.23$ |
| SimCLR + GeSSL | $68.38 \pm 0.18$ | $89.74 \pm 0.22$ | $69.65 \pm 0.16$ | $90.98 \pm 0.19$ | $71.30 \pm 0.19$ | $72.48 \pm 0.29$ |
| MoCo + GeSSL | $66.54 \pm 0.22$ | $88.19 \pm 0.23$ | $69.47 \pm 0.28$ | $90.34 \pm 0.28$ | $70.48 \pm 0.30$ | $72.81 \pm 0.21$ |
| SimSiam + GeSSL | $70.48 \pm 0.19$ | $88.34 \pm 0.17$ | $71.74 \pm 0.19$ | $89.28 \pm 0.30$ | $72.58 \pm 0.18$ | $74.55 \pm 0.25$ |
| Barlow Twins + GeSSL | $69.39 \pm 0.20$ | $89.40 \pm 0.21$ | $71.89 \pm 0.22$ | $90.32 \pm 0.14$ | $73.90 \pm 0.19$ | $74.91 \pm 0.23$ |
| SwAV + GeSSL | $68.93 \pm 0.19$ | $89.39 \pm 0.16$ | $71.47 \pm 0.10$ | $90.28 \pm 0.28$ | $72.48 \pm 0.19$ | $76.15 \pm 0.18$ |
| DINO + GeSSL | $69.39 \pm 0.19$ | $90.49 \pm 0.21$ | $72.84 \pm 0.19$ | $\mathbf{93.54 \pm 0.18}$ | $73.84 \pm 0.28$ | $76.15 \pm 0.20$ |
| VICRegL + GeSSL | $\mathbf{72.38 \pm 0.23}$ | $\mathbf{91.23 \pm 0.19}$ | $73.54 \pm 0.29$ | $93.17 \pm 0.30$ | $\mathbf{74.15 \pm 0.25}$ | $\mathbf{78.72 \pm 0.29}$ |

Table 10: The performance of adding task information in self-supervised models on different datasets.

| Evl.dataset | SimCLR+GeSSL | BYOL+GeSSL | Barlow Twins+GeSSL | VICRegL+GeSSL |
|---|---|---|---|---|
| CIFAR10 | +3.51 | +2.49 | +2.12 | +2.77 |
| Flower102 | +3.99 | +2.05 | +2.96 | +3.01 |
| Food101 | +1.81 | +2.35 | +1.96 | +1.99 |
| Aircraft | +2.55 | +2.86 | +2.19 | +2.30 |

**Experimental setup.** We compare our proposed GeSSL with the few-shot learning baselines as described in Table 4 (Subsection 5.5) on cross-domain few-shot learning. The details of the baselines are illustrated in Appendix E. We adopt six cross-domain few-shot learning benchmark datasets, and divided these datasets into two categories according to their similarity with ImageNet: i) high similarity: CUB Welinder et al. (2010), Cars Krause et al. (2013), and Places Zhou et al. (2017); ii) low similarity: CropDiseases Mohanty et al. (2016), ISIC Codella et al. (2018), and ChestX Wang et al. (2017). The $(N, A)$ in the tables means the $N$-way $A$-shot tasks with $N$ classes and $N \times A$ samples, where each class has $A$ samples augmented from the same image.

**Results**. Table 12 presents the performance of the model trained on miniImageNet and transfer to the six cross-domain few-shot learning benchmark datasets mentioned above. By observation, we further validate the performance of our proposed GeSSL: i) Effectiveness: achieves better results than the state-of-the-art baselines on almost all benchmark datasets; ii) Generalization: achieves nearly a 3% improvement compared to unsupervised few-shot Learning and self-supervised learning on the datasets with significant differences from the training phase; iii) Robustness: achieves better results than the PsCo Jang et al. (2023) which introduces out-of-distribution samples, even though we do not explicitly consider out-of-distribution samples on datasets with significant differences.

## F.4 UNIVERSALITY OF EXISTING SSL METHODS

Current self-supervised learning (SSL) models overlook the explicit incorporation of universality within their objectives, and the corresponding theoretical comprehension remains inadequate, posing challenges for SSL models to attain universality in practical, real-world applications Huang et al. (2021); Sun et al. (2020); Ericsson et al. (2022). Therefore, we propose a provable $\sigma$−measure

Table 11: Transfer learning on video tracking tasks. All methods use the same ResNet-50 backbone and are evaluated based on UniTrack.

| Method | SOT | | VOS | MOT | | MOTS | | PoseTrack |
|---|---|---|---|---|---|---|---|---|
| | $AUC_{XCorr}$ | $AUC_{DCF}$ | $\mathcal{J}$-mean | IDF1 | HOTA | IDF1 | HOTA | IDF1 |
| SimCLR | 47.3 / 51.9 | 61.3 / 50.7 | 60.5 / 56.5 | 66.9 / 75.6 | 57.7 / 63.2 | 65.8 / 67.6 | 67.7 / 69.5 | 72.3 / 73.5 |
| MoCo | 50.9 / 47.9 | 62.2 / 53.7 | 61.5 / 57.9 | 69.2 / 74.1 | 59.4 / 61.9 | 70.6 / 69.3 | 71.6 / 70.9 | 72.8 / 73.9 |
| SwAV | 49.2 / 52.4 | 61.5 / 59.4 | 59.4 / 57.0 | 65.6 / 74.4 | 56.9 / 62.3 | 68.8 / 67.0 | 69.9 /69.5 | 72.7 / 73.6 |
| BYOL | 48.3 / 55.5 | 58.9 / 56.8 | 58.8 / 54.3 | 65.3 / 74.9 | 56.8 / 62.9 | 70.1 / 66.8 | 70.8 / 69.3 | 72.4 / 73.8 |
| Barlow Twins | 44.5 / 55.5 | 60.5 / **60.1** | 61.7 / 57.8 | 63.7 / 74.5 | 55.4 / 62.4 | 68.7 / 67.4 | 69.5 / 69.8 | 72.3 / 74.3 |
| SimCLR+GeSSL | 50.3 / 54.0 | **63.1** / 53.7 | **62.6 / 58.5** | **69.7 / 77.7** | **60.** / **65.2** | 67.8 / **69.9** | 69.0 / 71.3 | 73.4 / 74.5 |
| BYOL+GeSSL | **51.5 / 57.4** | 60.3 / 58.9 | 60.7 / 57.0 | 67.4 / 76.9 | 57.9 / 64.2 | **72.5** / 68.3 | **73.2 / 71.3** | **74.7 / 75.3** |

Table 12: The cross-domain few-shot learning accuracies ($\pm95\%$ confidence interval). We transfer models trained on miniImageNet to six benchmark datasets with the C4-backbone. The best results are highlighted in **bold**. The $(N, A)$ means the $N$-way $A$-shot tasks with $N$ classes and $N \times A$ samples, where each class has $A$ samples augmented from the same image.

| Method | CUB | | Cars | | Places | |
|---|---|---|---|---|---|---|
| | (5,5) | (5,20) | (5,5) | (5,20) | (5,5) | (5,20) |
| *Unsupervised Few-shot Learning* | | | | | | |
| MetaSVEBM | 45.893 ± 0.334 | 54.823 ± 0.347 | 33.530 ± 0.367 | 44.622 ± 0.299 | 50.516 ± 0.397 | 61.561 ± 0.412 |
| MetaGMVAE | 48.783 ± 0.426 | 55.651 ± 0.367 | 30.205 ± 0.334 | 39.946 ± 0.400 | 55.361 ± 0.237 | 65.520 ± 0.374 |
| PsCo | 56.365 ± 0.636 | 69.298 ± 0.523 | 44.632 ± 0.726 | 56.990 ± 0.551 | 64.501 ± 0.780 | 73.516 ± 0.499 |
| *Self-supervised Learning* | | | | | | |
| SimCLR | 51.389 ± 0.365 | 60.011 ± 0.485 | 38.639 ± 0.432 | 52.412 ± 0.783 | 59.523 ± 0.461 | 68.419 ± 0.500 |
| MoCo | 52.843 ± 0.347 | 61.204 ± 0.429 | 39.504 ± 0.489 | 50.108 ± 0.410 | 60.291 ± 0.583 | 69.033 ± 0.654 |
| SwAV | 51.250 ± 0.530 | 61.645 ± 0.411 | 36.352 ± 0.482 | 51.153 ± 0.399 | 58.789 ± 0.403 | 68.512 ± 0.466 |
| SimCLR + GeSSL | 55.541 ± 0.456 | 64.489 ± 0.198 | 43.656 ± 0.199 | 55.841 ± 0.248 | 64.846 ± 0.300 | 72.651 ± 0.244 |
| MoCo + GeSSL | **57.485 ± 0.235** | 65.348 ± 0.279 | **45.348 ± 0.319** | 55.094 ± 0.248 | **66.489 ± 0.198** | **73.983 ± 0.251** |
| SwAV + GeSSL | 55.289 ± 0.190 | **65.839 ± 0.498** | 42.015 ± 0.315 | **56.481 ± 0.420** | 64.452 ± 0.350 | 72.237 ± 0.481 |

| Method | CropDiseases | | ISIC | | ChestX | |
|---|---|---|---|---|---|---|
| | (5,5) | (5,20) | (5,5) | (5,20) | (5,5) | (5,20) |
| *Unsupervised Few-shot Learning* | | | | | | |
| MetaSVEBM | 71.652 ± 0.837 | 84.515 ± 0.902 | 37.106 ± 0.732 | 48.001 ± 0.723 | 27.238 ± 0.685 | 29.652 ± 0.610 |
| MetaGMVAE | 72.683 ± 0.527 | 80.777 ± 0.511 | 30.630 ± 0.423 | 37.574 ± 0.399 | 24.522 ± 0.405 | 26.239 ± 0.422 |
| PsCo | **89.565 ± 0.372** | 95.492 ± 0.399 | 43.632 ± 0.400 | 54.886 ± 0.359 | 21.907 ± 0.258 | 24.182 ± 0.389 |
| *Self-supervised Learning* | | | | | | |
| SimCLR | 80.360 ± 0.488 | 89.161 ± 0.456 | 44.669 ± 0.510 | 51.823 ± 0.411 | 26.556 ± 0.385 | 30.982 ± 0.422 |
| MoCo | 81.606 ± 0.485 | 90.366 ± 0.377 | 44.328 ± 0.488 | 52.398 ± 0.396 | 24.198 ± 0.400 | 27.893 ± 0.412 |
| SwAV | 80.055 ± 0.502 | 89.917 ± 0.539 | 43.200 ± 0.356 | 50.109 ± 0.350 | 21.252 ± 0.439 | 28.270 ± 0.417 |
| SimCLR + GeSSL | 84.298 ± 0.428 | 94.438 ± 0.348 | **47.546 ± 0.402** | 55.486 ± 0.345 | **30.560 ± 0.277** | **34.343 ± 0.415** |
| MoCo + GeSSL | **85.667 ± 0.374** | **95.520 ± 0.345** | 46.437 ± 0.347 | **56.676 ± 0.280** | 29.258 ± 0.344 | 31.468 ± 0.290 |
| SwAV + GeSSL | 85.274 ± 0.345 | 94.667 ± 0.350 | 46.463 ± 0.291 | 55.203 ± 0.317 | 27.237 ± 0.355 | 32.130 ± 0.211 |

(Definition 3.2) in Section 3.2 to help evaluate the model universality, and further build GeSSL based on it to explicitly model universality into the SSL's learning objective. In this Section, we specifically quantify the universality scores of existing SSL methods based on $\sigma-$measure, and verify that our proposed GeSSL actually improves the model universality.

Specifically, we chose two scenarios based on images and videos to evaluate the model versatility following Liu et al. (2022b). The image-based tasks include linear probing (top-1 accuracy) with 800-epoch pre-trained models (LIN), semi-supervised classification (top-1 accuracy) using 1% subset of training data (SEMI), object detection (AP) on VOC dataset (VOC) and COCO dataset (COCO), instance segmentation ($AP^{mask}$) on COCO dataset (SEG). For video-based tasks, we compute rankings in terms of AUC for SOT, $\mathcal{J}$-mean for VOS, IDF-1 for MOT, IDF-1 for PoseTracking, and IDF-1 for MOTS, respectively. Next, we evaluate the $\sigma$-measurement scores of different baselines before and after the introduction of GeSSL and after training for 200 epochs. Among them, the better model is set to the result of ground truth, and the calculation of $\sigma$-measurement score is performed on a series of randomly sampled tasks.

Specifically, the $\sigma$-measurement score assesses the difference in performance between the learned model and the optimal model for each task. The optimal model is assumed to output the ground truth, and the performance difference is quantified using the KL divergence between the predicted and true

Table 13: Top-1 validation accuracy on ImageNet-1K dataset for ViT-B and ViT-L.

| Method | Epoch | ViT-B | ViT-L |
|---|---|---|---|
| data2vec 2.0 | 200/150 | 80.5 | 81.8 |
| data2vec 2.0 + GeSSL | 200/150 | 85.2 | 86.7 |

Table 14: Downstream classification accuracy of SimCLR-SAS on CIFAR-10.

| Method | Subset Size | Top-1 Accuracy (%) |
|---|---|---|
| SimCLR-SAS | 10% | 79.7 |
| SimCLR-SAS + GeSSL | 10% | 82.0 |

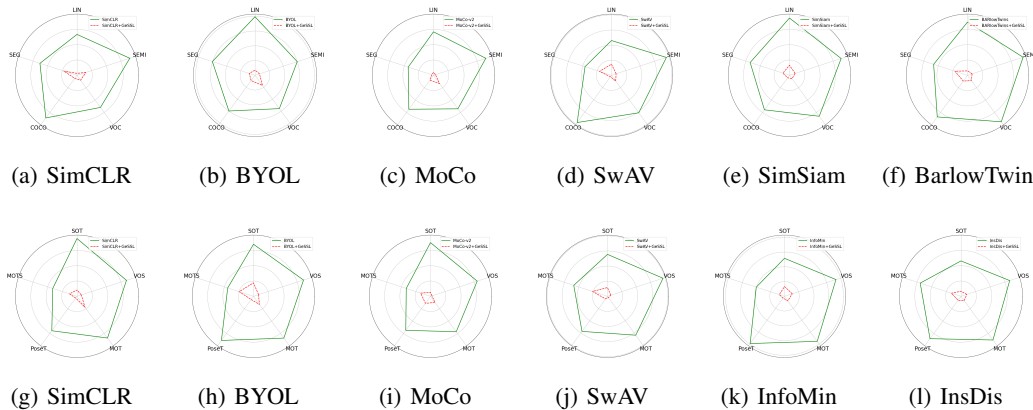

(a) SimCLR    (b) BYOL    (c) MoCo    (d) SwAV    (e) SimSiam    (f) BarlowTwins

(g) SimCLR    (h) BYOL    (i) MoCo    (j) SwAV    (k) InfoMin    (l) InsDis

Figure 6: Universality performance of different models on five image-based tasks (top row) and five video-based tasks (bottom row). We choose $\sigma-$measure as the measurement. It is worth noting that the smaller the $\sigma-$measurefen score, the better the effect. Meanwhile, we normalize the results of $\sigma-$measurefen scores on different datasets and compare the performance between baselines by comparing the corresponding branch of the fan chart.

class probability distributions. It compares the predicted class probabilities produced by classifier $\pi$ to the true labels across SSL tasks, such as comparing the predicted values $[0.81, 0.09, 0.03, 0.07]$ to the true labels $[1, 0, 0, 0]$. Take LIN task with SimCLR as an example, we train SimCLR and SimCLR+GeSSL on the COCO dataset for 200 epochs, then add a classification head after the feature extractor. A new mini-batch is input into both SimCLR and SimCLR+GeSSL to generate class probability distributions for each sample, and the KL divergence between these predicted and true distributions is calculated. After normalization, the scores for the LIN task are obtained, with similar evaluations conducted for other baselines and tasks.

Figure 6 shows the comparison results. Note that the lower $\sigma-$measure denotes the better performance. From the results, we can observe that: (i) the $\sigma$-measurement score of the existing SSL model is low and it is difficult to achieve good results in multiple domains and tasks; (ii) after the introduction of GeSSL, the $\sigma$-measurement score of the SSL models are significantly decreased. The results demonstrate that the existing SSL model has limited universality (proves the description in Section 1), and the performance improvement brought by GeSSL is achieved by improving the universality.

Considering that the above experiments evaluate the evaluation universality of SSL models, here, we construct the following numerical experiments to evaluate learning universality: In the first 20-200 epochs of training (each epoch contains multiple tasks), we evaluate the average performance of multiple $f_\theta^l$ in each epoch. Each $f_\theta^l$ is obtained by updating $f_\theta$ on the corresponding training tasks **with one step**. We calculate the accuracy of SimCLR before and after the introduction of GeSSL and the ratio $r$ of their effects on the CIFAR-10 data set. If $r < 1$, it means that the representation effect learned by the model in each epoch of training is better when introducing GeSSL. The results for every 20 epochs are shown in Table 15. The results show that: (i) $r$ is always less than 1, which proves that the representation effect learned after the introduction of GeSSL is significantly improved; (ii) after the introduction of GeSSL, the accuracy of the model is significantly improved, and it becomes stable after 80 epochs, i.e., great results can be achieved for even based on just one iteration and few data. These results show that "the model $f_\theta$ achieves comparable performance on each task quickly with few data during training" after introducing GeSSL.

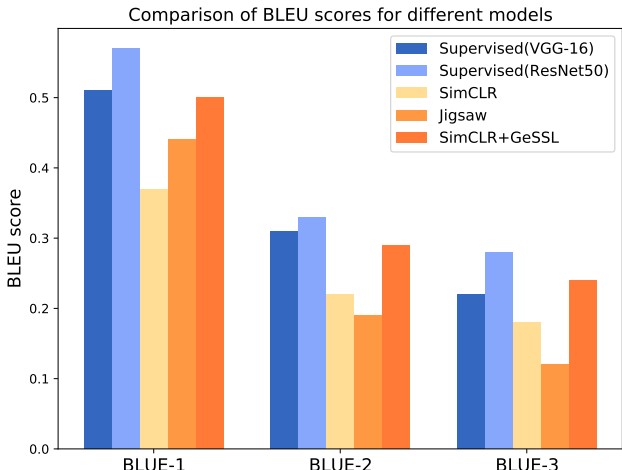

Figure 7: Comparison of BLEU scores for different models, comparing 2 fully supervised and 3 self-supervised pre-text tasks, trained on the Flickr8k.

Table 15: The performance of introducing GeSSL during training. All results are recorded during training using the $\sigma$-measurement.

| Metric | Training Epochs | | | | | | | | | |
|---|---|---|---|---|---|---|---|---|---|---|
| | 20 | 40 | 60 | 80 | 100 | 120 | 140 | 160 | 180 | 200 |
| Accuracy of SimCLR | 20.1 | 43.6 | 51.2 | 60.2 | 70.3 | 77.2 | 82.3 | 86.1 | 88.7 | 88.6 |
| Accuracy of SimCLR + GeSSL | 41.9 | 66.3 | 82.1 | 93.5 | 93.4 | 93.0 | 93.6 | 93.7 | 93.7 | 93.8 |
| Performance Ratio $r$ | 0.479 | 0.657 | 0.623 | 0.643 | 0.752 | 0.830 | 0.879 | 0.918 | 0.946 | 0.944 |

## F.5 EVALUATION ON GENERATIVE SELF-SUPERVISED LEARNING

In this Section, we evaluate the effectiveness of the proposed GeSSL on the generative self-supervised learning paradigm. We conduct experiments on three scenarios, including image generation, image captioning, and object detection and segmentation.

**Evaluation on Image Generation**   To explore the effect of GeSSL on generative SSL, we conduct a set of experiments on ImageNet-1K dataset Deng et al. (2009b). Specifically, we begin by conducting self-supervised pre-training on the ImageNet-1K (IN1K) training set. Following this, we carry out supervised training to assess the representations using either (i) end-to-end fine-tuning or (ii) linear probing. The results are reported as the top-1 validation accuracy for a single 224×224 crop. For this process, we utilize ViT-Large (ViT-L/16) Dosovitskiy et al. (2020) as the backbone. Note that ViT-L is very big (an order of magnitude bigger than ResNet-50 He et al. (2016)) and tends to overfit, as shown in Table 16. The comparison results are shown in Table 17. We can observe that GeSSL achieves stable performance improvements

**Evaluation on Image Captioning**   We use the commonly used protocol following Mohamed et al. (2022). The dataset we use to train the pretext task is the unlabeled part of MSCOCO dataset Vinyals et al. (2016b), which contains 123K images with an average resolution of $640 \times 480$ pixels. This dataset contains color and grayscale images. For downstream tasks, we use the Flicker8K dataset Hodosh et al. (2013). Next, we train it using pre-trained pre-text tasks supervised by VGG-16 and ResNet-50, as well as self-supervised pre-text tasks from SimCLR and Jigsaw Puzzle solutions. In the next step, to evaluate the results, we use the BLEU (Bilingual Evaluation Research) score as the evaluation metric, which evaluates the generated sentences against the reference sentences, where a perfect match is 1 and a perfect mismatch is 0, calculating scores for 1, 2, 3 and 4 cumulative n-grams. The results are shown in Figure 7. From the results, we can observe that after introducing the GeSSL framework we proposed, the model effect has been further improved, stably exceeding the SOTA of

Table 16: Comparison between models.

| Method | scratch, original | scratch, our impl. | baseline MAE | MAE + Our |
|--------|-------------------|--------------------|--------------|-----------|
| Top 1  | 76.5              | 82.5               | 85.3         | 87.2      |

Table 17: Comparisons with previous results on ImageNet-1K. The ViT models are B/16, L/16, H/14 Dosovitskiy et al. (2020). The pre-training data is the ImageNet-1K training set (except the tokenizer in BEiT was pre-trained on 250M DALLE data Ramesh et al. (2021)). All results are on an image size of 224, except for ViT-H with an extra result of 448.

| Method | pre-train data | ViT-B | ViT-L | ViT-H | ViT-H$_{448}$ |
|--------|----------------|-------|-------|-------|---------------|
| DINO   | IN1K           | 82.8  | -     | -     | -             |
| MoCo   | IN1K           | 83.2  | 84.1  | -     | -             |
| BEiT   | IN1K+DALLE     | 83.2  | 85.2  | -     | -             |
| MAE    | IN1K           | 83.6  | 85.9  | 86.9  | 87.8          |
| MAE+Ours | IN1K         | 86.9  | 87.6  | 88.9  | 89.1          |

the SSL method, and even approaching the supervised learning results. The results show that our proposed GeSSL can still achieve good results in generative self-supervised learning.

**Evaluation on Object Detection and Segmentation**    For object detection and segmentation, we fine-tune Mask R-CNN  He et al. (2017) end-to-end on COCO Lin et al. (2014b). The ViT backbone is adapted for use with FPN Lin et al. (2017). We report box AP for object detection and mask AP for instance segmentation. The results are shown in Table 18. Compared to supervised pre-training, our MAE performs better under all configurations. Our method still achieves optimal results, demonstrating its effectiveness.

F.6    EVALUATION ON MORE MODALITIES

GeSSL proposed in this work can be applied in various fields and domains, e.g., instance segmentation, video tracking, sample generation, etc., as mentioned before. Here, we provide the experiments of GeSSL on text modality-based datasets, i.e., IC03 and IIIT5K Yasmeen et al. (2020), which we have conducted before. We follow the same experimental settings as mentioned in Aberdam et al. (2021). The results

Table 19:  Performance on for text recognition.

| Methods | IIIT5K | IC03 |
|---------|--------|------|
| SimCLR Chen et al. (2020a) | 1.7 | 3.8 |
| SeqCLR Aberdam et al. (2021) | 35.7 | 43.6 |
| SimCLR + GeSSL | 19.0 | 19.2 |
| SeqCLR + GeSSL | 39.0 | 49.0 |

shown in Table 19 demonstrate that GeSSL achieves stable effectiveness and robustness in various modalities combined with the above experiments.

Table 18: COCO object detection and segmentation using a ViT Mask R-CNN baseline. All self-supervised entries use IN1K data without labels, and Mask AP follows a similar trend as box AP.

| Method | pre-train data | AP$^{box}$ | | AP$^{mask}$ | |
|--------|----------------|-------|-------|-------|-------|
|        |                | ViT-B | ViT-L | ViT-B | ViT-L |
| supervised | IN1K w/ labels | 47.9 | 49.3 | 42.9 | 43.9 |
| MoCo v3 | IN1K | 47.9 | 49.3 | 42.7 | 44.0 |
| BEiT | IN1K+DALLE | 49.8 | 53.3 | 44.4 | 47.1 |
| MAE | IN1K | 50.3 | 53.3 | 44.9 | 47.2 |
| MAE + Our | IN1K | 54.2 | 56.1 | 46.7 | 50.1 |

# G  DETAILS OF ABLATION STUDY

In this section, we introduce the experimental details and more comprehensive analysis of the ablation studies (Subsection 5.6), including influence of $\lambda$, model efficiency, role of loss, and implementation of bi-level optimization. In addition, we further conduct ablation experiments for task construction, and display the experimental settings and results in Appendix G.5

## G.1  INFLUENCE OF $\lambda$

This ablation study evaluates the effect of the hyperparameter $\lambda$ in the self-motivated target. Recall that GeSSL explicitly models universality into self-supervised learning, and as mentioned in Section 3.1 of the main text, universality involves two aspects, including: (i) learning universality, i.e., the model $f_\theta$ which learns universal representations during training, should achieve competitive performance on each task quickly with few data; (ii) evaluation universality, i.e., the trained $f_\theta^*$, which has learned universal representations, should adapt to different tasks simultaneously with minimal additional data. Therefore, we hope that GeSSL can enable the model to achieve optimal results based on a few update steps. Our experimental setup constraints several conditions: (i) fast adaptation: keep the update steps $K$ of the inner-loop optimization in a small range of $K \in [1, 15]$; (ii) few data: use miniImageNet as the benchmark dataset, and follow the settings of few-shot learning experiments; and (iii) performance evaluation: evaluate the effect of SimCLR + GeSSL, in addition to evaluating the accuracy under different $\lambda$, we can also compare with the results of few-shot learning experiments (Subsection 5.5 and Table 4).

The results of the ablation experiment about "influence of $\lambda$" are presented in Table 5 of the main text. Through further analysis, we derive two additional conclusions: (i) Combining with Table 4 of the main text, regardless of the value of $K$, SimCLR + GeSSL consistently outperforms SimCLR on miniImageNet, demonstrating the performance enhancement brought by GeSSL; (ii) Considering Figure 2 of the main text, despite the introduction of universality constraints by GeSSL, the computational efficiency of SimCLR + GeSSL remains better than that of SimCLR, proving the efficiency improvement brought by GeSSL.

## G.2  MODEL EFFICIENCY

This ablation study explores the efficiency of self-supervised models before and after applying GeSSL. Specifically, we choose five baselines, including SimCLR Chen et al. (2020a), MOCO Chen et al. (2020b), BYOL Grill et al. (2020), Barlow Twins Zbontar et al. (2021), and SwAV Caron et al. (2020). Then, we evaluate the accuracy, training hours, and parameter size of these models on STL-10 before and after applying our proposed GeSSL. We use the same linear evaluation setting as in Section 5.1 of the main text. The setting for GeSSL is "K=1" and "$\lambda = 10$". Finally, we plot the trade-off scatter plot by recording the average values of five runs. The results are shown in Figure 2 of the main text, where the horizontal axis represents the training hours and the vertical axis represents the accuracy. The center of each circle represents the result of the training time and accuracy of each model, and the area of the circle represents the parameter size. The numerical results of this experiment are shown in Table 21. From the results, we can see that: (i) GeSSL can significantly improve the performance and computational efficiency of self-supervised learning models; (ii) our designed self-motivated target achieves the goal of guiding the model update toward universality with few samples and fast adaptation; (iii) although GeSSL optimizes based on bi-level optimization, the impact of the increased parameter size of GeSSL is negligible.

Note that although the optimization method used by GeSSL is more complex, one of its core goals is to accelerate model convergence, i.e., achieve greater performance improvement per unit of time. This does not imply that GeSSL always requires fewer epochs to reach the optimal result. In fact, GeSSL uses approximate implicit differentiation with finite difference (AID-FD) for updates instead of conventional explicit second-order differentiation (as mentioned in Appendix G.4). Moreover, GeSSL constructs a self-

Table 20: Training cost per epoch of SSL models.

| Methods | Training Cost per Epoch (s) |
|---|---|
| SimCLR Chen et al. (2020a) | 12.8 |
| MOCO Chen et al. (2020b) | 16.9 |
| SimCLR + GeSSL | 9.4 |
| MOCO + GeSSL | 11.1 |

Table 21: Model analysis including parameter size, training time, and performance.

| Methods | Memory Footprint (MiB) | Parameter Size (M) | Training Time (h) | Accuracy (%) |
|---|---|---|---|---|
| SimCLR | 2415 | 23.15 | 4.15 | 90.5 |
| MOCO | 2519 | 24.01 | 4.96 | 90.9 |
| BYOL | 2691 | 25.84 | 6.98 | 91.9 |
| BarlowTwins | 2477 | 23.15 | 5.88 | 90.3 |
| SwAV | 2309 | 22.07 | 4.45 | 90.7 |
| SimCLR+GeSSL | 2713 | 26.05 | 3.36 | 93.1 |
| MOCO+GeSSL | 2801 | 27.01 | 4.17 | 94.2 |
| BYOL+GeSSL | 2902 | 28.05 | 5.64 | 94.5 |
| BarlowTwins+GeSSL | 2833 | 27.07 | 5.22 | 93.9 |
| SwAV+GeSSL | 2971 | 28.50 | 3.91 | 92.8 |

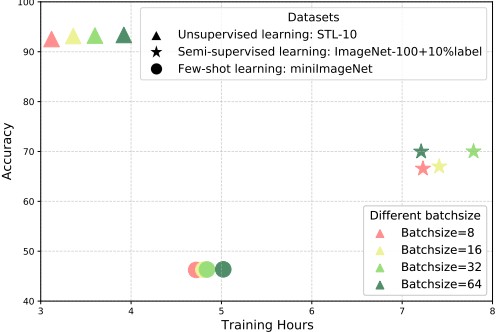 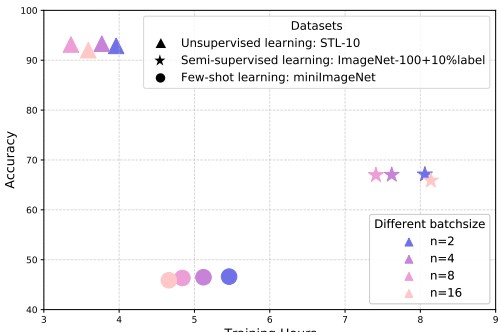

Figure 8: The effect of batchsize in SSL task construction (also the number of classes in SSL task) for GeSSL.

Figure 9: The effect of $n$ in the outer-loop optimization (also the number of SSL tasks that are learned simultaneously) for GeSSL.

motivated target that guides the model to optimize more effectively in a specific task. Therefore, the efficiency improvement is reflected in the computational efficiency and effectiveness of updates per epoch, rather than simply reducing the total number of epochs. Furthermore, to verify whether the efficiency improvement is attributable to a single epoch, we separately measured the computational overhead of SSL baseline algorithms after integrating GeSSL for a single epoch. The results, presented in Table 20, demonstrate that with a consistent batch size, GeSSL enhances the computational efficiency and the effectiveness of updates per epoch for the SSL baseline algorithms.

### G.3 ROLE OF LOSS

This ablation study explores the role of the loss function in the outer-loop optimization of GeSSL. The goal of the outer-loop optimization is to update the model towards universality, and the choice of loss function directly affects the model performance. Therefore, we select four commonly used loss functions, including MSE Tsai et al. (2020), cross-entropy De Boer et al. (2005), KL divergence Hershey & Olsen (2007), and Wasserstein distance Panaretos & Zemel (2019). We record the performance and training time of SimCLR + GeSSL with different losses on STL-10. These loss functions are computed as follows:

**MSE (mean squared error)** Tsai et al. (2020) calculates the mean of the squared difference between model predictions and true values. The advantage of MSE is that it is simple to calculate, and the disadvantage is that it is sensitive to outliers. The formula for MSE is:

$$\text{MSE}(y, \hat{y}) = \frac{1}{n} \sum_{i=1}^{n} (y_i - \hat{y}_i)^2 \tag{14}$$

where $y$ is the true value, $\hat{y}$ is the predicted value, and $n$ is the number of samples.

**Cross-entropy** De Boer et al. (2005) is a loss function used for classification problems, which calculates the difference between model-predicted probabilities and true probabilities. The advantage of cross-entropy is that it can reflect the uncertainty of the model, and the disadvantage is that it may cause the gradient to vanish or explode. The formula for cross-entropy is:

$$\text{CE}(y, \hat{y}) = -\sum_{i=1}^{n} y_i \log \hat{y}_i \tag{15}$$

where $y$ is the true probability, $\hat{y}$ is the predicted probability, and $n$ is the number of classes.

**KL divergence (Kullback-Leibler divergence)** Hershey & Olsen (2007) is a measure of the similarity between two probability distributions, which can be seen as the difference between cross-entropy and entropy. The advantage of KL divergence is that it can reflect the distance between distributions, and the disadvantage is that it is asymmetric and may be unbounded. The formula for KL divergence is:

$$\text{KL}(P\|Q) = \sum_{i} P(i) \log \frac{P(i)}{Q(i)} \tag{16}$$

where $P$ is the true distribution, $Q$ is the predicted distribution.

**Wasserstein distance** Panaretos & Zemel (2019) is a measure of the distance between two probability distributions, which can be seen as the minimum cost of transforming one distribution into another. The advantage of Wasserstein distance is that it can reflect the geometric structure of the distributions, and the disadvantage is that it is computationally complex and requires regularization. The formula for Wasserstein distance is:

$$\text{WD}(P, Q) = \inf_{\gamma \in \Pi(P,Q)} \mathbb{E}_{(X,Y) \sim \gamma}[\|X - Y\|] \tag{17}$$

where $P$ is the true distribution, $Q$ is the predicted distribution, $\Pi(P, Q)$ is the set of all joint distributions that couple $P$ and $Q$, and $\| \cdot \|$ is some distance measure.

From empirical analysis, Figure 3 in the main text provides the experimental results. We find that GeSSL achieves the best balance between accuracy and computational efficiency when using self-motivated target with KL divergence, i.e., the model achieves the highest accuracy in the shortest training time. Specifically, whether from the accuracy or the computational efficiency, applying KL divergence to evaluate the distribution difference and then update the model is much more efficient than applying MSE and cross-entropy losses. Although applying Wasserstein distance achieves similar accuracy, its computational time is significantly larger than applying KL divergence. Thus, we use KL divergence to optimize our model in the outer-loop optimization.

From theoretical analysis, the key "optimal universality properties" for a metric in practical applications include: (i) the ability to accurately quantify subtle differences between distributions, (ii) its utility in model optimization for stable and efficient convergence to the global optimum, (iii) applicability to various complex distributions, and (iv) computational efficiency. Accordingly, the superiority of KL divergence is reflected in three aspects Hershey & Olsen (2007); Goldberger et al. (2003); Shlens (2014), meeting these properties. Firstly, KL divergence is non-negative, and it is zero if and only if the two distributions are exactly the same, which is consistent with our intuitive understanding of difference Gong et al. (2021). It ensures the stability of KL divergence in handling subtle differences, meeting (i) and (iv). Secondly, KL divergence is a convex function, which means that optimizing it is more likely to converge to the global optimum, rather than getting stuck in the local optimum, particularly in high-dimensional problems Hershey & Olsen (2007). Thus, this ensures that KL divergence meets (ii). Additionally, as an extension of information entropy, KL divergence quantifies information loss and uncertainty, making it effective across various applications Goldberger et al. (2003), especially self-rewarding learning tasks, meeting (iii). In contrast, other metrics have notable limitations. MSE, based on Euclidean distance, is sensitive to outliers and fails to account for non-negativity or normalization of probability distributions Marmolin (1986); Chicco et al. (2021); Lebanon (2010), limiting its effectiveness in (i) and (iii). Cross-entropy, a special case of KL divergence, struggles with continuous distributions or when the true distribution isn't a one-hot vector De Boer et al. (2005); Botev et al. (2013), limiting its ability to finely measure complex distributions (i) and (iii). Lastly, while Wasserstein distance captures the overall shape difference between distributions, its high computational complexity and requirement for smoothness conditions make it less suited for high-dimensional cases Panaretos & Zemel (2019); Vallender (1974), hindering its

fulfillment of (iv). Thus, KL divergence achieves the optimal balance between theoretical robustness and computational feasibility, aligning with the "optimal universality properties" and resulting in better model generalization and lower training costs.

### G.4 IMPLEMENTATION OF THE BI-LEVEL OPTIMIZATION

The model of GeSSL is updated based on bi-level optimization, and the model gradients for each level are obtained by combining the optimal response Jacobian matrices through the chain rule. In practical applications, multi-level gradient computation requires a lot of memory and computation Choe et al. (2022), so we hope to introduce a more concise gradient backpropagation and update method to reduce the computational complexity. Specifically, we consider two types of gradient update methods, including iterative differentiation (ITD) Finn et al. (2017a) and approximate implicit differentiation (AID) Grazzi et al. (2020). We provide implementations of four popular ITD/AID algorithms, including ITD with reverse-mode automatic differentiation (ITD-RMAD) Finn et al. (2017a), AID with Neumann series (AID-NMN) Lorraine et al. (2020), AID with conjugate gradient (AID-CG) Rajeswaran et al. (2019), and AID with finite difference (AID-FD) Liu et al. (2018). We also choose the recently proposed optimizer, i.e., Lookahead Zhang et al. (2019) for comparison. We denote the the upper-level parameters and the lower-level parameters as $\theta$ and $\phi$, respectively. All the way of gradient update of the bi-level optimization are as follows:

**ITD-RMAD** Finn et al. (2017a), ITD with reverse-mode automatic differentiation applies the implicit function theorem to the lower-level optimization problem and computes the gradients of the upper-level objective with respect to the upper-level parameters using reverse-mode automatic differentiation. The update process is as follows:

- Solve the lower-level optimization problem $\phi^* = \arg\min_\phi L(\phi, \theta)$ using gradient descent.
- Compute the gradient of the upper-level objective $g(\theta) = F(\phi^*, \theta)$ with respect to $\theta$ using reverse-mode automatic differentiation:

$$\nabla_\theta g(\theta) = \nabla_\theta F(\phi^*, \theta) - \nabla_\phi F(\phi^*, \theta)^T (\nabla_\phi L(\phi^*, \theta))^{-1} \nabla_\theta L(\phi^*, \theta) \tag{18}$$

- Update the upper-level parameters using gradient descent or other methods: $\theta \leftarrow \theta - \alpha \nabla_\theta g(\theta)$.

**AID-NMN** Lorraine et al. (2020), AID with Neumann series, approximates the inverse of the Hessian matrix of the lower-level objective using a truncated Neumann series expansion and computes the gradients of the upper-level objective with respect to the upper-level parameters using forward-mode automatic differentiation. The update process is as follows:

- Solve the lower-level optimization problem $\phi^* = \arg\min_\phi L(\phi, \theta)$ using gradient descent.
- Compute the gradient of the upper-level objective $g(\theta) = F(\phi^*, \theta)$ with respect to $\theta$ using forward-mode automatic differentiation:

$$\begin{aligned} \nabla_\theta g(\theta) &= \nabla_\theta F(\phi^*, \theta) - \nabla_\phi F(\phi^*, \theta)^T (\nabla_\phi L(\phi^*, \theta))^{-1} \nabla_\theta L(\phi^*, \theta) \\ &\approx \nabla_\theta F(\phi^*, \theta) - \nabla_\phi F(\phi^*, \theta)^T \sum_{k=0}^{K} (-1)^k (\nabla_\phi^2 L(\phi^*, \theta))^k \nabla_\theta L(\phi^*, \theta) \end{aligned} \tag{19}$$

where $K$ is the truncation order of the Neumann series.

- Update the upper-level parameters using gradient descent or other methods: $\theta \leftarrow \theta - \alpha \nabla_\theta g(\theta)$.

**AID-CG** Rajeswaran et al. (2019), AID with conjugate gradient, solves a linear system involving the Hessian matrix of the lower-level objective using the conjugate gradient algorithm and computes the gradients of the upper-level objective with respect to the upper-level parameters using forward-mode automatic differentiation. The update process is as follows:

- Solve the lower-level optimization problem $\phi^* = \arg\min_\phi L(\phi, \theta)$ using gradient descent or other methods.

- Compute the gradient of the upper-level objective $g(\theta) = F(\phi^*, \theta)$ with respect to $\theta$ using forward-mode automatic differentiation:

$$
\begin{aligned}
\nabla_\theta g(\theta) &= \nabla_\theta F(\phi^*, \theta) \\
&\quad -\nabla_\phi F(\phi^*, \theta)^T (\nabla_\phi L(\phi^*, \theta))^{-1} \nabla_\theta L(\phi^*, \theta) \approx \nabla_\theta F(\phi^*, \theta) \\
&\quad -\nabla_\phi F(\phi^*, \theta)^T v
\end{aligned}
\tag{20}
$$

where $v$ is the solution of the linear system $(\nabla_\phi^2 L(\phi^*, \theta))v = \nabla_\theta L(\phi^*, \theta)$ obtained by the conjugate gradient algorithm.

- Update the upper-level parameters using gradient descent or other methods: $\theta \leftarrow \theta - \alpha \nabla_\theta g(\theta)$.

**AID-FD** Liu et al. (2018), AID with finite difference, approximates the inverse of the Hessian matrix of the lower-level objective using a finite difference approximation and computes the gradients of the upper-level objective with respect to the upper-level parameters using forward-mode automatic differentiation. The update process is as follows:

- Solve the lower-level optimization problem $\phi^* = \arg \min_\phi L(\phi, \theta)$ using gradient descent or other methods.
- Compute the gradient of the upper-level objective $g(\theta) = F(\phi^*, \theta)$ with respect to $\theta$ using forward-mode automatic differentiation:

$$
\begin{aligned}
\nabla_\theta g(\theta) &= \nabla_\theta F(\phi^*, \theta) \\
&\quad -\nabla_\phi F(\phi^*, \theta)^T (\nabla_\phi L(\phi^*, \theta))^{-1} \nabla_\theta L(\phi^*, \theta) \\
&\approx \nabla_\theta F(\phi^*, \theta) \\
&\quad -\nabla_\phi F(\phi^*, \theta)^T \frac{\nabla_\theta L(\phi^* + \epsilon \nabla_\theta L(\phi^*, \theta), \theta) - \nabla_\theta L(\phi^*, \theta)}{\epsilon}
\end{aligned}
\tag{21}
$$

where $\epsilon$ is a small positive constant for the finite difference approximation.

- Update the upper-level parameters using gradient descent or other methods: $\theta \leftarrow \theta - \alpha \nabla_\theta g(\theta)$.

**Lookahead** Zhang et al. (2019) introduces a novel approach to optimization by maintaining two sets of weights: the fast and the slow weights. The fast weights, $\theta_{\text{fast}}$, are updated frequently through standard optimization techniques, while the slow weights, $\theta_{\text{slow}}$, are updated at a lesser frequency. The key formula that updates the slow weights is given by:

$$
\theta_{\text{slow}} \leftarrow \theta_{\text{slow}} + \alpha(\theta_{\text{fast}} - \theta_{\text{slow}})
\tag{22}
$$

where $\alpha$ is a hyperparameter controlling the step size. This method aims to stabilize training and ensure consistent convergence.

The results shown in Figure 4 of the main text demonstrate that approximate implicit differentiation with finite difference also achieves optimal results on the SSL model. Our optimization process is also based on this setting.

## G.5 EFFECT OF TASK CONSTRUCTION

GeSSL learns from a series of self-supervised learning tasks that are constructed based on data augmentation (Subsection 2 in the main text). Specifically, the augmented data from the same image have significant entity similarity, so we assign the same class label $y_j \in \mathcal{Y}$ to the augmented data from the same image $x_j$. Therefore, a batch of SSL can be viewed as a multi-class classification problem, where each class contains two samples. Then, the training data of $n$ batches of self-supervised learning can form $n$ self-supervised learning tasks. The reliability of this view is also well recognized by the SSL community Oord et al. (2018); Hjelm et al. (2018); Tian et al. (2020b). Comparing them with the task construction of this work, they all construct the task concept based on approximate view invariance theory but with differences. Specifically, the previously proposed methods mainly focus on

contrastive SSL, where the classification task concept is to access the samples with the same content features for the same class and then according to the results to calculate mutual information for learning. This work considers both discriminative and generative self-supervised learning paradigms and presents a unified understanding of SSL tasks based on the presented alignment and regularization stage with pseudo-labeling. Meanwhile, we would like to clarify that understanding SSL from a task perspective is not the core contribution of our work, but rather part of the background for our proposed methodology.

Considering that our framework updates the self-supervised model $f_\theta$ in GeSSL based on these $n$ tasks simultaneously, the number of sampled samples per batch of self-supervised learning directly determines the class diversity of the data in the task. In this section, we further conduct ablation experiments on the batch size (the number of classes) of the tasks and the number of self-supervised learning tasks $n$ that are learned simultaneously.

Specifically, we choose the commonly used STL-10 for unsupervised learning, ImageNet with 10% label for semi-supervised learning, and miniImageNet for few-shot learning, and evaluate the performance of SimCLR + GeSSL under different batch sizes and different $n$ values. Figure 8 shows the impact of different batch sizes (i.e., the number of classes in the multi-class classification task) for SSL. The results show that SimCLR + GeSSL always outperforms SimCLR under any batch size. A larger batch size leads to a slightly larger performance improvement for SimCLR + GeSSL, but also increases the computational resource consumption. Therefore, in this study, we build tasks based on images with a batch size of $B = 16$ or $B = 32$. Figure 9 shows the impact of the update frequency $n$ (i.e., update $f_\theta$ every $n$ batches) for the outer-loop optimization. The results indicate that $n = 8$ is a better trade-off between model accuracy and time consumption. In the setting of our GeSSL, we also choose $n = 8$ as the hyperparameter setting.

In addition, considering that GeSSL updates every $n$ mini-batches, we evaluate the baseline performance under $n\times$ the original batch size. Specifically, we adopt the same experimental setup as in Figure 2, with the only difference being that we increase the batch size of the Sim-CLR baseline by a factor of $n$ and record the results. The results are shown in Table 22, which indicate that the performance of SimCLR, after converging with the larger training data, remains largely unchanged and still inferior to GeSSL.

Table 22: Performance on for a large batchsize.

| Methods | Accuracy | Training Cost |
|---|---|---|
| SimCLR Chen et al. (2020a) | 90.8 | 5.2 |
| SimCLR + GeSSL | 93.6 | 3.6 |

## H DIFFERENCES BETWEEN GeSSL AND META-LEARNING

In the main text, we have illustrated the differences between GeSSL and meta-learning and the advantages of GeSSL. In this section, we further elaborate on this and list different meta-learning methods for comparison.

Meta-learning Finn et al. (2017b); Wang et al. (2024b); Snell et al. (2017), often referred to as "learning to learn", has emerged as a prominent approach to improve the efficiency and adaptability of machine learning models, especially in scenarios with limited data. The fundamental idea behind meta-learning is to train models that can rapidly adapt to new tasks with minimal data by leveraging prior experiences gained from a range of related tasks.

Few-shot Learning Khodadadeh et al. (2019); Jang et al. (2023): One of the primary areas where meta-learning has demonstrated substantial impact is in few-shot learning. Methods like Model-Agnostic Meta-Learning (MAML) Finn et al. (2017b) aim to find a set of model parameters that are sensitive to changes in the task, allowing for quick adaptation to new tasks with just a few examples. Variants of MAML, such as First-Order MAML (FOMAML) and Reptile Nichol & Schulman (2018), reduce the computational complexity of the original algorithm while maintaining competitive performance.

Metric-based Approaches: Metric-based meta-learning methods, such as Matching Networks Sung et al. (2018) and Prototypical Networks Snell et al. (2017), learn an embedding space where similar tasks are closer together. These models perform classification by comparing the distance between new examples and a few labeled instances (support set) in this learned space, achieving remarkable results in few-shot classification tasks.

Memory-augmented Networks: Another line of research in meta-learning explores the use of external memory structures to facilitate rapid adaptation. Santoro et al introduced Memory-Augmented Neural Networks (MANNs) Santoro et al. (2016) that use an external memory to store and retrieve information about past tasks, enabling the model to perform well even in tasks with highly variable distributions.

Gradient-based Meta-learning: Beyond MAML, other gradient-based methods such as Meta-SGD Li et al. (2017) and Learning to Learn with Gradient Descent have been proposed. These methods modify the way gradients are used during the training of the model, either by learning the initial parameters (as in MAML) or by learning the learning rates for different parameters, allowing for more efficient adaptation.

Bayesian Meta-learning: Bayesian approaches to meta-learning, such as Bayesian MAML Zhang et al. (2021), offer a probabilistic framework for capturing uncertainty and improving generalization to new tasks. These methods have been particularly useful in scenarios where task distributions are diverse, and the model needs to account for uncertainty in task inference.

Meta-learning for Reinforcement Learning: Meta-learning has also been successfully applied in the domain of reinforcement learning (RL). Methods such as Meta-RL Yu et al. (2020) aim to train agents that can quickly adapt to new environments by leveraging the experience gained in previous tasks. These approaches have shown promise in enabling RL agents to solve tasks with minimal exploration, a crucial aspect for real-world applications where exploration can be costly or risky.

In summary, meta-learning has rapidly evolved as a versatile framework that enhances the ability of models to adapt quickly to new tasks, and operate efficiently in dynamic environments. Compared meta-learning with the proposed GeSSL, we can see that the main difference is that meta-learning only considers transferability, and does not model discriminability and generalization. First, the update of the outer model of meta-learning depends on the performance of the inner task-specific model. Considering that the model is based on episode training mechanism, it is only based on one update on a specific task. Therefore, if the model update on a specific task is insufficient, then the outer model is likely to be difficult to achieve good results on the task, affecting the discriminability. Secondly, the generalization evaluation of the meta-learning model depends on its performance on the query set, which pushes the model to overfit on the training tasks, thereby diminishing the model's ability to generalize.

## I  DISCUSSION OF DATA ISSUES

First, we would like to point out that even if all mini-batch data are independently and identically distributed (i.i.d.), it does not imply that the resulting tasks are homogeneous. On the contrary, under the i.i.d. assumption, task diversity can still be ensured in the following ways:

1. **Complexity of Data Distribution**: The i.i.d. assumption does not require data to be simple or homogeneous. The distribution can be complex, covering multiple classes and diverse sample characteristics. For instance, in a multi-class task, data can come from various classes, with high complexity within each class. Therefore, even if the data are sampled from the same distribution, the distribution itself can be complex enough to ensure task diversity, reflected by a rich feature space (e.g., high-dimensional data or different input types).

2. **Diversity Through Sampling**: Each mini-batch can be composed of different samples randomly drawn from the same distribution. This means that while the samples come from the same distribution, each mini-batch can have different sample combinations, with varying features and class ratios, presenting different learning challenges to the model.

3. **Data Augmentation**: In many deep learning and self-supervised learning methods, data augmentation is used to create diverse training tasks. Even if samples are i.i.d., using different augmentation techniques (e.g., cropping, rotation, color transformation) can provide the model with diverse inputs.

4. **SSL Task Construction**: We treat each mini-batch as a multi-class task, with each original sample corresponding to a label. Thus, different tasks have different label distributions.

Even if samples within a task are i.i.d., the label distribution varies across tasks, ensuring diversity.

5. **Theoretical Support**: Theoretically, the i.i.d. assumption does not restrict task homogeneity. Many theoretical works, such as Vapnik (1998) on statistical learning theory and Bengio et al. (2013) on representation learning, discuss how samples drawn independently from the same distribution can train models while maintaining task diversity and achieving generalization. These studies show that even under the i.i.d. assumption, tasks can encompass different data patterns and diverse features.

Even in cases of data scarcity or homogeneous tasks, not just data diversity, we have taken steps to ensure learning effectiveness:

1. **Definition Perspective**: In machine learning, data scarcity and homogeneous tasks can lead to overfitting to specific tasks, causing the model to learn all information, including background, making it hard to adapt to other tasks. Therefore, if we constrain the model to perform well on training data while maintaining effectiveness on unseen samples and tasks, we can ensure its robustness. As mentioned in Section 3.1, the defined universality considers both learning and evaluation levels, covering discriminability, generalizability, and transferability, involving known samples, unknown samples, and unseen tasks. Thus, even with data scarcity and homogeneous tasks, this definition ensures that the model learns a universal representation to maintain effectiveness.

2. **Modeling Perspective**: We further proposed GeSSL to model universality, including discriminability, generalizability, and transferability. As described in Section 3.4, for discriminability, GeSSL extracts key features from each mini-batch using limited data to achieve optimal performance. For generalizability, GeSSL ensures causal feature extraction during cross-task training. Finally, for transferability, GeSSL employs bi-level optimization to estimate the true task distribution from discrete training tasks. Thus, GeSSL models universality to ensure model effectiveness under limited data conditions.

3. **Empirical Perspective**: Experiments across over 25 baselines, 16 datasets, and five settings for both discriminative and generative SSL demonstrate stable and significant performance improvements with GeSSL, including in few-shot and cross-domain scenarios. This empirical evidence supports the effectiveness of our work in the face of data challenges.

We mitigate the impact of data homogeneity on three levels. First, in practical applications, it is challenging to ensure all sampled tasks are i.i.d. with homogeneous samples. Second, even under the i.i.d. assumption, diversity can be ensured as discussed in (1). Third, even if task diversity is difficult to achieve, we have addressed this in the definition, measurement, and modeling of universality by constraining discriminability, generalizability, and transferability to ensure the effectiveness of learned features. Extensive experiments support the effectiveness of the proposed method.

