# OpenReview forum: "On the Universality of Self-Supervised Representation Learning"
_ICLR.cc/2025/Conference — ICLR 2025 Conference Withdrawn Submission_

### Official Review · Reviewer_apsp · 2024-10-24

**Soundness:** 2
**Presentation:** 2
**Contribution:** 2
**Rating:** 5
**Confidence:** 3

**Summary:**

This paper advances self-supervised learning (SSL) by defining "Universality," which includes discriminability, generalizability, and transferability. This paper introducs GeSSL, a framework that models universality by enhancing discriminability with Kullback-Leibler divergence, promoting generalizability through causal feature extraction across tasks, and implementing a bi-level optimization for transferability.

**Strengths:**

This paper introduces a comprehensive framework for incorporating universality into self-supervised learning (SSL) by defining it through discriminability, generalization, and transferability. The σ-measurement quantifies performance gaps, strengthening the theoretical foundation of SSL methods. The GeSSL framework effectively integrates task-specific learning and focuses on causally invariant representations, enhancing adaptability and performance across diverse tasks. Overall, the paper addresses critical gaps in SSL methodologies and presents a novel approach supported by theoretical and empirical evidence.

**Weaknesses:**

1.The self-supervised learning (SSL) paradigm requires algorithms to exhibit good alignment and uniformity. However, the concepts of discriminability and alignment mentioned in the paper seem to overlap. Similarly, generalizability and transferability appear to be closely related to uniformity, as improved uniformity is essential to prevent collapse and ensure good generalizability and transferability of the model.

2.In line 230, the paper mentions M mini-batches. If these M mini-batches are assumed to be independently and identically distributed (i.i.d.), it raises concerns about how diversity between different mini-batches is guaranteed. If the diversity is insufficient, the assumption that supports the model’s effectiveness may not hold.

3.Meta-learning typically requires longer run times, raising questions about the relationship between the running time of this algorithm and that of the original methods. For instance, it would be beneficial to compare the time taken to run one epoch of SimCLR with the time required when incorporating the proposed algorithm. This comparison could provide insights into the practical implications of adopting the new method.

**Questions:**

Please see the weakness.

---

> ### Author Response · Authors · 2024-11-21
> **Responses for Weakness 1**
>
> **We sincerely appreciate the reviewer apsp's feedback and are encouraged by the presented paper's strengths. We will respond to each issue of the "Weaknesses" raised by the reviewer, hoping the following responses will eliminate the reviewer's concerns.**
>
> ### Responses for Weakness 1
>
> We agree with the reviewer that some concepts may seem overlapping. In fact, as mentioned in L57-67, the concept of universality was inspired by the evaluation methods used in most SSL and unsupervised representation learning approaches. However, we would like to emphasize that universality is the first unified concept proposed from the perspectives of discriminability, generalizability, and transferability in SSL, which is an important and fundamental concept that is different from previous work:
>
> **(i) Novelty**:
>
> 1. As mentioned in L33-48, most SSL methods, whether discriminative or generative, focus on identifying factors that contribute to effective representation learning. A long-standing challenge is defining what constitutes a "good" representation or model. Existing SSL methods typically evaluate effectiveness based on performance in various downstream tasks, considering it effective if the performance is good. This is precisely the "goal and practice of the current representation learning community" that the reviewer mentioned.
> 2. However, this approach introduces a deeper challenge: understanding why these methods yield higher-performing representations or models. As described in L125-131, while previous research (e.g., Wang et al., 2022, and Huang et al., 2021) theoretically demonstrated that SSL needs to constrain discriminability or explained its generalization ability across different downstream tasks from a task perspective (Ni et al., 2021; Eastwood et al., 2023; Balazevic et al., 2024), these explanations are limited to training and testing phases and fail to reveal how specific methods impact the quality of representations or models.
> 3. To address these issues, this work directly considers "what constitutes a good representation/model" rather than focusing on "what to do." Therefore, we explored the characteristics that a good representation/model should possess and, for the first time, proposed specific forms of discriminability, generalizability, and transferability in SSL, which together form the concept of universality. As mentioned in L180-191, this concept fundamentally differs from previous evaluations of universal representations.
> 4. Considering the definitions of discriminability, generalizability, and transferability, we provide a perspective for understanding "what SSL representation is" from three different aspects, distinguishing it from prior work that only focused on downstream task performance, as mentioned in L146-192 in Section 3.1.
>
> **(ii) Importance**:
>
> 1. First, once the definition and characteristics of a good representation/model are clarified, new SSL models can be designed accordingly. As done in this paper, when restated as an objective or loss function in SSL, SSL methods can directly guide the representation/model to possess discriminability, generalizability, and transferability during training, thereby endowing newly designed SSL methods with greater interpretability.
> 2. Furthermore, as mentioned in Section 3.1, Ahuja et al. (2020) and Wang et al. (2022) theoretically proved that to obtain good representations/models, SSL must constrain discriminability—i.e., intra-class compactness and inter-class separation. However, this does not explain why SSL models trained on one dataset can generalize to different downstream tasks. Ni et al. (2021) explained the generalizability of SSL methods from a task perspective but failed to address discriminability and generalizability. These limitations prompted us to propose a new understanding of SSL effectiveness. We first defined the concept of a good representation/model—universality—indicating that good representations should possess discriminability, generalizability, and transferability. We believe this definition of universality is crucial for the development of the SSL community.
> 3. Lastly, this concept inspired us to propose GeSSL. As described in Section 3.4 (Page 6), GeSSL integrates discriminability, generalizability, and transferability into SSL through the three steps outlined in Section 3.3, involving optimization objectives, parameter update mechanisms, and learning paradigms. Experiments across more than 25 baselines, 16 datasets, and five settings for both discriminative and generative SSL showed that GeSSL brought stable and significant performance improvements, accelerating model convergence (Tables 1-20, Figures 1-9). This empirical evidence demonstrates the effectiveness and importance of universality for SSL.
>
> Thus, although the concept of universality was inspired by the evaluation methods of the SSL paradigm, it is a valuable and novel concept that fundamentally differs from previous evaluations.

---

> ### Author Response · Authors · 2024-11-21
> **Responses for Weakness 2**
>
> (1) **Task Diversity Under i.i.d. Assumption**: we would like to point out that even if all mini-batch data are i.i.d., it does not imply that the resulting tasks are homogeneous. On the contrary, under the i.i.d. assumption, task diversity can still be ensured in the following ways:
>
> 1. **Complexity of Data Distribution**: The i.i.d. assumption does not require data to be simple or homogeneous. The distribution can be complex, covering multiple classes and diverse sample characteristics. For instance, in a multi-class task, data can come from various classes. Therefore, even if the data are sampled from the same distribution, the distribution itself can be complex enough to ensure task diversity, reflected by a rich feature space (e.g., high-dimensional data).
> 2. **Diversity Through Sampling**: Each mini-batch can be composed of different samples randomly drawn from the same distribution. This means that while the samples come from the same distribution, each mini-batch can have different sample combinations, with varying features, presenting different learning challenges to the model.
> 3. **Data Augmentation**: In many deep learning and self-supervised learning methods, data augmentation is used to create diverse training tasks. Even if samples are i.i.d., using different augmentation techniques (e.g., cropping, rotation, color transformation) can provide the model with diverse inputs.
> 4. **SSL Task Construction**: We treat each mini-batch as a multi-class task, with each original sample corresponding to a label. Thus, different tasks have different label distributions. Even if samples within a task are i.i.d., the label distribution varies across tasks, ensuring diversity.
> 5. **Theoretical Support**: Theoretically, the i.i.d. assumption does not restrict task homogeneity. Many theoretical works, such as [1] on statistical learning theory and [2] on representation learning, discuss how samples drawn independently from the same distribution can train models while maintaining task diversity and achieving generalization. These studies show that even under the i.i.d. assumption, tasks can encompass different data patterns and diverse features.
>
> [1] Statistical Learning Theory.
> [2] Representation Learning: A Review and New Perspectives.
>
> (2) **Ensuring Effectiveness Despite Data Scarcity or Homogeneous Tasks**: Even in cases of data scarcity or homogeneous tasks, not just data diversity, we have taken steps to ensure learning effectiveness:
>
> 1. **Definition Perspective**: In machine learning, data scarcity and homogeneous tasks can lead to overfitting to specific tasks, causing the model to learn all information, including background, making it hard to adapt to other tasks. Therefore, if we constrain the model to perform well on training data while maintaining effectiveness on unseen samples and tasks, we can ensure its robustness. As mentioned in L57-82 and L146-157, the defined universality considers both learning and evaluation levels, covering discriminability, generalizability, and transferability, involving known samples, unknown samples, and unseen tasks. Thus, even with data scarcity and homogeneous tasks, this definition ensures that the model learns a universal representation to maintain effectiveness.
> 2. **Modeling Perspective**: We further proposed GeSSL to model universality, including discriminability, generalizability, and transferability. As described in Section 3.4, for discriminability, GeSSL extracts key features from each mini-batch using limited data to achieve optimal performance. For generalizability, GeSSL ensures causal feature extraction during cross-task training. Finally, for transferability, GeSSL employs bi-level optimization to estimate the true task distribution from discrete training tasks. Thus, GeSSL models universality to ensure model effectiveness under limited data conditions.
> 3. **Empirical Perspective**: Experiments across over 25 baselines, 16 datasets, and five settings for both discriminative and generative SSL demonstrate stable and significant performance improvements with GeSSL, including in few-shot and cross-domain scenarios (Tables 1-20, Figures 1-9). This empirical evidence supports the effectiveness of our work in the face of data challenges.
>
> **Summary**: We mitigate the impact of data homogeneity on three levels. First, in practical applications, it is challenging to ensure all sampled tasks are i.i.d. with homogeneous samples. Second, even under the i.i.d. assumption, diversity can be ensured as discussed in (1). Third, even if task diversity is difficult to achieve, we have addressed this in the definition, measurement, and modeling of universality by constraining discriminability, generalizability, and transferability to ensure the effectiveness of learned features. Extensive experiments support the effectiveness of the proposed method.
>
> Therefore, even under the i.i.d. assumption mentioned by the reviewer, model effectiveness is still ensured.

---

> ### Author Response · Authors · 2024-11-21
> **Responses for Weakness 3**
>
> We have provided experimental results and detailed analysis in Section 5.6 of the main text (Page 9, Figure 2, L479-484) and Appendix G.2 (Page 31, Table 20, L1649-1665) in the original version. The results show that GeSSL significantly improves SSL performance and computational efficiency with the same batch size, while the impact of the increased parameter size of GeSSL is negligible. GeSSL uses approximate implicit differentiation with finite difference (AID-FD) for updates instead of conventional second-order derivatives for a single epoch. Moreover, GeSSL constructs a self-motivated target that guides the model to optimize more effectively for specific tasks. Thus, GeSSL can address the limits of the computational overhead. We further emphasized this in Appendix G.2 with the experimental results based on the reviewer’s valuable suggestion in the revised version.
>
>
>
> Finally, we would like to once again thank the reviewer apsp for his/her time and effort, and hope that the above responses address the concerns.

---

> ### Author Response · Authors · 2024-11-27
> **Response to  Reviewer apsp**
>
> We thank the reviewer e5gk’s feedback but respectfully disagree with the comments.
>
> The reviewer say that a model with good alignment and uniformity is, to some extent, already a good representation model. We agree with this. But, a question is that **why good alignment and uniformity can lead to a good representation model**? This is a remaining problem for existing SSL methods.
>
> Our proposed method is motivated by this problem, and **our contribution is to construct new SSL methods from an interpretable perspective.** Meanwhile, **the proposed Definition for Good Representation can provide guidance for proposing new SSL methods.**
>
> Therefore, we insist that our work is solid.

---

### Official Review · Reviewer_79r3 · 2024-11-01

**Soundness:** 3
**Presentation:** 2
**Contribution:** 2
**Rating:** 6
**Confidence:** 3

**Summary:**

This paper focuses on a news representation quality evaluation of self-supervised learning (SSL) frameworks. In particular, the authors first introduce a definition for SSL universality, which combines three available factors, including discriminability, generalizability, and transferability, together with a σ-measurement to quantify it. They next propose GeSSL, a novel framework that models universality by unifying a self-motivated target for discriminability, a multi-batch collaborative update mechanism for generalizability, and a task-based bi-level learning paradigm for transferability. Experimental results on benchmark datasets demonstrate the superior performance of GeSSL compared to the original baselines.

**Strengths:**

The paper is well-written and easy to follow. Its idea of introducing a definition of measurable universality for SSL settings and a novel SSL framework (GeSSL) is novel and interesting. The experimental results on several benchmark datasets consistently show the benefit of using GeSSL compared to the mentioned self-supervised baselines.

**Weaknesses:**

My major concerns are about the actual novelty and the theoretical part of the paper. In particular:
- The definition of universality (Def. 3.1) is not mathematically well-defined since it contains several unclear terms, such as "competitive performance", "quickly", "few samples", and "comparable performance". I suggest the authors provide precise mathematical definitions or quantitative thresholds for these terms. For example, what specific metrics or benchmarks would constitute "competitive performance"? What time or iteration limit defines "quickly"? This would help make the definition more rigorous.
- The definition of a "$\sigma$-measurement" (Def. 3.2) is not well-defined since we do not know the term $f_{\Phi_l}^{*}$. Also, it is not unclear how the KL term in Eq. (1) is computed. Could the authors explicitly define $f_{\Phi_l}^{*}$ and provide more details on how the KL divergence is calculated in practice, including any approximations or estimation techniques used?
- The main theoretical contribution of the paper (Theorem 4.1) is not very meaningful. In particular, assuming that the proof is true (I did not have enough time to carefully verify it), the inequality $\tilde f_\theta - f_\theta \leq 0$ does not imply that $\tilde f_\theta$ is a good approximation of $f_\theta$.  I suggest the authors provide additional theoretical results or empirical evidence to demonstrate the practical significance of this inequality. For example, what does this inequality imply about the performance or convergence of the algorithm in practice? Also, it would be better to provide the upper bound of $\|\tilde f_\theta - f_\theta\|$.
- As mentioned in the paper (lines 193-294), GeSSL seems to be a computationally expensive and not scalable method since it requires the computation of the second-order derivatives (Hessian) w.r.t the (high-dimensional) parameters. Could the authors provide a more detailed analysis of the computational complexity of their method, perhaps including comparisons with existing approaches? Also, can the authors discuss potential approximations or optimizations that could make the method more scalable for high-dimensional problems?

**Questions:**

Please see my comments about the weaknesses of the paper.

---

> ### Author Response · Authors · 2024-11-21
> **Responses for Weakness 1**
>
> **We sincerely appreciate the reviewer 79r3's feedback and are encouraged by the presented paper's strengths. We will respond to each issue of the "Weaknesses" raised by the reviewer in turn, hope the following responses will eliminate the reviewer's concerns.**
>
> ### Responses for Weakness 1
>
> We have added the formal descriptions of the mentioned terms to Definition 3.1 and thank the reviewer for the valuable suggestions:
>
> 1. **"competitive performance" & "comparable performance"**: The performance of model $ f_{\theta} $ on task $ X_{tr,l}^{aug} $ is considered competitive if and only if $ \mathcal{L}(f_{\theta}, X_{tr,l}^{aug}) \leq \mathcal{L}(f_{\phi_l}^*, X_{tr,l}^{aug}) + \epsilon $, where $ L(f_{\theta}, X_{tr,l}^{aug}) $ represents the loss of the model on the training mini-batch task, $ \mathcal{L}(f_{\phi_l}^*, X_{tr,l}^{aug}) $ represents the loss of the task-specific optimal model, and $ \epsilon > 0 $ is an arbitrary positive number.
> 2. **"quickly"**: Let $ T_{max} $ denote the maximum number of training iterations or epochs. Model $ f_{\theta} $ achieves "competitive performance" under the condition $ t \leq T_{max} $, where $ t $ represents the number of iterations or time required to reach competitive performance.
> 3. **"few samples"**: Defined as a fraction of the total available data. For a task $ X_{tr,l}^{aug} $ with $ N $ samples, the model achieves "competitive performance" using $ |X_{tr,l}^{aug}| = \alpha N $, where $ 0 < \alpha \ll 1 $.

---

> ### Author Response · Authors · 2024-11-21
> **Responses for Weakness 2**
>
> (1) **What $ f\_{\phi\_l}^{*} $ is**: As mentioned in L196-199, $ f\_{\phi\_{l}}^{\*} $ represents "task-specific optimal models (ground-truth with 100% accuracy)." In the definition of $\sigma(f\_{\theta}^{\*})$, $\pi(f\_{\phi\_l}^{\*}(x))$ is the output of the task-specific optimal model $ f\_{\phi\_l}^{*} $, which is assumed to output the ground truth (L199), i.e., the true one-hot vector corresponding to sample $x$.
>
> (2) **KL Divergence Calculation**:
>
> 1. **Formal Explanation**: From Eq. 1 and L208-210 of Definition 3.2, the KL term ${\rm KL}(\pi(f\_{\theta}^{\*}(x))|\pi(f\_{\phi\_l}^{\*}(x)))$ evaluates the difference between the output class probability distribution $\pi(f\_{\theta}^{\*}(x))$ of model $ f\_{\theta}^{\*} $ and the output distribution $\pi(f\_{\phi\_l}^{\*}(x))$ of the task-specific optimal model $ f\_{\phi\_l}^{\*} $. Considering that Section 2 treats an SSL mini-batch as a multi-class task, "$\pi$ is the auxiliary classification head" that outputs the class probability distribution for a sample. Specifically, $\pi(f\_{\theta}^{\*}(x))$ represents the predicted result of model $ f\_{\theta}^{\*} $, i.e., the predicted class vector. $\pi(f_{\phi\_l}^{\*}(x))$ represents the output of the task-specific optimal model $ f_{\phi_l}^{\*} $, which is assumed to output the ground truth (L199), i.e., the true one-hot vector of the label. Thus, as mentioned in L212-215, the KL term calculates the difference between the predicted class vector of model $ f_{\theta}^{*} $ for sample $x$ and the corresponding true label vector.
>
> 2. **Example**: Suppose a specific task $ X_{te,l}^{aug} $ contains four original images. After augmentation, we obtain eight samples corresponding to four classes (pseudo-labels), with two samples per class. Suppose sample $x$ belongs to the first class, so its true class probability distribution is $[1, 0, 0, 0]$, which is also the output of $ f_{\phi_l}^{\*} $. If $\pi(f_{\theta}^{\*}(x))$ outputs $[0.81, 0.09, 0.03, 0.07]$, indicating that $x$ is predicted to belong to the first class, the KL term measures the difference between $[0.81, 0.09, 0.03, 0.07]$ and the true label $[1, 0, 0, 0]$, i.e., $ D_{KL}(P || Q) = \sum_{i=1}^{4} P(i) \log\left(\frac{P(i)}{Q(i)}\right) = 0.0924 $.
>
> (3) **How $\sigma(f_{\theta}^{*})$ is Calculated in Practice**: First, "$\sigma(f_{\theta}^{\*})$ measures the performance gap between the trained SSL model $ f_{\theta}^{\*} $ and the task-specific optimal models," i.e., "$\sigma(f_{\theta}^{\*}) = \sum\_{X\_{te,l}^{aug} \in X_{te}^{aug}} \sum\_{x \in X_{te,l}^{aug}} {\rm KL}(\pi(f_{\theta}^{\*}(x))|\pi(f_{\phi_l}^{\*}(x)))$." Second, based on (2) 2, the KL term measures the performance of $ f_{\theta}^{\*} $ on each sample. Therefore, in practice, $\sigma(f_{\theta}^{\*})$ is calculated by evaluating the KL divergence between the output of model $ f_{\theta}^{\*} $ and the true class probability distribution across all samples ($\sum_{x \in X_{te,l}^{aug}}$) in all training tasks ($\sum_{X_{te,l}^{aug} \in X_{te}^{aug}}$).

---

> ### Author Response · Authors · 2024-11-21
> **Responses for Weakness 3**
>
> (1) Firstly, we would like to apologize for the concern caused by the omission of intermediate steps in the conclusion. In fact, we have provided the upper bound for this inequality in the proof in Appendix B (Pages 19-20), i.e., in Eq. 13, we obtain $\tilde{f}\_{\theta} - f\_{\theta} \le \frac{\beta}{\alpha} \mu(\tilde{f}\_{\theta}, f_{\theta}^l) + o(\beta(\alpha + \beta))$. Furthermore, as mentioned in L1077-1079, "As $\alpha$ and $\beta$ become small or even zero, the residuals vanish exponentially, where $o(\beta(\alpha + \beta)) \approx 0$." Additionally, when the model converges, we assume $\mu(\tilde{f}\_{\theta}, f_{\theta}^l) \approx 0$, where $\mu$ corresponds to the KL divergence. In summary, $\tilde{f}\_{\theta} - f\_{\theta} \le \frac{\beta}{\alpha} \mu(\tilde{f}\_{\theta}, f_{\theta}^l) + o(\beta(\alpha + \beta)) \approx 0$. Therefore, in Theorem 4.1, we use $\tilde{f}\_{\theta} - f_{\theta} \le 0$ to indicate that $\tilde{f}\_{\theta}$ is at least as good as (or better than) $f_{\theta}$ in terms of performance, with no degradation. This is also what we present in Theorem 4.1. Based on the reviewer's valuable suggestions, we supplemented the intermediate steps in the theorem to provide a clearer performance guarantee.
>
> (2) Secondly, as mentioned in Eq. 9 and Eq. 10 of Appendix B, this inequality measures the performance difference before and after introducing GeSSL into the SSL model, which is also illustrated in Section 4, L322-364. We have conducted experiments on this inequality in Appendix F.4, which use $\sigma$-measurement to evaluate $\tilde{f}\_{\theta} - f_{\theta}$ for specific tasks. The results indicate that $\tilde{f}\_{\theta} - f_{\theta} \le 0$, i.e., the model performance improves after GeSSL optimization.

---

> ### Author Response · Authors · 2024-11-21
> **Responses for Weakness 4**
>
> (1) **Computational Complexity**: First, regarding computational complexity, we have provided a detailed analysis in Section 5.6 of the main text (Page 9, Figure 2, L479-484) and Appendix G.2 (Page 31, Table 20, L1649-1665), including comparisons with existing methods. The results indicate that despite GeSSL being based on bi-level optimization, it significantly improves performance while enhancing computational efficiency. GeSSL guides the model to optimize more effectively for specific tasks, accelerating convergence by reducing computational costs per epoch.
>
> (2) **Scalability**: We would like to clarify that GeSSL is scalable and is not significantly limited by the computational costs associated with second-order derivatives. As mentioned in Section 5.6 and Appendix G.4, GeSSL uses approximate implicit differentiation with finite difference (AID-FD) for updates instead of conventional explicit second-order differentiation. This approach estimates derivatives using numerical finite differences, avoiding complex partial derivative expressions, which is particularly suitable for high-dimensional scenarios (Liu et al. (2018)). The results in Figure 4 support this. Additionally, as detailed in Appendix C and the code, GeSSL optimizes its handling of high-dimensional data through mini-batch splitting, multi-GPU operations, and load balancing, etc., ensuring scalability.
>
> (3) **Experimental Validation**: Lastly, we conducted extensive experiments across more than 25 baselines, 16 datasets, and five different settings for both discriminative and generative SSL. These experiments include: transfer learning on video data or pixel-level tasks like segmentation (Section 5.4 and Appendix F.2); unsupervised learning on large datasets such as ImageNet (Section 5.2 and Appendix F.1); and evaluations across different modalities and domains (Appendix F.5 and F.6). The consistent performance improvements of GeSSL in these scenarios demonstrate its capability to handle high-dimensional problems.
>
>
>
> Finally, we would like to once again thank the reviewer 79r3 for his/her time and effort, and hope that the above responses address the concerns.

---

> ### Author Response · Authors · 2024-11-23
> **Responses for "Official Comment by Reviewer 79r3"**
>
> Dear Reviewer 79r3,
>
> We sincerely appreciate for your feedback, which has greatly encouraged us. We are really glad to hear that our rebuttal helped address your concerns and that you increase your rating. Your insightful opinions indeed helped us a lot. For any other followed discussion, we are still more than happy to engage. Meanwhile, we have made improvements in the revised version according to your valuable suggestions, including:
>
> - For Weakness 1: We have added formal explanations to Definition 3.1, including the terms "competitive performance", "quickly", "few samples", and "comparable performance".
> - For Weakness 2: We have further clarified and emphasized the significance of $f_{\Phi_l}^{*}$ and $\pi$ in Section 3.2, and polished the relevant calculation details in Appendix B (L1035-1063).
> - For Weakness 3: We have incorporated the upper bound in the proofs into Theorem 4.1 of the main text.
> - For Weakness 4: We have emphasized the scalability of GeSSL in the main text.
>
> We would like to express our gratitude again for your recognition of our work and for the time and effort you have dedicated to reviewing it.
>
> Best regards,
> The Authors

---

### Official Review · Reviewer_e5gk · 2024-11-02

**Soundness:** 2
**Presentation:** 2
**Contribution:** 1
**Rating:** 3
**Confidence:** 3

**Summary:**

The paper aims to formalize the notion of "goodness of a representation". The authors introduce the definition comprising three simple characteristics of a representation to be good: (i) discriminability, (ii) generalization and (iii) transferability. Furthermore, the authors develop the methodology that instantiates this definition. This methodology is compatible with existing SSL approaches. The authors evaluate the proposed approach on a variety of tasks and in conjunction with a few SSL methods.

**Strengths:**

* The proposed methodology is compatible with many existing SSL methods and leads to their improvements.

**Weaknesses:**

* The presentation of the paper does not convincingly demonstrates the novelty behind the proposed definitions for the representations to be "good". Specifically, the stated principles are already the ultimate goal of the entire representation learning community, thus, it is not clear what specifically novel authors introduce in their narration.

* Given that the proposed methodology is applied to existing SSL methods, the results demonstrate only minor quantitative improvements upon the corresponding SSL method.

**Questions:**

* I suggest authors to restructure their tables and display the results of their method next to the corresponding used SSL method to simplify parsing the results.

* Given that authors propose new definitions and argue that the existing approaches do not meet these definitions, I would expect some experimental evidence towards to support this.

---

> ### Author Response · Authors · 2024-11-21
> **Responses for Weakness 1**
>
> **We sincerely appreciate the reviewer e5gk's feedback and are encouraged by the presented paper's strengths. We will respond to each issue of the "Weaknesses" and "Questions" raised by the reviewers in turn, hope the following responses will eliminate the reviewer's concerns.**
> ### Responses for Weakness 1
> As mentioned in L57-67, the concept of universality draws inspiration from the evaluation methods commonly used in SSL and unsupervised representation learning. However, we emphasize that universality is the first concept in SSL that addresses discriminability, generalizability, and transferability simultaneously, making it a fundamental and novel departure from previous work. Below, we outline its novelty and necessity.
>
> **(1) The Novelty of the Proposed 'Universality'**
>
> 1. As stated in Lines 33-48, both discriminative and generative SSL methods primarily focus on identifying factors that contribute to effective representation learning. However, a key question remains: what defines a "good" representation or model? Current SSL methods generally assess effectiveness by evaluating performance on various downstream tasks—precisely what the reviewer refers to as "the goal and practice of the current representation learning community."
> 2. This approach, however, presents a deeper challenge: understanding why these methods produce high-quality representations or models. As described in Lines 125-131, while some studies (e.g., Wang et al., 2022; Huang et al., 2021) have shown that SSL must constrain the discriminability of features or explain generalization from a task perspective (e.g., Ni et al., 2021; Eastwood et al., 2023; Balazevic et al., 2024), these explanations are limited to training and testing performance, making it difficult to understand how specific methodological choices influence representation quality.
> 3. To address these challenges, our work shifts from focusing on "what to do" in SSL to exploring "what constitutes a good representation/model." Thus, we identified the essential characteristics of an effective representation/model and, for the first time, proposed specific forms of discriminability, generalizability, and transferability in SSL, which together define universality. As explained in Lines 180-191, this concept fundamentally differs from prior evaluations of general representations, providing a more intrinsic perspective.
>
> **(2) The Importance of the Proposed 'Universality'**
>
> 1. Defining what makes a good representation/model enables us to design new SSL models accordingly. In this paper, by framing universality as an objective or loss function, SSL methods can be explicitly guided to ensure discriminability, generalizability, and transferability during training, resulting in more interpretable SSL models.
> 2. Moreover, as discussed in Section 3.1, Ahuja et al. (2020) and Wang et al. (2022) theoretically demonstrated that SSL must constrain feature discriminability (e.g., intra-class compactness and inter-class separation) to achieve good representations. However, this does not explain why SSL models trained on one dataset can generalize across different downstream tasks. While Ni et al. (2021) addressed generalization from a task perspective, they did not fully consider discriminability or generalizability. These gaps led us to redefine SSL effectiveness. We introduced the concept of universality, stating that good representations must exhibit discriminability, generalizability, and transferability. We believe that this definition is essential for advancing the SSL field.
>
> **(3) The Novelty and Importance of the Proposed Method GeSSL**
>
> 1. As described in Section 3.4 (Page 6), GeSSL incorporates discriminability, generalizability, and transferability into SSL through three key steps, outlined in Section 3.3: optimization objectives, parameter update mechanisms, and learning paradigms. Specifically, GeSSL employs Kullback-Leibler divergence to refine future state predictions for discriminability, cross-task learning to ensure generalizability by extracting causal features, and bi-level optimization to model transferability through a meta-learning approach. This is the first method to systematically integrate all three aspects—discriminability, generalizability, and transferability—into the SSL learning paradigm, demonstrating their theoretical and practical importance.
> 2. Extensive experiments on over 25 baselines, 16 datasets, and five settings show that GeSSL leads to significant and consistent performance improvements and accelerates convergence (Tables 1-20, Figures 1-9). These empirical results validate the effectiveness and importance of GeSSL in the SSL framework.

---

> ### Author Response · Authors · 2024-11-21
> **Responses for Weakness 2**
>
> Analyzing the experimental results on five settings and 16 datasets, we demonstrate the effectiveness of GeSSL from four perspectives:
>
> (1) **Performance**: As shown in Tables 1-18, Figure 2, and Figure 6, with GeSSL, the model's performance improved by over 2.1% compared to SOTA baselines in unsupervised and semi-supervised settings, by over 2.7% in transfer learning settings, and by 3.8% in few-shot learning settings. Especially in cross-domain few-shot learning, the average improvement reached 5.4%. For both discriminative and generative SSL, the introduction of GeSSL consistently led to performance improvements, demonstrating its effectiveness in OOD and data-scarce scenarios.
>
> (2) **Efficiency**: As shown in Figure 2 and Table 20, GeSSL reduced the computational cost required for SSL baselines to converge, guiding the model to optimize more effectively in specific tasks.
>
> (3) **Stability**: As shown across all tables and figures, the introduction of GeSSL resulted in stable performance improvements, regardless of baseline and task.
>
> (4) **Universality**: As shown in Figure 5 and Table 15, the universality score of the model increased by nearly threefold compared to the original baseline after introducing GeSSL.
>
> **Therefore, these empirical results demonstrate the stable and significant performance improvements brought by GeSSL from four perspectives, rather than "only minor quantitative improvements."**

---

> ### Author Response · Authors · 2024-11-21
> **Responses for Question 1**
>
> Thank you for suggesting this point, we will adjust the table arrangement, i.e., placing the "+GeSSL" results under the corresponding baselines according to the suggestion.

---

> ### Author Response · Authors · 2024-11-21
> **Responses for Question 2**
>
> (1) We have provided empirical support for this point in Appendix F.4 (Pages 27-29, L1444-1541, Figure 5, and Table 15).
>
> 1. Experimental Details: We constructed two sets of experiments. First, as the proposed $\sigma$-measurement is a provable metric, we quantified the universality score of existing SSL methods and verified that our proposed GeSSL indeed improves the model's universality. We selected two scenarios based on images and videos, covering 7 benchmark datasets and ten tasks. We evaluated the $\sigma$-measurement scores of different baselines before and after introducing GeSSL, as well as after training. Second, for the learning process, we constructed the metric $r$, which represents the accuracy of SimCLR before and after introducing GeSSL and the ratio of their effects on the CIFAR-10 dataset. We assessed the average performance of multiple  $f_{\theta}^l$  over the first 20-200 epochs of training. If  $r<1$, it indicates that the model performs better in each training epoch after introducing GeSSL.
> 2. Empirical Support: As mentioned in (Page 27-28, L1455-1500), Figure 5 shows that (i) the $\sigma$-measurement scores of existing SSL models are low and fail to achieve good performance across multiple domains and tasks, with "existing approaches do not meet these definitions"; (ii) after introducing GeSSL, the $\sigma$-measurement scores of SSL models significantly improve, enhancing universality. This satisfies evaluation universality. As mentioned in (Page 28, L1500-1511), Table 15 shows that after introducing GeSSL, "the model $f_{\theta}$ uses limited data during training and quickly achieves great performance across tasks." This satisfies learning universality.
>
> **Therefore, these experimental results across various tasks demonstrate that existing approaches do not meet these definitions, but GeSSL does.**
>
> (3) The experiments we constructed in Table 1-20 show that GeSSL, built according to the proposed definitions, stabilizes and significantly improves model performance, demonstrating the contribution of the defined framework to existing SSL work.
>
>
>
> Finally, we would like to once again thank the reviewer e5gk for his/her time and effort, and we hope that the above responses address the concerns.

---

> ### Author Response · Authors · 2024-11-25
> **Responses for "Official Comment by Reviewer e5gk"**
>
> We thank the reviewer e5gk’s feedback but respectfully disagree with the comments.
>
> ---
>
> Both the references provided by the reviewer and previous SSL works can be summarized as **do what leads to a good representation** (as elaborated in detail in "Responses for Weakness 1"). These works evaluate the effectiveness of **do what** through experiments, such as discriminative tasks, generalization tasks, and transferability tasks, to demonstrate its utility. However, they do not address the question of **what constitutes a good representation**. Specifically, while they validate the effectiveness of *"do what"*, **they fail to explain how the representation is transformed through this process**. In contrast, our work directly focuses on defining *a good representation*. **We formalize the concepts of feature discriminability, generalizability, and transferability as key characteristics of *a good representation* and propose corresponding learning objectives**. The most significant value of this approach lies in enabling the interpretation of representations based on these formally defined characteristics, i.e., **feature interpretability**. From this perspective, our contribution is substantial and meaningful.
>
> ---
>
> If the reviewer insists that such formalization is meaningless, then by the same reasoning, numerous research fields, including formal logic, formal verification, and others [1-6], would also lose their value. These fields fundamentally aim to formalize understanding into learnable objectives.
>
> ---
>
> Regarding experimental validation, our method achieves an average improvement of over 4% on various scenarios, which is a significant advancement in the field of self-supervised learning. Moreover, we have conducted extensive experiments across diverse tasks to demonstrate the effectiveness of our method (as elaborated in detail in "Responses for Weakness 2"). The Friedman test shows that, for significance levels of ${\alpha _F} = 0.1$, ${\alpha _F} = 0.05$, and ${\alpha _F} = 0.025$, our method significantly exceeds the critical values of the F-distribution. Thus, the experimental results strongly substantiate the validity of our work.
>
> ---
>
> Therefore, we firmly stand by our viewpoint and respectfully disagree with the reviewer e5gk’s assessment of our work. At the same time, we sincerely hope that AC and other reviewers can engage in a discussion on this matter.
>
> ---
>
> [1] Interpret Your Decision: Logical Reasoning Regularization for Generalization in Visual Classification
>
> [2] Systematic generalization: what is required and can it be learned?
>
> [3] Robustness of Nonlinear Representation Learning
>
> [4] Interpretable and accurate fine-grained recognition via region grouping.
>
> [5] A benchmark for interpretability methods in deep neural networks
>
> [6] Less is more: Fewer interpretable region via submodular subset selection

---

### Official Review · Reviewer_yE1V · 2024-11-03

**Soundness:** 2
**Presentation:** 3
**Contribution:** 2
**Rating:** 6
**Confidence:** 4

**Summary:**

The authors propose that a robust representation or model should exhibit universality, defined by three core attributes: (i) discriminability—performing well on training samples; (ii) generalization—achieving strong performance on unseen datasets; and (iii) transferability—adapting effectively to new tasks with distribution shifts. To explicitly capture universality within the self-supervised learning (SSL) process, the authors introduce a novel framework, GeSSL. GeSSL employs KL divergence to distill knowledge from future states into the current state, thus lowering training loss; it promotes generalizability by extracting causal features through multi-task learning (multiple mini-batches); and it incorporates a bi-level optimization mechanism to structure SSL learning in a meta-learning format, enhancing transferability.

**Strengths:**

The paper is easy to follow, though the abstract is somewhat overwhelming.

The authors conduct comprehensive experiments across a diverse range of applications and datasets, effectively demonstrating the proposed method's effectiveness.

**Weaknesses:**

1. Some important information, such as the classifier $\pi$, the parameter M and reference, is not well-described in the paper. Please see the questions section for more details. Clarifying these aspects would greatly enhance the reader's understanding of the paper.

&nbsp;

2. Section 2: Treating each mini-batch in the SSL training phase as a multi-class classification task is well-established in the SSL literature. For example, contrastive loss, widely used in frameworks like SimCLR and MoCo, functions as an instance discriminator [1,2,3]. The authors should acknowledge this in the text.


&nbsp;

3. I think as the authors state MEASUREMENT OF UNIVERSALITY is one of main contribution, F.4 should be appear as the result on the main text.

&nbsp;

[1] Oord, Aaron van den, Yazhe Li, and Oriol Vinyals. "Representation learning with contrastive predictive coding." arXiv preprint arXiv:1807.03748 (2018).

[2] Hjelm, R. Devon, et al. "Learning deep representations by mutual information estimation and maximization." arXiv preprint arXiv:1808.06670 (2018).

Tian, Yonglong, et al. "What makes for good views for contrastive learning?." Advances in neural information processing systems 33 (2020): 6827-6839.

**Questions:**

**Q1:** Line 149-153: "According to Theorem 9 in Ahuja et al. (2020), if a model achieves good performance across multiple different tasks, it can be considered to have learned causal representations. Moreover, Schölkopf et al. (2021) and Ahuja et al. (2023) conclude that causality in representations is a sufficient condition for generalizability. Thus, a generalizable representation should be causally invariant across multiple tasks, enabling the model to achieve very low loss on these tasks using the same representation"

The authors refer to Theorem 9 in Ahuja et al. (2020), but I was unable to find it in that paper. Furthermore, as I understand it in the context of causal representation, multitasking for causal representation implies using a shared representation, z=f(x), to perform multiple tasks on the same representation. However, in GeSSL, multitasking is applied across different mini-batches (i.e., different samples x), which do not yield the same representation. This approach may still allow the model to rely on spurious features for good performance. Could the authors clarify this?

&nbsp;

**Q2:** The authors repeatedly mention employing the classifier $\pi$ to output class probabilities for the mini-batch, but I couldn't find any part of the paper describing how this is obtained.

Additionally, if π is a classifier per mini-batch, it would need to be retrained at every iteration. On the other hand, if $\pi$ is a classifier for the entire dataset, does that imply prior knowledge of the task before training?

&nbsp;

**Q3:** (σ-measurement): When referring to π in the context of $\sigma$-measurement, does π denote tasks like LIN, SEMI, or AP for image-based examples? I’m unclear on how the model is evaluated using $\sigma$-measurement in the ablation study F.4: Universality of Existing SSL Methods. Could the authors clarify?

Additionally, since the authors identify the measurement of universality as a primary contribution, F.4 should appear as a main result in the core text.

&nbsp;

**Q4:** GeSSL processes M mini-batches simultaneously during training:
-I couldn’t find any information on the size of M or any ablation study regarding M.

- Since SSL performance is highly dependent on batch size, it would be beneficial if the authors compared GeSSL with a baseline that uses M× the original batch size.

In Table 20, the model analysis includes parameter size, training time, and performance, showing that GeSSL achieves better results than the baseline with less training time (in hours). However, GeSSL’s optimization is more intensive, utilizing ($\lambda=10$) and a second-order gradient-based approach. Does this mean GeSSL requires fewer epochs to converge? If so, Table 9 shows some baselines achieving better performance in 400 epochs than GeSSL does in 1000 epochs.

---

> ### Author Response · Authors · 2024-11-21
> **Responses for Question 1**
>
> **We sincerely appreciate the reviewer yE1V's feedback and are encouraged by the presented paper's strengths. We will respond to each issue of the "Questions" and "Weaknesses" raised by the reviewers in turn, hope the following responses will eliminate the reviewer's concerns.**
> ### Responses for Question 1
> (1) For the concept of causal representation with references:
>
> 1. We sincerely apologize for the confusion caused by the labeling error. In fact, we originally intended to refer to [1]. The reason for stating "According to Theorem 9 in Ahuja et al. (2020)" was because Theorem 1 in [1] references Theorem 9 from [2] (Page 7), which shows that causal invariance can be generalized to a wide range of unseen environments. Therefore, we mistakenly referred to it as "Theorem 9 in Ahuja et al. (2020)." We sincerely apologize again for this oversight.
>
> 2. To explain how "if a model achieves..." was derived from [1], we provide the following outline. First, in the theoretical framework of IRM, a core assumption is that causal representations exhibit invariance across different environments ([1], Section 2.1). After constructing an invariant predictor to formalize this assumption (Section 3.2), [1] proves in Theorem 1 that when a model's representations are consistent across different tasks or environments, it relies on causal mechanisms rather than environment-specific biases. Moreover, Theorem 9 from [2] (as referenced in Page 7 of [1]) further demonstrates the generalizability of this conclusion. Then, Theorem 2 further proves the cross-environment robustness of causal representations, i.e., "the model must perform well in both training and testing environments." This environment invariance, along with invariant risk minimization, jointly supports the statement in L149-153, "if a model achieves...".
>
> Thus, we can derive the statement "if a model achieves..." based on "Ahuja et al. (2020)".
>
> [1] Invariant Risk Minimization Games. https://arxiv.org/abs/2002.04692; [2] Invariant Risk Minimization.
>
> (2) We provide another outline to explain the concept of "causal representation" and how GeSSL ensures the learning of causal representations, to clarify any potential misunderstanding:
>
> 1. Concept of Causal Representation: As mentioned in (1), causal representation requires the model to satisfy environment invariance and risk minimization. In other words, the model should use the learned causal features to achieve optimal results across different tasks (i.e., environments). The goal is to extract content features that represent task labels while excluding environment-specific style features (such as background information) so that the model can accurately predict labels based on causal features.
> 2. Form of Causal Representation: For multiple tasks (i.e., different batches), different representations are generated, which include content features and style features. Content features are shared across tasks and contribute to decisions for all tasks, thus the model needs to learn them to improve performance in downstream tasks. Style features, on the other hand, are task-specific, contributing only to the current task and should not be learned by the model, as they could negatively impact performance in other tasks. Style features reflect differences between tasks, such as augmentation methods and background information, and these differences are crucial for learning causal representations—if all task distributions are identical, the model may incorrectly treat all features as causally relevant. Therefore, we expect different SSL task distributions (i.e., different representations), while constraining the model to achieve consistent and optimal performance across these tasks, ensuring that the representations only include shared content features, excluding style features.
> 3. Obtaining Causal Representation: GeSSL learns $f_\theta$ through multiple batch tasks, ensuring that the final model achieves environment invariance across all training tasks, thereby achieving optimal performance (risk minimization).
>
> Therefore, GeSSL leverages causal invariance to constrain the model to learn causal representations, thereby ensuring generalizability.

---

> ### Author Response · Authors · 2024-11-21
> **Responses for Question 2**
>
> (1) $\pi$ is the classification head embedded in the model. We have described and further emphasized $\pi$ and the related calculation in Sections 3.2, 3.3 and Appendix B.
>
> (2) $\pi$ is part of the model and is therefore updated along with the rest of the model. Before training, the entire model, including $\pi$, is randomly initialized without relying on prior knowledge.

---

> ### Author Response · Authors · 2024-11-21
> **Responses for Question 3**
>
> (1) We would like to clarify that, as mentioned on Page 4, L212-214, $\pi$ refers to the classifier, specifically the classification head embedded into the model, which outputs the class probability distribution for a sample, not the data itself.
>
> 1. We have stated in L209-210 (Definition 3.2) and Algorithm 1 that "$\pi$ is an auxiliary classification head used to generate class probability distributions."
> 2. Specifically, in Section 2, we reframe SSL from a task perspective, treating each mini-batch as a multi-class classification task. To obtain pseudo-labels for each sample, we introduce an additional classification head $\pi$ (see Definition 3.2, L209-210, and Algorithm 1) to output the class probability distribution for each sample. For instance, if a mini-batch contains four original images, after augmentation, we have eight samples corresponding to four classes (pseudo-labels), with two samples per class. For a particular sample, $\pi(f_{\theta}^{*}(x))$ may output $[0.81, 0.09, 0.03, 0.07]$, indicating that the sample is predicted to belong to the first class.
>
> (2) We provide an outline explaining how $\sigma$-measurement is used for evaluation (Appendix F.4):
>
> 1. **Calculation of $\sigma$-measurement score (Section 3.2, Page 4)**: $\sigma$-measurement evaluates the difference in performance between the learned model and the optimal model for each task. The optimal model is assumed to output the ground truth, and the performance difference is measured by the KL divergence between the predicted and optimal class probability distributions. In other words, $\sigma$-measurement compares the predicted labels to the true labels across all SSL tasks, specifically by comparing the class probability vector output by classifier $\pi$ to the true labels. For example, the difference between the predicted values $[0.81, 0.09, 0.03, 0.07]$ and the true labels $[1, 0, 0, 0]$.
> 2. **Experimental Setup (Page 27-28, L1447-1500)**: Take the mentioned LIN task with SimCLR (Figure 5 (a), Page 28) as an example, we first train SimCLR and SimCLR+GeSSL on the COCO dataset for 200 epochs (L1491-1492). A classification head is added after the feature extractor (L209-210, L245-246), and a new mini-batch is then input into SimCLR and SimCLR+GeSSL to obtain the class probability distributions for each sample. The KL divergence between the predicted and true distributions is calculated for both. After normalization, the scores for the LIN task in Figure 5 (a) are obtained. Other baselines and tasks are evaluated similarly.
>
> (3) We have integrated the experimental results from Appendix F.4 into the main text according to the reviewer's valuable suggestions.

---

> ### Author Response · Authors · 2024-11-21
> **Responses for Question 4**
>
> (1) We have provided the ablation study on $M$ in Appendix G.5 (Page 35, L1858-1882, Figures 7-8).
>
> 1. Experimental results in three different scenarios—unsupervised, semi-supervised, and few-shot—show that when $n=8$ (referred to "M") , a good trade-off performance is achieved, which is also our chosen hyperparameter setting.
> 2. Thanks to the reviewer's suggestion, we conducted a toy experiment to evaluate the performance of SimCLR and SimCLR+GeSSL under $M \times$ the original batch size. The settings are consistent with the trade-off experiments in Figure 2. The results are shown in the table below. The performance of SimCLR, after convergence on a larger training dataset, remains largely unchanged and is still lower than GeSSL.
>
> | Methods        | Top-1 Accuracy | Training Time |
> | -------------- | -------------- | ------------- |
> | SimCLR         | 90.8           | 5.2           |
> | SimCLR + GeSSL | 93.6           | 3.6           |
>
> (2) We provide an outline to explain why GeSSL accelerates convergence and address questions related to Table 9:
>
> 1. Although the optimization method used by GeSSL is more complex, one of its core goals is to accelerate model convergence, i.e., achieve greater performance improvement per unit of time. This does not imply that GeSSL always requires fewer epochs to reach the optimal result. In fact, as mentioned in Section 5.6 and Appendix G.4, GeSSL uses approximate implicit differentiation with finite difference (AID-FD) for updates instead of conventional explicit second-order differentiation. Moreover, GeSSL constructs a self-motivated target that guides the model to optimize more effectively in a specific task. Therefore, the efficiency improvement is reflected in the computational efficiency and effectiveness of updates per epoch, rather than simply reducing the total number of epochs.
> 2. We also have carefully compared Table 9, which shows that incorporating GeSSL consistently outperforms the baseline methods over 400 epochs, with further improvement at 1000 epochs. Moreover, the performance of the model with GeSSL over 400 epochs exceeds that of the corresponding baseline algorithm over 1000 epochs. If possible, we sincerely hope the reviewer can point out the results indicating "achieving better performance in 400 epochs than GeSSL does in 1000 epochs."

---

> ### Author Response · Authors · 2024-11-21
> **Responses for Weakness 1**
>
> Thanks for the reviewer's suggestions. We have provided detailed explanations in **Responses to Q1-Q4** and refined the manuscript, hoping to alleviate the concerns.

---

> ### Author Response · Authors · 2024-11-21
> **Responses for Weakness 2**
>
> We appreciate the reviewer for recommending these papers. We have carefully reviewed and cited them in Section 2 and discussed in Appendix G.5 due to space limitation.

---

> ### Author Response · Authors · 2024-11-21
> **Responses for Weakness 3**
>
> Thanks for the reviewer's valuable suggestions, we have incorporated the experimental results from F.4 into the main text.
>
> Finally, we would like to once again thank the reviewer yE1V for his/her time and effort, and we hope that the above responses address the concerns.

---

> ### Comment · Reviewer_yE1V · 2024-11-28
>
> Thank you for conducting the additional ablation study on $M$ in Appendix G.5. This effectively addresses my concern related to Q4.
>
> As the allowance for editing the paper submission may be closing soon, I recommend that the authors highlight the changes or additional sections in a different color for easier reference (this is optional and entirely up to the authors' preference).
>
> I am still a bit unclear about the classifier $\pi$ and not entirely convinced by the explanation of the causal-representation part. I may need some additional time to reflect on it and will provide further comments shortly.
>
> Best regards,
>
> Reviewer yE1V

---

> ### Author Response · Authors · 2024-11-28
> **Responses for "Official Comment by Reviewer yE1V"**
>
> Dear Reviewer yE1V:
>
> We appreciate your time and follow-up discussion. We are really glad to hear that our rebuttal helped address some of your concerns.
>
> We have highlighted the changes in the revised version according to your conductive suggestion. Additionally, thank you for pointing the concerns out, we would like to provide further clarification regarding $\pi$ and causal representation, hoping to address your concerns and confusion:
>
> ---
>
> For $\pi$:
>
> Because that we regard each mini-batch as a multi-class classification task. Also, for a pair in the augmentation dataset, the anchor sample is regarded as cluster center of a class, the remaining augmented sample in this pair is regarded as the possitive sample of this class, and the remaining other augmented sample in the augmentation dataset is regarded as negative samples of this class. In other word, given the anchor sample, the whole augmentation dataset is divided into a binary classification problem.
> Thus, we can use the contrastive loss to calculate the probability distribution of the possitive sample to the cluster center. We can also say that the output of the classifier $\pi$ is modeled by the contrastive loss. For a possitive sample, we denote the value of it as $a$, then, the corresponding one-hot vector for it is $[a, 1-a]$. Then, the whole multi-class classification task is divided into many binary classification problem, and each one corresponds to an anchor sample. As a result,  for all augmented samples, we can obtain its corresponding one-hot vector. In all, the classifier $\pi$ consists of  a classification head and a contrastive loss.
>
> ---
>
> For causal representation part:
>
> In this paper, we do not propose the theoretical analysis to why the learned representation is causal-representation. We obtained it directly based on existing theoretical analysis. According to the existing theoretical analysis, the key to obtain the causal-representation is that to learning a invarient feature extractor that is suitable for multiple domains/tasks. In this paper, each training mini-batch is regarded as a task or domain, and for all tasks, we learn a invarient feature extractor.
>
> ---
>
> For any other followed discussion, we are still more than happy to engage.
>
> Best regards,
>
> The Authors

---

### Official Review · Reviewer_ETz8 · 2024-11-05

**Soundness:** 3
**Presentation:** 3
**Contribution:** 3
**Rating:** 6
**Confidence:** 4

**Summary:**

This paper introduce a new self-supervised learning (SSL) framework to improve the universality by explicitly modeling discriminability, generalizability, and transferability. The method combines self-motivated target learning, multi-batch collaborative updates, and task-based bi-level optimization to better balance model performance across different tasks and domains. The approach is validated on multiple scenarios (unsupervised learning, semi-supervised learning, transfer learning, few-shot learning) showing consistent improvements over existing SSL methods.

**Strengths:**

1. Different from previous works that mainly focus on discriminability, the authors propose to use a self-motivated target and bi-level optimization to explicitly model universality properties, leading to more robust and transferable models.


2. The authors have conducted comprehensive experiments across multiple learning paradigms and datasets. Thorough ablation studies demonstrate the necessity of each component (self-motivated target, multi-batch updates, bi-level optimization). The theoretical analysis provides performance guarantees through the $\sigma$-measurement and proves convergence properties.

3. The proposed method can be applied to enhance various existing SSL methods. The extensive empirical results demonstrate consistent improvements when integrating GeSSL with different baseline approaches like SimCLR, MoCo, BYOL etc., showing the broad applicability of the proposed method.

**Weaknesses:**

1. It remains unclear to me why using KL divergence alone is sufficient for measuring distribution differences in the self-motivated target. The authors argue that KL divergence better reflects the uncertainty between distributions compared to other metrics like MSE or cross-entropy, but don't fully explain why this specific choice leads to optimal universality properties compared to alternatives like Wasserstein distance.

2. My another concern is how the method balances computational efficiency with model performance. While the authors demonstrate performance improvements, the bi-level optimization introduces additional memory overhead and computational complexity. The optimal trade-off between efficiency and universality lacks thorough analysis, especially for resource-constrained scenarios.

**Questions:**

1. Why choose KL divergence over alternatives like Wasserstein distance? Were there any empirical observations that led to this choice?

2. In Section 3.3, you mention K is typically set to 1 due to batch size and training complexity constraints. Could you elaborate on how sensitive the model performance is to different K values?

---

> ### Author Response · Authors · 2024-11-21
> **Responses for Weakness 1**
>
> **We sincerely appreciate the reviewer ETz8's feedback and are encouraged by the presented paper's strengths. We will respond to each issue of the "Weaknesses" and "Questions" raised by the reviewers in turn, hope the following responses will eliminate the reviewer's concerns.**
>
> ### Responses for Weakness 1
>
> Thank you for pointing it out. We have provided a detailed explanation in Appendix G.3 (L1665-1764, pages 31-33) on the advantages of KL divergence from both theoretical and empirical perspectives. Here is an outline:
>
> (1) In practical applications, "optimal universality properties" require that the chosen metric possesses the following characteristics: (i) accurately quantifies subtle differences between distributions; (ii) can be effectively used for model optimization, ensuring stable and efficient convergence to the global optimum; (iii) is applicable to a variety of complex distributions; (iv) has efficient computational performance. These four points collectively determine which metric is chosen.
>
> (2) From a theoretical perspective (pages 31-33, L1668-1737 and L1745-1764), KL divergence has key properties that meet the "optimal universality properties" outlined in (1) (L1746-1752):
>
> 1. Non-negativity and equivalence determination: KL divergence is always non-negative and equals zero only when the two distributions are identical. This ensures stability when handling subtle differences between complex distributions, fulfilling (i) and (iv).
> 2. Convexity and optimization efficiency: The convexity of KL divergence increases the likelihood of converging to the global optimum during optimization, avoiding local minima, which is crucial for high-dimensional learning problems [1], thereby satisfying (ii). [1] Approximating the Kullback Leibler divergence between Gaussian mixture models.
> 3. Generalization of entropy: As an extension of information entropy, KL divergence quantifies information loss and uncertainty, effectively measuring distribution differences across various applications, particularly in self-rewarding learning tasks, satisfying (iii).
>
> (3) In contrast, other metrics have significant limitations (L1752-1764):
>
> 1. MSE: Based on Euclidean distance, MSE is sensitive to outliers and ignores properties like non-negativity and normalization of probability distributions, which limits (i) and (iii).
> 2. Cross-entropy: Can be seen as a special case of KL divergence, but its effectiveness is limited when the true distribution is not a one-hot vector or when it involves continuous distributions, lacking the ability to finely measure complex distributions, thus limiting (i) and (iii).
> 3. Wasserstein distance: It captures the overall shape difference between distributions but has high computational complexity, especially in high-dimensional scenarios. Additionally, it requires the distribution to satisfy certain smoothness conditions, making it difficult to fulfill (iv).
>
> (4) Empirical results (pages 9-10, L484-L488, Figure 3, and page 33, L1737-1745) indicate that using KL divergence allows GeSSL to achieve the highest accuracy within the shortest training time, outperforming MSE, cross-entropy, or Wasserstein distance in effectively balancing accuracy and computational efficiency.
>
> In summary, KL divergence achieves the optimal balance between theoretical robustness and computational feasibility, aligning with the "optimal universality properties" and resulting in better model generalization and lower training costs. Thus, we chose KL divergence along to ensure a balance between accuracy and computational overhead.
>
> We have further emphasized this in the main text according to the reviewers' valuable suggestions.

---

> ### Author Response · Authors · 2024-11-21
> **Responses for Weakness 2**
>
> We have provided experimental results and a detailed analysis of the trade-off between computational efficiency and performance in Section 5.6 (Page 9, Figure 2, L479-484) and Appendix G.2 (Page 31, Table 20, L1649-1665). The results show that:
>
> > "despite GeSSL optimizes based on bi-level optimization, it significantly improves SSL performance and computational efficiency, while the impact of the increased parameter size of GeSSL is negligible."
>
> In fact, as mentioned in Section 5.6 and Appendix G.4, GeSSL uses approximate implicit differentiation with finite difference (AID-FD) for updates instead of conventional explicit second-order differentiation. Furthermore, GeSSL constructs a self-motivated target, which guides the model to optimize more effectively towards better performance in specific tasks. Therefore, GeSSL is able to manage the computational cost while effectively handling complex problems in high-dimensional data, free from the computational limitations typically associated with bi-level optimization.

---

> ### Author Response · Authors · 2024-11-21
> **Responses for Question 1**
>
> (i) The selection of KL divergence is based on theoretical analysis and empirical observations, as detailed in Section 5.6 (Page 9, L484-488 and Figure 3) and Appendix G.3 (Pages 31-33, L1665-1764).
>
> (ii) Please refer to **Responses for Weakness 1** for more analysis.

---

> ### Author Response · Authors · 2024-11-21
> **Responses for Question 2**
>
> We have provided the experimental results on the sensitivity of K values in Section 5.6 (Page 9, L475-479) and Table 5, covering model accuracy and training time. The results show that the model achieves optimal performance at $K=1$ and $\lambda=10$, with high accuracy and low computational cost. As K increases, the impact on model performance is minimal (also influenced by $\lambda$). Under the same K setting, the optimal accuracy for different $\lambda$ values varies by less than 1%, indicating ease of adjustment in practical applications.
>
> Finally, we would like to once again thank the reviewer ETz8 for his/her time and effort, and we hope that the above responses address the concerns.

---

### Author Response · Authors · 2024-11-28
**Global Response: Novelty**

Dear PC, SAC, AC, and Reviewers:

**We sincerely would like to express our great gratitude to you for the time and effort in this work.**

**First of all**, we are honored that this work has been recognized by the reviewers and received the Rating of (6,6,6,3,5), where the comments affirmed that our work is Novel and Innovative, Theoretically Robust, and Empirically Effective. Meanwhile, we have fine-tuned this paper according to the reviewers' valuable suggestions.

**Secondly**, after rebuttal and discussion, we are pleased that most concerns were addressed, with Reviewer 79r3 deciding to improve the score. For now, the only concern from Reviewer e5gk and Reviewer apsp (2/5 who give the negative rating),  lies in the differences between this work and previous SSL methods.  Considering that they did not respond to our further replies, we would like to discuss this issue again here to achieve consensus and clarify misunderstandings.

---

Reviewer e5gk stated:

> "I argue that reaching the properties is already an ultimate goal of community, thus, in essence modern evaluation protocols consider diverse downstream datasets within a given downstream task and diverse set of tasks for transfer."

Reviewer apsp stated:

> “ the authors **interpret** what constitutes a good representation/model through the lens of universality, specifically from the perspectives of discriminability, generalizability, and transferability. However, a model with good alignment and uniformity is, to some extent, already a good representation model.”

---

In this regard, we would like to clarify:

---

Both the mentioned references previous SSL works can be summarized as **do what leads to a good representation**. These works evaluate the effectiveness of **do what** through experiments, such as discriminative tasks, generalization tasks, and transferability tasks, to demonstrate its utility. However, they do not address the question of **what constitutes a good representation**. Specifically, while they validate the effectiveness of *"do what"*, **they fail to explain how the representation is transformed through this process**.

**Reviewer asap** says that a model with good alignment and uniformity is, to some extent, already a good representation model. We agree with this. But, a question is that **why good alignment and uniformity can lead to a good representation model**? For example, SimCLR has demonstrated through experiments that contrastive loss achieves strong performance across various tasks, but it does not explain why the loss is effective. [1] theoretically shows that contrastive loss can be interpreted as achieving alignment and uniformity in the feature space, but it does not explain why they are key to learning good representations. [2] proves that contrastive loss forms an upper bound of supervised loss, suggesting that minimizing contrastive loss can effectively minimize supervised loss and thus imply the learning of good representations. However, [1] assumes that positive and negative samples are perfectly selected, whereas the success of SimCLR and the extensions are not based on this assumption. Therefore, [1] does not sufficiently explain why SimCLR achieves good representations. This is a remaining problem for SSL.

[1] Understanding Contrastive Representation Learning through Alignment and Uniformity on the Hypersphere

[2] A Theoretical Analysis of Contrastive Unsupervised Representation Learning

In contrast, our work directly focuses on defining and interpreting *a good representation* (**which is affirmed by all the reviewers**). **We formalize the concepts of feature discriminability, generalizability, and transferability as key characteristics of *a good representation* and propose corresponding learning objectives**. The most significant value of this approach lies in enabling the interpretation of representations based on these formally defined characteristics, i.e., **feature interpretability**. From this perspective, our contribution is substantial and meaningful.

If Reviewer e5gk and apsp insist that such formalization is meaningless, then by the same reasoning, numerous fields like formal logic, formal verification, interpretable ML, etc., would also lose their value. They fundamentally aim to formalize understanding into learnable objectives.

Regarding experiments, our method achieves an average improvement of over 4% on various tasks, which is a significant advancement in SSL. We have conducted extensive experiments which all demonstrate the effectiveness. The Friedman test shows that, for significance levels of ${\alpha _F} = 0.1$, ${\alpha _F} = 0.05$, and ${\alpha _F} = 0.025$, our method significantly exceeds the critical values of the F-distribution. Thus, the experimental results substantiate the validity of our work.

---

Therefore, we insist that our work is solid.

Lastly, we would like to deeply thank you again for your time and effort on this work.

Best regards,

The Authors

---

### Note · Authors · 2025-05-25

I have read and agree with the venue's withdrawal policy on behalf of myself and my co-authors.

---

### Meta-Review · Area_Chair_Ximk · 2024-12-17

**Metareview:**

This paper characterizes a good representation in self-supervised representation learning should posess universality which consists of discriminability, generalization, and transferability. The proposeed GeSSL frameowrk explicitly models universality by using a self-motivated target for discriminability with KL divergence, a multi-batch collaborative update to aim for generalizable features, and task-based bi-level optimization to enhance transferability. The experiments show the improvement when added to previous self-supervised learning methods. Theoretical analysis is provided.

The main concern expressed by reviewers (including Reviewer e5gk) is overlap with existing self-supervised learning concepts. There were a lot of discussions during the rebuttal period regarding this point, but some reviewers remained unconvinced. Reviewer e5gk also expressed concerns about the methodological novelty, regarding the similarity with meta learning approaches such as MAML-like methods. Although a direct response was not provided regarding this point, Appendix H addresses this point to some extent (and the strong empirical results also support the proposed method.) Another reviewer pointed out that the rigor of the definitions was lacking, but the reviewer was satisfied with the rebuttal and raised the score to 6.

Overall, we had 3 borderline-positive scores (6, 6, 6) and two negative score (5, 3). We would like to recommend rejection at this time, since the paper needs to further differentiate and position the work better within the existing literature. We hope the authors can include a lot of the discussions during the rebuttal phase into future versions of the paper.

**Additional Comments On Reviewer Discussion:**

Please see the meta review.

---

### Decision · Program_Chairs · 2025-01-22

Reject